# Convergence of Two-Timescale Markovian Stochastic Approximations with Applications in Reinforcement Learning

**Vagul Mahadevan** [1]  **Claire Chen** [2]  **Shuze Daniel Liu** [3] [4]  **Shangtong Zhang** [5]

## Abstract

This work studies the convergence of two-timescale stochastic approximations (SA), a class of iterative algorithms that update two sets of parameters in fast and slow timescales respectively. Notable examples of two-timescale SA in reinforcement learning (RL) include temporal difference learning with gradient correction (TDC) and actor-critic methods. Previously, the stability (i.e., boundedness) and convergence of two-timescale SA were only established under i.i.d. noise. This work instead establishes the stability and convergence of two-timescale SA under Markovian noise, a setup that is more realistic in RL. Notably, we do not need to use any projection operator and the noise does not need to live in a compact space. Our key technical novelty is to control the fast timescale parameter with the running max of the slow timescale parameter, instead of with the current slow timescale parameter, as most prior works do. As a key application, we establish the first almost sure convergence of TDC with eligibility traces under off-policy learning with linear function approximation.

## 1. Introduction

Stochastic approximation (SA, Robbins & Monro (1951); Benveniste et al. (1990); Kushner & Yin (2003); Borkar (2009)) concerns algorithms that recursively and randomly update a vector of parameters. Prominent SA algorithms include stochastic gradient descent (Kiefer & Wolfowitz,

[1]Metron, Reston, VA, USA [2]Division Office Physics, Math and Astronomy, California Institute of Technology, Pasadena, CA, USA [3]Data Science Lab, MIT, Cambridge, MA, USA [4]Mitch Daniels School of Business, Purdue University, West Lafayette, IN, USA [5]Department of Computer Science, University of Virginia, Charlottesville, VA, USA. Correspondence to: Vagul Mahadevan <vagulgm@gmail.com>, Shangtong Zhang <shangtong@virginia.edu>.

*Proceedings of the 43rd International Conference on Machine Learning*, Seoul, South Korea. PMLR 306, 2026. Copyright 2026 by the author(s).

1952) and temporal difference learning (Sutton, 1988). Many SA algorithms involve computations of two sets of parameters. One set of parameters, referred to as the fast timescale, is updated with a large step size. The other set of parameters, referred to as the slow timescale, is updated with a small step size. A key characteristic is that the smaller step size is asymptotically negligible compared with the larger step size; i.e., asymptotically, the fast timescale parameter is updated as if the slow timescale parameter were static. The two-timescale structure can therefore be regarded as a special implementation of a nested loop, where the fast timescale corresponds to the inner loop while the slow timescale corresponds to the outer loop. The advantage of the two-timescale implementation over a nested loop implementation is that the slow timescale (the outer loop) is continually updated and does not need to wait for the completion of the fast timescale (the inner loop). These SA algorithms are called *two-timescale* SA (Borkar, 1997) and are common in reinforcement learning (RL, Sutton & Barto (2018)). For example, in actor-critic algorithms (Sutton et al., 1999; Konda, 2002), the fast timescale is the critic that estimates the value of the agent's policy, and the slow timescale is the actor, which is the policy itself.

A fundamental challenge of the theoretical analysis of two-timescale SA is showing almost sure convergence. The seminal work Borkar (1997) establishes almost sure convergence when (i) the noise in the two-timescale SA is i.i.d. and (ii) the stability of the two-timescale SA is assumed. Here, by stability, we mean the almost sure boundedness of the parameters. It is well known that stability is usually a prerequisite for establishing almost sure convergence with ODE approaches (Kushner & Yin, 2003; Borkar, 2009) and in many works on two-timescale SA, e.g., Karmakar & Bhatnagar (2018), stability is simply assumed. However, for many RL algorithms instantiating two-timescale SA, the noise is a Markov chain, and the stability cannot be assumed directly. Efforts have been made to relax both (i) and (ii). For example, Lakshminarayanan & Bhatnagar (2017) establish the stability of two-timescale SA with i.i.d. noise. Borkar et al. (2025); Liu et al. (2025b) establish the stability of single-timescale SA with Markovian noise. However, to our knowledge, no prior work can establish the stability (and thus convergence) of two-timescale SA

*Table 1.* Comparison between our work and representative works regarding stability and almost sure convergence of SA. Single-timescale is a special case of two-timescale, and i.i.d. noise is a special case of Markovian noise. By "coupled dynamics," we mean the fast and slow parameters impact each other, rather than only the slow parameters affecting the fast parameters. Notably, our work is the only one that applies to TDC($\lambda$).

| | Single Timescale | Two Timescale | Coupled Dynamics | No Projection | i.i.d. Noise | Markovian Noise | Noncompact Noise |
|---|---|---|---|---|---|---|---|
| Borkar (2009) | ✓ | | | | ✓ | ✓ | |
| Lakshminarayanan & Bhatnagar (2017) | ✓ | ✓ | ✓ | ✓ | ✓ | | |
| Karmakar & Bhatnagar (2021) | ✓ | ✓ | | ✓ | ✓ | ✓ | |
| Liu et al. (2025b) | ✓ | | | ✓ | ✓ | ✓ | ✓ |
| Panda & Bhatnagar (2025) | ✓ | ✓ | ✓ | | ✓ | ✓ | ✓ |
| **Our Work** | ✓ | ✓ | ✓ | ✓ | ✓ | ✓ | ✓ |

with Markovian noise, a gap that this work closes. Prior attempts include Karmakar & Bhatnagar (2021); Panda & Bhatnagar (2025). However, Karmakar & Bhatnagar (2021) assume that the two sets of parameters are not coupled and the Markovian noise evolves in a compact space. Panda & Bhatnagar (2025) use additional projections to enforce boundedness (and thus stability). Consequently, none of them applies to representative two-timescale RL algorithms such as temporal difference learning with gradient correction and eligibility traces (TDC($\lambda$), Sutton et al. (2009); Yu (2017)). By contrast, our result requires milder assumptions, as can be seen in Table 1. To summarize, this work makes the following three contributions:

1. We establish the stability of two-timescale SA with Markovian noise under mild assumptions (Theorem 3.2) without using additional projections.

2. Building on the stability, we further establish the almost sure convergence (Theorem 3.3), where the Markovian noise can evolve in a possibly unbounded and uncountable space.

3. Applying our new theoretical results, we establish the first almost sure convergence of TDC($\lambda$) under off-policy learning and linear function approximation (Theorem 7.2).

### 1.1. Technical Innovation

This work establishes the stability of two-timescale SA by extending the ODE@$\infty$ approach in Borkar & Meyn (2000). Essentially, this work can be viewed as a combination of Lakshminarayanan & Bhatnagar (2017) and Liu et al. (2025b). Techniques from Lakshminarayanan & Bhatnagar (2017) are used to address (i) and techniques from Liu

et al. (2025b) are used to address (ii). However, to enable this seemingly straightforward combination, we need to introduce a major methodological innovation. Namely, we connect the parameters in the two timescales by *bounding the parameter in the fast timescale with the running max of the parameter in the slow timescale*. This methodology is a novel contribution to the literature since prior works on two-timescale SA, such as Kushner & Yin (2003); Mokkadem & Pelletier (2006); Lakshminarayanan & Bhatnagar (2017); Dalal et al. (2018); Yaji & Bhatnagar (2020); Doan (2021a); Zeng et al. (2024); Panda & Bhatnagar (2025), either use stronger assumptions or show that the parameter in one timescale is bounded by the parameter in the other timescale at the same step. These requirements cannot be established when combining Lakshminarayanan & Bhatnagar (2017) and Liu et al. (2025b). By contrast, we show that a weaker condition is sufficient for combining Lakshminarayanan & Bhatnagar (2017) and Liu et al. (2025b). Specifically, the parameter in the fast timescale does not need to be bounded by the parameter in the slow timescale at the same step. Instead, it only needs to be bounded by the maximal parameter in the slow timescale seen so far, as the rescaling scheme in Lakshminarayanan & Bhatnagar (2017) uses the maximal scaling factor seen thus far. Precisely speaking, the methodological innovation is Lemma 3.1, which tracks the maximal slow parameter previously encountered.

## 2. Background

In this work, we study the general two-timescale SA scheme: given an initial $x_0 \in \mathbb{R}^{d_1}$ and $y_0 \in \mathbb{R}^{d_2}$, we recursively

generate sequences of vectors $\{x_n\}$ and $\{y_n\}$ by

$$x_{n+1} = x_n + \alpha(n)H(x_n, y_n, W_{n+1}) \qquad (1)$$
$$y_{n+1} = y_n + \beta(n)G(x_n, y_n, W_{n+1}) \qquad (2)$$

We also define for convenience the concatenation $z_n \doteq (x_n, y_n)$. Each $z_n$ is an element of $\mathbb{R}^{d_1+d_2}$.

In this scheme, $\{\alpha(n)\}_{n=0}^{\infty}$ and $\{\beta(n)\}_{n=0}^{\infty}$ are sequences of deterministic learning rates and $\{W_n\}_{n=1}^{\infty}$ is a sequence of random noise in a general space $\mathcal{W}$ (not necessarily compact), while $H : \mathbb{R}^{d_1+d_2} \times \mathcal{W} \to \mathbb{R}^{d_1}$ is a function that maps the current iterate $z_n$ and noise $W_{n+1}$ to the actual incremental update of $x$ (the fast timescale iterate) and $G : \mathbb{R}^{d_1+d_2} \times \mathcal{W} \to \mathbb{R}^{d_2}$ maps $z_n$ and $W_{n+1}$ to the incremental update of $y$ (the slow timescale iterate).

This is a two-timescale scheme since the stepsizes satisfy $\lim_{n \to \infty} \frac{\beta(n)}{\alpha(n)} = 0$. Thus, since we can consider $y_n$ to remain almost constant relative to $x_n$ for large $n$ in the fast timescale, the behavior of $x_n$ can be studied by analyzing the ordinary differential equation (ODE)

$$\frac{\mathrm{d}x(t)}{\mathrm{d}t} = h(x(t), y) \qquad (3)$$

where $y$ is a fixed constant and $h(x, y) \doteq \mathbb{E}[H(x, y, \omega)]$ (with respect to a specific probability distribution). In the slow timescale, we can consider $x_n$ to have already converged to a value that depends on $y_n$, so we study the ODE

$$\frac{\mathrm{d}y(t)}{\mathrm{d}t} = g(\lambda(y(t)), y(t)) \qquad (4)$$

where $\lambda(y)$ denotes the equilibrium of (3) (i.e., the $x$ such that $h(x, y) = 0$ for fixed $y$) and $g(x, y) \doteq \mathbb{E}[G(x, y, \omega)]$. The asymptotic behavior of the discrete, stochastic iterates $\{z_n\}$ can then be characterized by the continuous, deterministic trajectories of the ODEs (3) and (4). This is called the ODE method (see Borkar (2009) for a comprehensive background). For this ODE approximation to be valid, the iterates must be bounded; this is known as *stability*. One needs to show

$$\sup_n \|z_n\| < \infty \quad \text{a.s.},$$

which is widely considered challenging (Borkar, 2009), but stability is a critical prerequisite for convergence and many works in the realm of two-timescale SA rely on it (Borkar, 2009; Karmakar & Bhatnagar, 2018; Borkar, 2025; Hu et al., 2024; Borkar, 2025).

## 3. Main Results

We begin by giving a structural overview of the rest of the paper (see Figure 1 for a high-level visual). In this section, we will describe our assumptions and present our

main results on stability and convergence of two-timescale algorithms. In Section 4, we give a detailed overview of the proof of Lemma 3.1, which states that the current fast timescale iterate is bounded by the largest slow timescale iterate seen previously. This proof is a fast timescale analysis. In Section 5, we give an overview of the proof of Theorem 3.2, which establishes stability (boundedness) over all the iterates with probability one. This is a slow timescale analysis. We then utilize the stability results to prove Theorem 3.3, which shows the convergence of two-timescale SA algorithms under Markovian noise, in Section 6. Finally, in Section 7, we apply our results to prove the convergence of the off-policy RL algorithm TDC($\lambda$).

### 3.1. Assumptions

We now briefly discuss our assumptions (the full details are in Appendix B).

In **Assumption B.1**, we assume that our Markov chain $\{W_n\}$ has a unique stationary distribution, a standard assumption in RL.

In **Assumption B.2**, we also assume our learning rates follow standard SA conditions. Importantly, since we are analyzing a two-timescale scheme, we have $\lim_{n \to \infty} \frac{\beta(n)}{\alpha(n)} = 0$.

Then, in **Assumption B.4**, we make assumptions about the functions $H$ and $G$. For any $c \in [1, \infty)$, define

$$H_c(x, y, w) \doteq \frac{H(cx, cy, w)}{c}, G_c(x, y, w) \doteq \frac{G(cx, cy, w)}{c}.$$

The functions $H_c$ and $G_c$ are rescaled versions of the functions $H$ and $G$ and will be used to construct rescaled iterates, a key technique in the ODE method (see, e.g., Borkar & Meyn (2000); Borkar (2009)). Just as in those works, we need the existence of some sort of limiting functions $H_\infty$ and $G_\infty$ for $H_c$ and $G_c$ respectively, when $c \to \infty$.

In **Assumption B.5**, we take $H_c, G_c, H_\infty, G_\infty$ to be Lipschitz, so that their growth is well-characterized and to guarantee existence and uniqueness of the ODEs of interest, and define their respective expectations $h_c, g_c, h_\infty, g_\infty$ with respect to the stationary distribution of the Markov chain.

Then, in **Assumption B.6**, we define the limiting ODEs, which are central to the ODE method:

$$\frac{dx(t)}{dt} = h_\infty(x(t), y), \quad \frac{dy(t)}{dt} = g_\infty(\lambda_\infty(y(t)), y(t))$$

which have the unique globally asymptotically stable equilibria $\lambda_\infty(y)$ (where $\lambda_\infty : \mathbb{R}^{d_2} \to \mathbb{R}^{d_1}$ is a homogeneous Lipschitz map with $\lambda_\infty(0) = 0$) and 0 respectively. We define similar objects for $h$ and $g$.

Finally, in **Assumption B.7**, we take a natural assumption showing that over long timescales, the average of certain

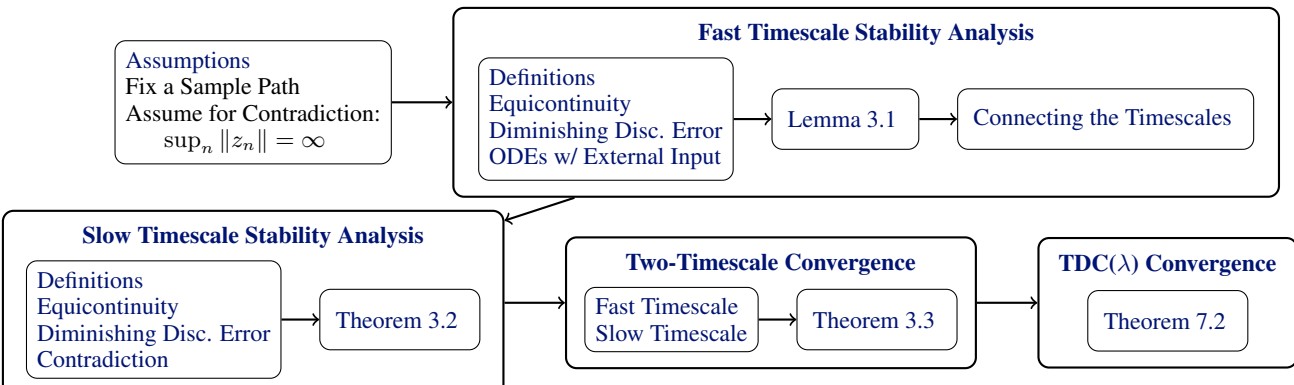

*Figure 1.* High-level proof roadmap of this work.

functions is close to the expectation. This regularity assumption about the noise is central to the averaging technique of Kushner & Yin (2003) that we apply.

These assumptions are well-supported in literature that studies stability of SA (Borkar, 2009; Lakshminarayanan & Bhatnagar, 2017; Liu et al., 2025b). We can easily verify that important RL algorithms like TDC($\lambda$) satisfy them rather than simply assuming that stability holds, so our results are widely applicable to general methods.

### 3.2. Results

We present our first result, which is a *key methodological innovation*:

**Lemma 3.1** (Max Slow Iterate Controls Fast Iterate)**.** *Let the Assumptions in Appendix B hold. Then, there exists a sample path dependent constant K such that for all n,*

$$\|x_n\| \le K(1 + \|y_n^{max}\|) \quad a.s.$$

*where $y_n^{max} \doteq y_m$ such that $m = \arg\max_{i \le n} \|y_i\|$ (with ties broken arbitrarily).*

We discuss its proof in Section 4. Intuitively, this result tells us that the size of the fast timescale iterates is bounded by the largest slow timescale iterate seen thus far. In the long history of the ODE method, no previous work has attempted to establish and use a result bounding one timescale by the maximum of the other to tie the timescales together (see the introduction for more detail). Having established that the slow timescale iterates control the fast timescale iterates, we are able to prove stability:

**Theorem 3.2** (Stability of Two-Timescale Iterates)**.** *The iterates $\{z_n\}$ are stable, i.e.,*

$$\sup_n \|z_n\| < \infty \quad a.s.$$

See Section 5 for a discussion of the proof. Once the boundedness of the iterates is established, convergence can be

shown. We can now guarantee that the iterates converge to a *bounded, nonempty* invariant set as the iterates evolve in a compact set, ensuring the existence of subsequential limits.

**Theorem 3.3** (Convergence of Two-Timescale Updates)**.** *Let Assumptions B.1 - B.7 hold. Then the iterates $\{x_n\}$ generated by* (1) *converge almost surely to the equilibrium depending on $\{y_n\}$ as follows:*

$$\lim_{n \to \infty} \|x_n - \lambda(y_n)\| = 0$$

*where $\lambda : \mathbb{R}^{d_2} \to \mathbb{R}^{d_1}$ is a Lipschitz map and $\lambda(y)$ is the globally asymptotically stable equilibrium of $\frac{dx(t)}{dt} = h(x(t), y)$. With this convergence established, we can then establish the main two-timescale convergence result:*

$$\lim_{n \to \infty} \|z_n - (\lambda(y^*), y^*)\| = 0.$$

The discussion of the proof is in Section 6. As with stability, we first establish a convergence result for the fast timescale and then the slow timescale separately. This type of result is the goal of the ODE method. In the proof of Theorem 3.3, we show convergence to bounded invariant sets, and the assumptions we place on the ODEs (in particular, the existence of unique globally asymptotically stable equilibria) characterize the invariant sets. Since $\frac{dy(t)}{dt} = g(\lambda(y(t)), y(t))$ has a unique globally asymptotically stable equilibrium, its invariant set is the singleton set $\{y^*\}$, ensuring convergence to the optimal value.

## 4. Bounding the Fast Timescale Iterates $x_n$

In this section, we will give a discussion of the proof of Lemma 3.1; see Appendix C for the full details. To prove Lemma 3.1, we conduct analysis in the fast timescale. This allows us to treat the slow timescale iterates as almost constant relative to the fast iterates. Note that the lemmas are derived on an arbitrary sample path $\{x_0, y_0, \{W_i\}_{i=1}^\infty\}$ such that the assumptions in Section 3 hold, so we omit "*a.s.*" from the lemma statements for conciseness.

## 4.1. Setting up the Timeline

We begin by splitting the positive real axis $[0, \infty)$ into chunks of length $\{\alpha(i)\}_{i=0,1,\ldots}$. We then collect these segments together into larger intervals $\{[T_n, T_{n+1})\}_{n=0,1,\ldots}$, where the sequence $\{T_n\}$ has the property that as $n$ grows large, we have $T_{n+1} - T_n \approx T$. We define

$$t(0) \doteq 0, \quad t(n) \doteq \sum_{i=0}^{n-1} \alpha(i) \quad n = 1, 2, \ldots$$

For all $T > 0$, define

$$m(T) = \max\{i | T \geq t(i)\}$$

as the maximal $i$ where $t(i)$ is no greater than $T$. We define

$$T_0 = 0, \quad T_{n+1} = t(m(T_n + T) + 1).$$

Intuitively, $T_{n+1}$ is a little to the right of $T_n + T$ on the real axis. As $n$ grows large, the interval $[T_n, T_{n+1})$ contains more time steps as $\alpha(i)$ is decreasing to 0.

## 4.2. Defining the Scaled Iterates

We start by fixing a sample path $\{x_0, y_0, \{W_i\}_{i=1}^{\infty}\}$. We take $\bar{z}(t)$ to be the piecewise constant interpolation of $z_n$ at points $\{t(n)\}_{n=0,1,\ldots}$, i.e.,

$$\bar{z}(t) \doteq (\bar{x}(t), \bar{y}(t)) \doteq \begin{cases} (x_0, y_0) & t \in [0, t(1)) \\ (x_1, y_1) & t \in [t(1), t(2)) \\ (x_2, y_2) & t \in [t(2), t(3)) \\ \ldots \end{cases}$$

Now we describe our rescaling of $\bar{z}(t)$.

**Definition 4.1** (Fast Timescale Rescaling). $\forall n \in \mathbb{N}, t \in [0, T + 1)$, define

$$r_n \doteq \max\{1, r_{n-1}, \|\bar{z}(T_n)\|\}, r_0 \doteq \max\{1, \|\bar{z}(0)\|\},$$

$$\tilde{z}_n(t) \doteq (\tilde{x}_n(t), \tilde{y}_n(t)) \doteq \frac{\bar{z}(T_n + t)}{r_n}.$$

This implies

$$\forall n \in \mathbb{N}, \|\tilde{z}_n(0)\| \leq 1.$$

Through the definition, we ensure that the sequence $\{r_n\}$ is monotonic, forcing the (to be defined) ODE discretization error to diminish over the entire sequence rather than just a subsequence, which is necessary in the two-time scale case.

We can regard $\tilde{z}_n(t)$ as the Euler discretization of $z_n(t)$ defined below.

**Definition 4.2.** $\forall n \in \mathbb{N}, t \in [0, T + 1)$, define $z_n(t) = (x_n(t), y_n(t))$ as the solution to the ODE system

$$\frac{dx_n(t)}{dt} = h_{r_n}(x_n(t), y_n(t)), \quad \frac{dy_n(t)}{dt} = 0$$

with initial condition $z_n(0) = \tilde{z}_n(0)$.

We want to show that the error of the discretization diminishes asymptotically. Precisely speaking, the discretization error is defined as $f_n(t) \doteq \tilde{z}_n(t) - z_n(t)$, and we want to show that $f_n(t)$ diminishes to 0 uniformly in $t$ as $n \to \infty$. So, we analyze the following three sequences of functions:

$$\{\tilde{z}_n(t)\}_{n=0}^{\infty}, \{z_n(t)\}_{n=0}^{\infty}, \{f_n(t)\}_{n=0}^{\infty}.$$

In particular, we show that they are all *equicontinuous in the extended sense*, as this is required to apply the Arzelà-Ascoli theorem (Appendix A.4). We defer the relevant definitions, statements, and proofs to the Appendix.

## 4.3. Selecting a Convergent Subsequence

To prove stability, we need to show that $\sup_n \|z_n\| < \infty$. Observe the inequality $\|z_{m(T_n)}\| = \|\bar{z}(T_n)\| \leq r_n$. Therefore, if we had $\sup_n r_n < \infty$, the result would come instantly. So, we assume that $\sup_n r_n = \infty$ to show that Lemma 3.1 holds even in this case. According to the Arzelà-Ascoli theorem in the extended sense (Appendix A.4), a sequence of equicontinuous functions always has a subsequence of functions that uniformly converges to a continuous limit. We use this to identify a subsequence of interest.

**Lemma 4.3.** *Suppose $\sup_n r_n = \infty$. Take an arbitrary subsequence $\{n_{k,0}\}_{k=0}^{\infty} \subseteq \{0, 1, 2, \ldots\}$. Then there is a subsequence $\{n_k\}_{k=0}^{\infty} \subseteq \{n_{k,0}\}_{k=0}^{\infty}$ such that there exist some continuous functions $f^{\lim}(t)$ and $\tilde{z}^{\lim}(t)$ such that $\forall t \in [0, T + 1)$,*

$$\lim_{k \to \infty} f_{n_k}(t) = f^{\lim}(t), \quad \lim_{k \to \infty} \tilde{z}_{n_k}(t) = \tilde{z}^{\lim}(t),$$

*where both convergences are uniform in $t$ on $[0, T + 1)$. Furthermore, let $z^{\lim}(t) = (x^{\lim}(t), y^{\lim}(t))$ denote the unique solution to the ODE system*

$$\frac{dx^{\lim}(t)}{dt} = h_{\infty}(x^{\lim}(t), y^{\lim}(t)), \quad \frac{dy^{\lim}(t)}{dt} = 0$$

*with the initial condition $z^{\lim}(0) = \tilde{z}_n^{\lim}(0)$. Then $\forall t \in [0, T + 1)$, we have*

$$\lim_{k \to \infty} z_{n_k}(t) = z^{\lim}(t),$$

*where the convergence is uniform in $t$ on $[0, T + 1)$.*

Its proof is in Appendix G.4. We will use the subsequence $\{n_k\}$ intensively.

### 4.4. Diminishing Discretization Error

We show that $\lim_{n\to\infty} \|f_n(t)\| = 0$ for all $t \in [0, T+1)$. We start by first proving that the discretization error diminishes along the sequence $\{n_k\}$, i.e., that

$$\lim_{k\to\infty} \|f_{n_k}(t)\| = \|f^{\lim}(t)\| = 0.$$

This means $\tilde{z}_{n_k}(t)$ is close to $z_{n_k}(t)$ as $k \to \infty$. From Lemma G.11, we have $\lim_{k\to\infty} \|f_{n_k}(t)\| = 0$ uniformly in $t$ on $[0, T+1)$, so the discretization error goes to 0 on $\{n_k\}$.

Because we are working with a two-timescale scheme, convergence to 0 along a subsequence will not be sufficient. As we defined $r_n$ to be monotonic, this enabled us to chose an arbitrary subsequence $\{f_{n_{k,0}}\}$ which itself had a subsequence $\{f_{n_k}\}$ that converges to 0. We can now use Lemma A.6 to upgrade to full sequence convergence, confirming that $\{f_n\}$ also converges to 0. Thus, the discretization error diminishes along the entire sequence. That is,

$$\lim_{k\to\infty} \|f_n(t)\| = 0$$

for all $t \in [0, T+1)$.

### 4.5. ODEs with External Inputs.

In the previous section, we showed that the discrete iterates generated by the two-timescale stochastic algorithm approximate the deterministic trajectories of certain ODEs. We now need to show that these trajectories of the ODEs exhibit certain behavior, including approaching their equilibria.

We present some notation concerning ODEs with external inputs. This is useful since, in the limit, we can treat the slow iterates as constant external inputs.

**Definition 4.4** (ODE Trajectory Notation)**.** We use the notation $\eta_c^{y(t)}(t, x)$ to denote the solution of the ODE

$$\frac{\mathrm{d}x(t)}{\mathrm{d}t} = h_c(x(t), y(t))$$

with the initial condition $x$. Under this notation, $x_n(t)$ can be written $\eta_{r_n}^{\tilde{y}_n(0)}(t, \tilde{x}_n(0))$.

The following lemma shows that within a certain amount of time, the trajectory of the ODE will be pulled close to the equilibrium determined by the external input.

**Lemma 4.5** (Lemma 1 from Chapter 3.2 of Borkar (2009))**.** *Let $K \subset \mathbb{R}^{d_1}$ be compact and fix $y \in \mathbb{R}^{d_2}$. Given any $\epsilon > 0$, there exists a $T_\epsilon > 0$ such that for all initial conditions $x \in K$, we have $\eta_\infty^y(t, x) \in B(\lambda_\infty(y), \epsilon)$[1] for all $t \geq T_\epsilon$.*

The next lemma shows that if the external input is close to some constant $y$, then the trajectory remains close to the trajectory we would obtain if $y$ were the external input.

---

[1] $B(x, r)$ denotes the open ball centered at $x$ with radius $r$.

**Lemma 4.6** (Lemma 2 from Chapter 3.2 of Borkar (2009))**.** *Let $y \in \mathbb{R}^{d_2}, [0, T]$ be a given time interval, and $\rho$ be a small positive constant with $y'(t) \in B(y, \rho)$. Then,*

$$\left\| \eta_c^{y'(t)}(t, x) - \eta_\infty^y(t, x) \right\| \leq (L\rho + \epsilon(c))Te^{LT}.$$

*for any initial $x \in \mathbb{R}^{d_1}$ and for all $t \in [0, T]$.*

### 4.6. Completing the Proof.

The most involved result left is Lemma 4.7, which tells us that if the $x$ component is too much larger than the $y$ component, the norm of the iterate at $T_{n+1}$ cannot be larger than that of the iterate at $T_n$. We use the results about ODEs with external inputs to ensure that the $x$ component of the trajectory gets pulled to the equilibrium, which is close to 0.

**Lemma 4.7** (Limiting Growth of $\|\bar{z}(T_n)\|$)**.** *There exists a $C_1 > 1$ such that if $\|\bar{x}(T_n)\| > C_1(1 + \|\bar{y}(T_n)\|)$, then $\|\bar{z}(T_{n+1})\| \leq \|\bar{z}(T_n)\|$. Consequently, $r_{n+1} = r_n$.*

Lemma 4.8 tells us that the iterate at each $T_n$ is bounded by the largest $y$ component seen so far in the sequence of times $T_n$. To do this, we use an inductive argument. Intuitively, we combine the previous lemma, which prevents $z$ from growing if the $x$ component is too much larger than the $y$ component, with Lemma H.15 which prevents $z$ from growing too much within one period $T$, ensuring that $x$ can never get too much larger than $y$.

**Lemma 4.8** (Boundedness of $\|\bar{z}(T_n)\|$)**.** *There exists a constant $C_2$ such that for all $n$,*

$$\|\bar{z}(T_n)\| \leq C_1 C_2 (\max_{m \leq n} \|\bar{y}(T_m)\| + 1).$$

Finally, while we have ensured that for all $T_n$, $\bar{z}(T_n)$ is not too much larger than $\bar{y}(T_n)$, we need the result to also hold for the iterates that lie in between $T_n$ and $T_{n+1}$. Lemma 4.9 takes care of this.

**Lemma 4.9** (Boundedness of $\|x_n\|$)**.** *There exists a constant $C_3$ such that for all $n$,*

$$\|x_n\| \leq C_1 C_2 C_3 (\|y_n^{max}\| + 1),$$

By setting $K$ equal to $C_1 C_2 C_3$, this concludes the proof of Lemma 3.1.

**Connecting the Timescales.** We present the following result, showing that eventually, in the fast timescale, the fast iterates will track the equilibria $\lambda_\infty(\tilde{y}_n(t))$. In the fast timescale analysis, the slow iterates were regulated by the fact that the slow step sizes are small. However, in the slow timescale analysis, the fast step sizes can grow large. This result regulates the behavior of the fast iterates.

**Lemma 4.10.** *For any $\epsilon > 0$, there is some $T_\epsilon$ and some $N_\epsilon$ such that for all $n > N_\epsilon$, $\|\tilde{x}_n(t) - \lambda_\infty(\tilde{y}_n(t))\| \leq \epsilon$ for all $t \in [0, T_\epsilon]$.*

## 5. Stability of the Two-Timescale Iterates

We give the proof of Theorem 3.2, restated below:

**Theorem** (3.2 Restated). *The iterates $\{z_n\}$ are stable, i.e.,*

$$\sup_n \|z_n\| < \infty \quad a.s.$$

We now conduct analysis in the slow timescale, but the structure is very similar to the fast timescale analysis, so we only highlight the key differences. Note that we are still working with the same sample path $\{x_0, y_0, \{W_i\}_{i=1}^{\infty}\}$ from the last section.

We again split the positive real axis $[0, \infty)$, but this time into chunks of length $\{\beta(i)\}_{i=0,1,\ldots}$ instead of $\{\alpha(i)\}_{i=0,1,\ldots}$. We then redefine the scaled iterates for the slow timescale. This time the scaling factor is defined differently, so that it will be at least as large as the largest iterate seen so far:

**Definition 5.1** (Slow Timescale Rescaling).

$$r_n \doteq \max \left\{ 1, \left\| z_{m(T_n)}^{max} \right\| \right\}$$

where $z_n^{max} \doteq z_m$ such that $m = \arg\max_{i \leq n} \|z_i\|$ (with ties broken arbitrarily).

Defining the scaling factor this way in the slow timescale is an innovation. It ensures that at each time $T_n$, the iterate, when scaled by the slow timescale's $\{r_n\}$ values, is no larger than when scaled by the appropriate fast timescale's $\{r_n\}$ values. We need this to link the timescales in Lemma 5.3.

We can regard $\tilde{y}_n(t)$ as the Euler's discretization of $y_n(t)$ defined below.

**Definition 5.2.** $\forall n \in \mathbb{N}, t \in [0, T+1)$, define $y_n(t)$ as the solution to the ODE

$$\frac{dy_n(t)}{dt} = g_{r_n}(\lambda_\infty(y_n(t)), y_n(t))$$

with initial condition $y_n(0) = \tilde{y}_n(0)$.

The discretization error is then defined as $f_n(t) \doteq \tilde{y}_n(t) - y_n(t)$; just considering the slow component. Again, we show equicontinuity (here Lemma 3.1 plays a big role in controlling the growth of the fast iterates) and apply Arzelà-Ascoli to obtain a convergent subsequence.

To show that the discretization error diminishes, we must prove an involved result. It requires us to work in both timescales to show that although the iterates get rescaled less often in the slow timescale, they still do not grow too much faster than the iterates in the fast timescale. So, we show that even in the slow timescale, the fast timescale iterates converge to the equilibria dependent on the slow timescale iterates, formalizing the intuition that in the slow timescale, we can consider the fast iterates to have converged:

**Lemma 5.3** (Fast Iterates Approach Equilibrium). *For all $\epsilon > 0$, there exists some $N$ such that for all $n > N$,*

$$\|\tilde{x}_n(t) - \lambda_\infty(\tilde{y}_n(t))\| < \epsilon$$

*for all $t \in [0, T]$.*

We can now deal with the error term (41) unique to the slow timescale, and show the discretization error decreases to zero (42). We then obtain ODE results similar to single-timescale stability results (Chapter 3 of Borkar (2009)). As the limiting ODE is attracted to its globally asymptotically stable equilibrium, 0, our iterates cannot grow without bound. Thus, the sequence $r_n$ is bounded, creating a contradiction. We conclude that Theorem 3.2 holds true.

## 6. Convergence of the Two-Timescale Iterates

We discuss the proof of Theorem 3.3, restated below:

**Theorem** (3.3 Restated). *Let Assumptions B.1 - B.7 hold. Then the iterates $\{x_n\}$ generated by (1) converge almost surely to the equilibrium depending on $\{y_n\}$ as follows:*

$$\lim_{n \to \infty} \|x_n - \lambda(y_n)\| = 0$$

*where $\lambda : \mathbb{R}^{d_2} \to \mathbb{R}^{d_1}$ is a Lipschitz map and $\lambda(y)$ is the globally asymptotically stable equilibrium of $\frac{dx(t)}{dt} = h(x(t), y)$. With this convergence established, we can then establish the main two-timescale convergence result:*

$$\lim_{n \to \infty} \|z_n - (\lambda(y^*), y^*)\| = 0.$$

The proof mirrors the stability proof but is simpler. We first show a convergence result in the fast timescale. Namely, we show that the fast iterates $x_n$ converge to $\lambda(y_n)$. Then, we obtain the full convergence result through analysis in the slow timescale. The difference is that we no longer need to use rescaling, since we know that the iterates are bounded by Theorem 3.2, and we now prove results on $t \in (-\infty, \infty)$ as this simplifies showing convergence to an invariant set.

First, we show that the discretization error between $\{z_n\}$ and the ODE system

$$\frac{dx(t)}{dt} = h(x(t), y(t)), \quad \frac{dy(t)}{dt} = 0 \qquad (5)$$

diminishes in the fast timescale. Then, due to Theorem 3.2, we argue that since the set of limit points $Z$ of $\{z_n\}$ is bounded and nonempty, it is an invariant set of (5). We then show that no subsequence of $\{z_n\}$ converges to a point not in $Z$, confirming that $\{z_n\}$ converges to $Z$. Since the invariant set of (5) is $\{(\lambda(y), y) : y \in \mathbb{R}^{d_2}\}$, we have

$$\lim_{n \to \infty} \|x_n - \lambda(y_n)\| = 0.$$

We then pursue the slow timescale argument and show that $\{y_n\}$ converges to the invariant set of the ODE

$$\frac{\mathrm{d}y(t)}{\mathrm{d}t} = g(\lambda(y(t)), y(t))$$

which is the singleton set containing $y^*$, the equilibrium of the ODE. This, combined with the fast timescale convergence result, yields our desired result:

$$\lim_{n \to \infty} \|z_n - (\lambda(y^*), y^*)\| = 0.$$

# 7. Applications in Reinforcement Learning

In this section, we give some context on RL and prove the convergence of an important off-policy RL algorithm, TDC($\lambda$), using our main results (Section 3).

We consider a Markov Decision Process (MDP) with state space $S$, action space $A$, reward function $r : S \times A \to \mathbb{R}$, transition function $p : S \times S \times A \to [0, 1]$, and discount factor $\gamma \in [0, 1)$. At time $t$, an agent with a control policy $\pi : S \times A \to [0, 1]$ will, given the current state $S_t$, sample an action $A_t \sim \pi(\cdot|S_t)$. Upon taking action $A_t$, the environment yields the reward $R_t \doteq r(S_t, A_t)$ to the agent and transitions to the next state $S_t \sim p(\cdot|S_t, A_t)$. We define the return at time $t$ as $G_t \doteq \sum_{i=t}^{\infty} \gamma^{i-1} R_t$. Based on this, we can define the value function $v_\pi(s) \doteq \mathbb{E}_{\pi,p}[G_t|S_t = s]$. Since the state space $S$ could be quite large, we wish to approximate $v_\pi(s)$ using a vector of parameters $\theta \in \mathbb{R}^K$ where $K \ll |S|$. Specifically, we define $v_{\theta,\pi}(s) \doteq \phi(s)^\top \theta \approx v_\pi(s)$, where $\phi : S \to \mathbb{R}^K$ is the function which extracts the features from the state $s$.

The task of computing or approximating $v_\pi(s)$ is called *policy evaluation*, and it is one of the most important problems in RL. Temporal Difference (TD) learning methods (Sutton, 1988) have been the most effective at this task. They operate through *bootstrapping*, the practice of updating estimates using other estimates. In the case of TD methods, they update the estimated value of a current state using the estimated value of sampled future states.

Another important technique is *off-policy learning*, where we estimate the value of a target policy $\pi$ by using data from a behavior policy $\mu$. This has many advantages as it enables us to run many policies in parallel and train offline, improve data-efficiency (Sutton et al., 2011; Liu & Zhang, 2024; Liu et al., 2025a;c), and enforce safety by using a safe behavior policy to evaluate a riskier target policy (Dulac-Arnold et al., 2019; Chen et al., 2025).

Unfortunately, when combining bootstrapping and off-policy learning with function approximation (necessary in large state spaces), naive TD methods can diverge. This pitfall is called the deadly triad (Sutton & Barto, 2018; Zhang, 2022). Gradient temporal difference learning (GTD) was

developed to break the deadly triad (Sutton et al., 2008). While convergent, it is slow. Temporal Difference learning with gradient correction (TDC) was first proposed in Sutton et al. (2009) as a modification of GTD that is nearly as fast as standard TD but still convergent. In addition to having the parameter vector $\theta$ for the linear approximation of the value function, it has a second parameter vector $\nu$ to estimate one of the expectations in the algorithm. TDC is a two-timescale algorithm as the updates to $\nu$ run on a faster timescale, which empirically leads to faster convergence of the algorithm. In this work, we study TDC($\lambda$), i.e., TDC with eligibility traces. Eligibility traces are a powerful tool for credit assignment, a critical challenge in RL, and have been a fundamental part of RL since the inception of the field (Barto & Sutton, 1981). Although eligibility traces empirically speed up convergence (Sutton & Barto, 2018), they can introduce difficulties in analysis.

**Definition 7.1** (TDC($\lambda$)).

$$
\begin{aligned}
e_t =& \lambda \gamma \rho_{t-1} e_{t-1} + \phi_t, \quad (6) \\
\delta_t =& R_{t+1} + \gamma \phi_{t+1}^\top \theta_t - \phi_t^\top \theta_t, \\
\nu_{t+1} =& \nu_t + \alpha_t \left( \rho_t \delta_t e_t - \phi_t \phi_t^\top \nu_t \right), \\
\theta_{t+1} =& \theta_t + \beta_t (\rho_t \delta_t e_t - \rho_t (1 - \lambda) \gamma \phi_{t+1} e_t^\top \nu_t).
\end{aligned}
$$

The eligibility trace $e_t$ is an exponential average of the features $\phi_t \doteq \phi(S_t)$ (weighted by importance sampling ratio $\rho_t$, which is necessary in off-policy learning). $\delta_t$ is the TD error. $\theta_t$ and $\nu_t$ parametrize the approximations of the value function and of the gradient correction respectively. The version of TDC($\lambda$) we study in Definition (6) is referred to as GTDb in Yu (2017).

**Theorem 7.2.** *Take the assumptions in Appendix F. Then, by applying Theorem 3.3, TDC with eligibility traces converges almost surely.*

*Remark* 7.3. One can apply Theorem 3.3 similarly to show that two-timescale GTD with eligibility traces (GTDa in Yu (2017)) also converges.

The assumptions are not very strong. We assume the state space $S$ and the action space $A$ are finite, the Markov chain $\{S_t\}$ is irreducible, and that from any state, all possible actions have a positive probability of being chosen (as this is an off-policy algorithm, this ensures coverage of the target policy we are estimating the value of). These assumptions are well-established in the literature (Yu, 2017; Liu et al., 2025b). From this, it is easy to verify that TDC($\lambda$) satisfies all assumptions in Appendix B, so Theorem 3.3 applies. Applying Theorem 3.3, we confirm that TDC with eligibility traces converges (see Appendix F for the detailed proof).

## 8. Related Work

This work is primarily concerned with the almost sure convergence of two-timescale SA in an asymptotic sense without a rate. For single-timescale SA, almost sure convergence rates are also established for specific algorithms such as $Q$-learning (Szepesvári, 1997), linear $Q$-learning (Liu et al., 2025d), and linear TD (Tadić, 2002), for linear SA (Chong et al., 1999; Tadic, 2004; Kouritzin & Sadeghi, 2015), and for general SA under various noise conditions (Koval & Schwabe, 2003; Vidyasagar, 2023; Karandikar & Vidyasagar, 2024). The most general ones are Qian et al. (2024); Liu et al. (2026), which establish almost sure convergence rates for general contractive SA under Markovian noise. Extending these results to two-timescale SA is a direction for future work. The asymptotic almost sure convergence of general contractive single-timescale Markovian SA is also recently formally verified in Lean by Zhang (2025). Extending this formal verification framework to two-timescale SA is also a direction for future work.

In addition to almost sure convergence, there are many other modes of convergence, e.g., $L^2$ convergence. The $L^2$ convergence rates of two-timescale SA are established under various noise and algorithmic conditions (Doan, 2021a;b; 2022). One particularly relevant work is Chandak et al. (2025). The main result of Chandak et al. (2025) is an $L^2$ convergence rate for two-timescale Markovian SA. But Chandak et al. (2025) also have a claim about the almost sure convergence of two-timescale Markovian SA, exactly overlapping with our main result. We believe that their claim about almost sure convergence does not hold. Particularly, their proof of almost sure convergence relies crucially on their claim that if the iterates are bounded in expectation, then they are bounded almost surely (i.e., $\sup_n \mathbb{E}\left[\|x_n\|^2\right] < \infty \implies \sup_n \|x_n\| < \infty$ a.s.), which is false. A counterexample (e.g., $x_n = \sqrt{n}$ with probability $1/n$ and $0$ otherwise) can be easily constructed by the second Borel-Cantelli lemma.

TDC$(\lambda)$ is one representative two-timescale RL algorithm. Other two-timescale RL algorithms include actor-critic algorithms (Sutton et al., 1999; Konda & Tsitsiklis, 1999; Zhang et al., 2020; 2022), average-reward gradient TD algorithms (Zhang et al., 2021a), and TD with target networks (Zhang et al., 2021b). Investigating whether our results can be applied to these algorithms is a possible future work. Many previous convergence analyses of TDC do not handle eligibility traces (Sutton et al., 2008; Xu et al., 2019), i.e., they set $\lambda = 0$. The only prior work that handles eligibility traces properly for TDC is Yu (2017), which, however, only analyzes projected variants of TDC. The fundamental reason is that even if the state space of the Markov chain $\{(S_t, A_t)\}$ is finite, with eligibility traces, we must instead consider the chain $\{(S_t, A_t, e_t)\}$, which now evolves in an uncount-

able space. Even worse, off-policy learning makes the trace $e_t$ unbounded as well, as the product of the importance sampling ratio can be unbounded. Prior works resort to additional projection (Yu, 2015; 2017) or truncation (Zhang & Whiteson, 2022) to handle the unboundedness of the eligibility traces. This work instead directly handles the unboundedness via averaging following the techniques in Kushner & Yin (2003); Liu et al. (2025b).

## 9. Conclusion

This work establishes the first almost sure convergence results for general two-timescale Markovian SA under mild assumptions. This work can be viewed as a combination of Lakshminarayanan & Bhatnagar (2017) and Liu et al. (2025b), both of which are extensions of the seminal work Borkar & Meyn (2000). Key to our analysis is a methodological innovation that bounds the faster iterates with the running max of the slower iterates. On the other hand, Borkar & Meyn (2000) are recently extended by Borkar et al. (2025) to include not only Markovian noise but also (functional) central limit theorems. A fruitful direction for future work is thus to investigate whether our methodological innovation can be applied to extend the (functional) central limit theorems of Borkar et al. (2025) to two-timescale SA.

## Acknowledgements

This work is supported in part by the US National Science Foundation under the awards III-2128019, SLES-2331904, and CAREER-2442098, the Commonwealth Cyber Initiative's Central Virginia Node under the award VV-1Q26-001, and a Cisco Faculty Research Award. Additionally, we thank the reviewers whose comments and suggestions improved the quality of our work.

## Impact Statement

This paper presents work whose goal is to advance the field of Machine Learning. There are many potential societal consequences of our work, none which we feel must be specifically highlighted here.

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

# A. Mathematical Background

**Theorem A.1** (**Gronwall Inequality**). *(Lemma 6 in Section 11.2 in Borkar (2009)) For a continuous function $u(\cdot) \geq 0$ and scalars $C, K, T \geq 0$,*

$$u(t) \leq C + K \int_0^t u(s)ds \quad \forall t \in [0, T]$$

*implies*

$$u(t) \leq Ce^{tK}, \forall t \in [0, T].$$

**Theorem A.2** (**Gronwall Inequality in the Reverse Time**). *For a continuous function $u(\cdot) \geq 0$ and scalars $C, K, T \geq 0$,*

$$u(t) \leq C + K \int_t^0 u(s)ds \quad \forall t \in [-T, 0]$$

*implies*

$$u(t) \leq Ce^{-tK}, \forall t \in [-T, 0].$$

For a proof, see Appendix A.2 of Liu et al. (2025b).

**Theorem A.3** (**Discrete Gronwall Inequality**). *(Lemma 8 in Section 11.2 in Borkar (2009)) For nonnegative sequences $\{x_n, n \geq 0\}$ and $\{a_n, n \geq 0\}$ and scalars $C, L \geq 0$,*

$$x_{n+1} \leq C + L \sum_{i=0}^n a_i x_i \quad \forall n$$

*implies*

$$x_{n+1} \leq Ce^{L \sum_{i=0}^n a_i} \quad \forall n.$$

**Theorem A.4** (The Arzelà-Ascoli Theorem in the Extended Sense on $[0, T)$). *Let $\{t \in [0, T) \mapsto g_n(t)\}$ be equicontinuous in the extended sense. Then, there exists a subsequence $\{g_{n_k}(t)\}$ that converges to some continuous limit $g^{\text{lim}}(t)$, uniformly in $t$ on $[0, T)$.*

For a proof, see Appendix A.4 of Liu et al. (2025b).

**Theorem A.5** (Moore-Osgood Theorem for Interchanging Limits). *If $\lim_{n \to \infty} a_{n,m} = b_m$ uniformly in $m$ and $\lim_{m \to \infty} a_{n,m} = c_n$ for each large $n$, then both $\lim_{m \to \infty} b_m$ and $\lim_{n \to \infty} c_n$ exists and are equal to the double limit, i.e.,*

$$\lim_{m \to \infty} \lim_{n \to \infty} a_{n,m} = \lim_{n \to \infty} \lim_{m \to \infty} a_{n,m} = \lim_{\substack{n \to \infty \\ m \to \infty}} a_{n,m}.$$

**Lemma A.6** (Sub-subsequence Lemma). *Let $x_n$ be a sequence in some metric space. If every subsequence $x_{n_k}$ itself has a subsequence $x_{n_{k_j}}$ that converges to the same limit $x$, then $x_n$ converges to $x$.*

*Proof.* Suppose we have a sequence $x_n$ where every subsequence $x_{n_k}$ itself has a subsequence $x_{n_{k_j}}$ that converges to the same limit $x$. For contradiction, assume that $x_n$ does not converge to $x$, so there is a subsequence $x_{n_{k,0}}$ that is always at least some distance $\epsilon$ away from $x$. By assumption, $x_{n_{k,0}}$ has a subsequence $x_{n_{k,1}}$ that converges to $x$, which contradicts that $x_{n_{k,0}}$ is always at least some $\epsilon$ away from $x$. So, it must be that $x_n$ converges to $x$. $\square$

# B. Main Assumptions

**Assumption B.1.** The Markov chain $\{W_n\}$ has a unique invariant probability measure (i.e., stationary distribution), denoted by $d_{\mathcal{W}}$.

Although the uniqueness and even the existence of the invariant probability measure can be relaxed, we use Assumption B.1 for simplification. Additionally, due to the way updates are defined ((1) and (2)), we start counting $\{W_n\}$ from $n = 1$.

**Assumption B.2.** The learning rates $\{\alpha(i)\}, \{\beta(i)\}$ are positive, decreasing, and satisfy

$$\sum_{i=0}^{\infty} \alpha(i) = \sum_{i=0}^{\infty} \beta(i) = \infty,$$

$$\lim_{i \to \infty} \alpha(i) = \lim_{i \to \infty} \beta(i) = 0$$

$$\frac{\alpha(i) - \alpha(i+1)}{\alpha(i)} = \mathcal{O}\left(\alpha(i)\right), \tag{7}$$

$$\frac{\beta(i) - \beta(i+1)}{\beta(i)} = \mathcal{O}\left(\beta(i)\right),$$

$$\lim_{i \to \infty} \frac{\beta(i)}{\alpha(i)} = 0. \tag{8}$$

These conditions on the learning rates are quite common in stochastic approximation. The last condition is necessary for the two-timescale formulation we are using.

*Remark* B.3. For any $\alpha(n) = \frac{B_1}{(n+B_2)^{\gamma_\alpha}}, \beta(n) = \frac{B_3}{(n+B_4)^{\gamma_\beta}}$ with $\gamma_\alpha, \gamma_\beta \in (0.5, 1], \gamma_\beta > \gamma_\alpha$, and all $B_i$ positive, one can verify that all the conditions in Assumption B.2 are satisfied.

Now, we make assumptions about the functions $H$ and $G$. For any $c \in [1, \infty)$, define

$$H_c(x, y, w) \doteq \frac{H(cx, cy, w)}{c} \tag{9}$$

$$G_c(x, y, w) \doteq \frac{G(cx, cy, w)}{c}. \tag{10}$$

The functions $H_c$ and $G_c$ are rescaled versions of the functions $H$ and $G$ and will be used to construct rescaled iterates, a key technique in the ODE method (see, e.g., Borkar & Meyn (2000); Borkar (2009); Liu et al. (2025b)). Just as in those works, we need the existence of some sort of limiting functions for $H_c$ and $G_c$ when $c \to \infty$.

**Assumption B.4.** There exist measurable functions $H_\infty(x, y, w)$ and $G_\infty(x, y, w)$, functions $\kappa_H(c), \kappa_G(c) : \mathbb{R} \to \mathbb{R}$, and measurable functions $b_H(x, y, w), b_G(x, y, w)$ such that for all $x, y, w$

$$H_c(x, y, w) - H_\infty(x, y, w) = \kappa_H(c) b_H(x, y, w) \tag{11}$$

$$G_c(x, y, w) - G_\infty(x, y, w) = \kappa_G(c) b_G(x, y, w)$$

$$\lim_{c \to \infty} \kappa_H(c) = \lim_{c \to \infty} \kappa_G(c) = 0$$

There also exists a measurable function $L_b(w)$ such that for all $x, x', y, y', w$

$$\|b_H(x, y, w) - b_H(x', y', w)\| \le L_b(w)\|(x, y) - (x', y')\|,$$

$$\|b_G(x, y, w) - b_G(x', y', w)\| \le L_b(w)\|(x, y) - (x', y')\|.$$

Additionally, the expectation $L_b \doteq \mathbb{E}_{w \sim d_{\mathcal{W}}}[L_b(w)]$ is well-defined and finite.

Assumption B.4 provides details on how $H_c$ and $G_c$ converge to $H_\infty$ and $G_\infty$ when $c \to \infty$.

We now assume that the functions $H_c$ and $G_c$ are Lipschitz. This will guarantee that the corresponding ODEs exist and are unique.

**Assumption B.5.** There exists a measurable function $L(w)$ such that for any $x, x', y, y', w$,

$$\|H(x, y, w) - H(x', y', w)\| \leq L(w)\|(x, y) - (x', y')\|, \tag{12}$$
$$\|H_\infty(x, y, w) - H_\infty(x', y', w)\| \leq L(w)\|(x, y) - (x', y')\|, \tag{13}$$
$$\|G(x, y, w) - G(x', y', w)\| \leq L(w)\|(x, y) - (x', y')\|,$$
$$\|G_\infty(x, y, w) - G_\infty(x', y', w)\| \leq L(w)\|(x, y) - (x', y')\|.$$

Moreover, the following expectations are well-defined and finite for any $x, y$:

$$h(x, y) \doteq \mathbb{E}_{w \sim d_\mathcal{W}}[H(x, y, w)],$$
$$h_\infty(x, y) \doteq \mathbb{E}_{w \sim d_\mathcal{W}}[H_\infty(x, y, w)],$$
$$g(x, y) \doteq \mathbb{E}_{w \sim d_\mathcal{W}}[G(x, y, w)],$$
$$g_\infty(x, y) \doteq \mathbb{E}_{w \sim d_\mathcal{W}}[G_\infty(x, y, w)],$$
$$L \doteq \mathbb{E}_{w \sim d_\mathcal{W}}[L(w)].$$

The functions $x, y \mapsto H_c(x, y, w)$ and $x, y \mapsto G_c(x, y, w)$ share the same Lipschitz constant $L(w)$ as the functions $x, y \mapsto H(x, y, w)$ and $x, y \mapsto G(x, y, w)$. Similarly to (9) and (10), we define

$$h_c(x, y) \doteq \frac{h(cx, cy)}{c},$$
$$g_c(x, y) \doteq \frac{g(cx, cy)}{c}.$$

The following assumption is necessary to ensure that the rescaled iterates can converge to the trajectory of the limiting ODE.

**Assumption B.6.** The ODE

$$\frac{dx(t)}{dt} = h(x(t), y)$$

has a unique globally asymptotically stable equilibrium $\lambda(y)$ where $\lambda : \mathbb{R}^{d_2} \to \mathbb{R}^{d_1}$ is a Lipschitz map. Additionally, we assume that the ODE

$$\frac{dy(t)}{dt} = g(\lambda(y(t)), y(t))$$

has a unique globally asymptotically stable equilibrium.

As $c \to \infty$, $h_c(x, y)$ converges to $h_\infty(x, y)$ uniformly in $(x, y)$ on any compact subsets of $\mathbb{R}^{d_1+d_2}$. The limiting ODE

$$\frac{dx(t)}{dt} = h_\infty(x(t), y)$$

has a unique globally asymptotically stable equilibrium $\lambda_\infty(y)$ where $\lambda_\infty : \mathbb{R}^{d_2} \to \mathbb{R}^{d_1}$ is a Lipschitz map. $\lambda_\infty$ is homogeneous, i.e., $\lambda_\infty(cy) = c\lambda_\infty(y)$. And $\lambda_\infty(0) = 0$.

As $c \to \infty$, $g_c(x, y)$ converges uniformly to $g_\infty(x, y)$ on compact subsets of $\mathbb{R}^{d_1+d_2}$. The limiting ODE

$$\frac{dy(t)}{dt} = g_\infty(\lambda_\infty(y(t)), y(t))$$

has 0 as its unique globally asymptotically stable equilibrium.

The map $\lambda_\infty$ tells us, for a particular external input $y$, what $x$ (the parameters of the inner loop) should converge to. This idea and the notation for it come from Chapter 6 of Borkar (2009).

**Assumption B.7.** Let $\gamma$ denote any of the following functions:

$$
\begin{aligned}
w &\mapsto H(x, y, w) \quad (\forall x, y), \\
w &\mapsto G(x, y, w) \quad (\forall x, y), \\
w &\mapsto L_b(w), \\
w &\mapsto L(w).
\end{aligned}
$$

We have, for any initial condition $W_1$,

$$
\lim_{n \to \infty} \alpha(n) \sum_{i=1}^{n} \left( \gamma(W_i) - \mathbb{E}_{w \sim d_{\mathcal{W}}}[\gamma(w)] \right) = 0 \quad \text{a.s.}
$$

$$
\lim_{n \to \infty} \beta(n) \sum_{i=1}^{n} \left( \gamma(W_i) - \mathbb{E}_{w \sim d_{\mathcal{W}}}[\gamma(w)] \right) = 0 \quad \text{a.s.} \tag{14}
$$

The assumption holds for all functions $\gamma$ on the same probability-one set.

*Remark* B.8. Once more, consider $\beta(n) = \frac{B_1}{(n+B_2)^{\gamma_\beta}}$ as an example. For $\gamma_\beta = 1$, (14) is implied by the following Law of Large Numbers (LLN)

$$
\lim_{n \to \infty} \frac{1}{n} \sum_{i=1}^{n} \left( \gamma(W_i) - \mathbb{E}_{w \sim d_{\mathcal{W}}}[\gamma(w)] \right) = 0 \quad \text{a.s.} \tag{LLN}
$$

For $\gamma_\beta \in (0.5, 1]$, (14) is implied by the following Law of the Iterated Logarithm (LIL)

$$
\left\| \sum_{i=1}^{n} \left( \gamma(W_n) - \mathbb{E}_{w \sim d_{\mathcal{W}}}[\gamma(w)] \right) \right\| \leq \zeta \sqrt{n \log \log n} \quad \text{a.s.,} \tag{LIL}
$$

where $\zeta$ is a sample path dependent finite constant.

# C. Proof of Lemma 3.1

This is a more detailed version of Section 4. In this section, we will prove Lemma 3.1. Section C.1 sets up the notation and explains our fast timescale analysis. Section C.2 defines the rescaled iterates and some important functions. Section C.3 assumes for contradiction that stability does not hold and identifies a resulting subsequence of interest. Section C.4 demonstrates convergence along this subsequence and uses this to show that the convergence holds for the entire sequence. Section C.5 defines some notation and provides some results about ODEs with external inputs. Finally, Section C.6 uses results from the previous sections to complete the proof. Lemmas in this section are derived on an arbitrary sample path $\{x_0, y_0, \{W_i\}_{i=1}^{\infty}\}$ such that the assumptions in Appendix B hold. Thus, we omit "$a.s.$" from the lemma statements for conciseness.

## C.1. Splitting up the Timeline

Here, we perform a fast timescale analysis. First, we split the positive real axis $[0, \infty)$ into chunks of length $\{\alpha(i)\}_{i=0,1,\ldots}$. We then collect these segments together into larger intervals $\{[T_n, T_{n+1})\}_{n=0,1,\ldots}$, where the sequence $\{T_n\}$ has the property that as $n$ grows large, we have $T_{n+1} - T_n \approx T$. We define

$$t(0) \doteq 0,$$
$$t(n) \doteq \sum_{i=0}^{n-1} \alpha(i) \quad n = 1, 2, \ldots \quad .$$

For all $T > 0$, define

$$m(T) = \max \{i | T \geq t(i)\} \tag{15}$$

to be the maximal $i$ where $t(i)$ is no greater than $T$. To visualize, $t(m(T))$ is a bit to the left of $T$ on the real axis. So, $t(m(T))$ satisfies the following:

$$t(m(T)) \leq T < t(m(T) + 1) = t(m(T)) + \alpha(m(T)),$$
$$t(m(T)) > T - \alpha(m(T)).$$

Define

$$T_0 = 0,$$
$$T_{n+1} = t(m(T_n + T) + 1).$$

Intuitively, $T_{n+1}$ is a little to the right of $T_n + T$ on the real axis.

We define

$$\alpha(i) = \beta(i) = 0 \quad \forall i < 0,$$
$$m(t) = 0 \quad \forall t \leq 0, \tag{16}$$

for simplifying notations. For any given function $f$ with domain $\mathcal{W}$, its asymptotic rate of change is defined as

$$\limsup_n \sup_{-\tau \leq t_1 \leq t_2 \leq \tau} \left\| \sum_{i=m(t(n)+t_1)}^{m(t(n)+t_2)-1} \alpha(i)[f(W_{i+1}) - \mathbb{E}_{w \sim d_{\mathcal{W}}}[f(w)]] \right\|.$$

This asymptotic rate of change helps us to describe the asymptotic regularity of $\{f(W_n)\}$ and we lean on its usefulness in studying stochastic approximation. We refer the reader to Sections 5.3.2 and 6.2 of Kushner & Yin (2003) for a detailed exposition of this tool. In this work, we have deferred the contents concerning asymptotic rate of change to Appendices G.1 and G.2 since the statements and proofs are very similar to results in Liu et al. (2025b).

### C.2. Defining the Scaled Iterates

We start by fixing a sample path $\{x_0, y_0, \{W_i\}_{i=1}^\infty\}$. We will take $\bar{z}(t)$ to be the piecewise constant interpolation[2] of $z_n$ at points $\{t(n)\}_{n=0,1,\ldots}$, i.e.,

$$\bar{z}(t) \doteq (\bar{x}(t), \bar{y}(t)) \doteq \begin{cases} (x_0, y_0) & t \in [0, t(1)) \\ (x_1, y_1) & t \in [t(1), t(2)) \\ (x_2, y_2) & t \in [t(2), t(3)) \\ \vdots \end{cases}$$

By (15), we also have

$$\bar{z}(t) \doteq z_{m(t)}. \tag{17}$$

By (1), $\forall n \geq 0$, we have

$$\bar{z}(t(n+1)) = \bar{z}(t(n)) + (\alpha(n)H(\bar{z}(t(n)), W_{n+1}), \beta(n)G(\bar{z}(t(n)), W_{n+1})).$$

Now we describe our rescaling of $\bar{z}(t)$.

**Definition C.1.** $\forall n \in \mathbb{N}, t \in [0, T+1)$, define

$$\tilde{z}_n(t) \doteq (\tilde{x}_n(t), \tilde{y}_n(t)) \doteq \frac{\bar{z}(T_n + t)}{r_n}$$

where

$$r_n \doteq \max\{1, r_{n-1}, \|\bar{z}(T_n)\|\}, r_0 \doteq \max\{1, \|\bar{z}(0)\|\}. \tag{18}$$

This implies

$$\forall n \in \mathbb{N}, \|\tilde{z}_n(0)\| \leq 1. \tag{19}$$

Moreover[3],

$\forall n \in \mathbb{N}, t \in [0, T+1)$,

$$\tilde{z}_n(t) = \frac{\bar{z}(T_n) + \sum_{i=m(T_n)}^{m(T_n+t)-1}(\alpha(i)H(\bar{z}(t(i)), W_{i+1}), \beta(i)G(\bar{z}(t(i)), W_{i+1}))}{r_n}$$

$$= \tilde{z}_n(0) + \sum_{i=m(T_n)}^{m(T_n+t)-1}(\alpha(i)H_{r_n}(\tilde{z}_n(t(i) - T_n), W_{i+1}), \beta(i)G_{r_n}(\tilde{z}_n(t(i) - T_n), W_{i+1})).$$

*Remark* C.2. Note that we depart from Liu et al. (2025b) in our definition of the rescaling factor $r_n$ (18). Through the definition, we ensure that the sequence $\{r_n\}$ is monotonic, which enables us to obtain convergence over the entire sequence rather than just a subsequence.

We also depart from prior works by defining the sequence of rescaled functions $\{t \mapsto \tilde{z}_n(t)\}$ which share the domain $[0, T+1)$, as opposed to rescaling the function $\bar{z}(t)$ directly (often denoted as $\hat{z}$). This consistent, larger domain greatly simplifies our arguments, while prior works must handle the diminishing excess part $[T, T_{n+1} - T_n)$, which can get messy.

---

[2]It also works if we consider a piecewise linear interpolation following Borkar (2009). The piecewise linear interpolation, however, will significantly complicate the presentation. We, therefore, follow Kushner & Yin (2003) and Liu et al. (2025b) and use the piecewise constant interpolation.

[3]In this paper, we use the convention that $\sum_{k=i}^j \alpha(k) = \sum_{k=i}^j \beta(k) = 0$ when $j < i$.

We can regard $\tilde{z}_n(t)$ as the Euler's discretization of $z_n(t)$ defined below.

**Definition C.3.** $\forall n \in \mathbb{N}, t \in [0, T+1)$, define $z_n(t) = (x_n(t), y_n(t))$ as the solution to the ODE system

$$\frac{dx_n(t)}{dt} = h_{r_n}(x_n(t), y_n(t))$$
$$\frac{dy_n(t)}{dt} = 0$$

with initial condition

$$z_n(0) = \tilde{z}_n(0). \tag{20}$$

We can also write $z_n(t)$ as

$$z_n(t) = \tilde{z}_n(0) + \int_0^t (h_{r_n}(z_n(s)), 0)ds. \tag{21}$$

Ideally, we would like to see that the error of the discretization diminishes asymptotically. Precisely speaking, the discretization error is defined as

$$f_n(t) \doteq \tilde{z}_n(t) - z_n(t)$$

and we want to show that $f_n(t)$ diminishes to 0 uniformly in $t$ as $n \to \infty$. To accomplish this, we need to analyze the following three sequences of functions

$$\{t \mapsto \tilde{z}_n(t)\}_{n=0}^\infty, \{z_n(t)\}_{n=0}^\infty, \{f_n(t)\}_{n=0}^\infty.$$

In particular, we show that they are all *equicontinuous in the extended sense*. We defer the relevant definitions, statements, and proofs to Appendix G.3 as they are quite similar to the analogous sections in Liu et al. (2025b).

### C.3. A Convergent Subsequence

The ultimate goal would be to show that

$$\sup_n \|z_n\| < \infty.$$

Observe the inequality

$$\forall n, \quad \left\|z_{m(T_n)}\right\| = \|\bar{z}(T_n)\| \le r_n.$$

Therefore, if we had

$$\sup_n r_n < \infty,$$

then the result would come easily. So, we assume for contradiction that $\sup_n r_n = \infty$. However, we can't obtain a contradiction in this section and must first show Lemma 3.1.

According to the Arzelà-Ascoli theorem in the extended sense (Theorem A.4), a sequence of equicontinuous functions always has a subsequence of functions that uniformly converge to a continuous limit. In the following, we use this to identify a particular subsequence of interest.

**Lemma C.4 (4.3 Restated).** *Suppose $\sup_n r_n = \infty$. Take an arbitrary subsequence $\{n_{k,0}\}_{k=0}^\infty \subseteq \{0, 1, 2, \ldots\}$. Then there is a subsequence $\{n_k\}_{k=0}^\infty \subseteq \{n_{k,0}\}_{k=0}^\infty$ such that there exist some continuous functions $f^{\lim}(t)$ and $\tilde{z}^{\lim}(t)$ such that $\forall t \in [0, T+1)$,*

$$\lim_{k \to \infty} f_{n_k}(t) = f^{\lim}(t),$$
$$\lim_{k \to \infty} \tilde{z}_{n_k}(t) = \tilde{z}^{\lim}(t), \tag{22}$$

*where both convergences are uniform in $t$ on $[0, T + 1)$. Furthermore, let $z^{\lim}(t) = (x^{\lim}(t), y^{\lim}(t))$ denote the unique solution to the ODE system*

$$\frac{dx^{\lim}(t)}{dt} = h_\infty(x^{\lim}(t), y^{\lim}(t))$$
$$\frac{dy^{\lim}(t)}{dt} = 0$$

*with the initial condition*

$$z^{\lim}(0) = \tilde{z}_n^{\lim}(0),$$

*in other words,*

$$z^{\lim}(t) = \tilde{z}^{\lim}(0) + \int_0^t (h_\infty(z^{\lim}(s)), 0)ds. \tag{23}$$

*Then $\forall t \in [0, T + 1)$, we have*

$$\lim_{k\to\infty} z_{n_k}(t) = z^{\lim}(t),$$

*where the convergence is uniform in $t$ on $[0, T + 1)$.*

Its proof is in Appendix G.4. We use the subsequence $\{n_k\}$ intensively in the remaining proofs.

### C.4. Diminishing Discretization Error

Recall that $f_n(t)$ denotes the discretization error between $\tilde{z}_n(t)$ and $z_n(t)$. In this section, we will show that $\lim_{n\to\infty} \|f_n(t)\| = 0$ for all $t \in [0, T + 1)$. We start by first proving that the discretization error diminishes along the sequence $\{n_k\}$, i.e., that

$$\lim_{k\to\infty} \|f_{n_k}(t)\| = \|f^{\lim}(t)\| = 0.$$

This means $\tilde{z}_{n_k}(t)$ is close to $z_{n_k}(t)$ as $k \to \infty$. For any $t \in [0, T + 1)$, we have

$$
\begin{aligned}
&\lim_{k\to\infty} \|f_{n_k}(t)\| \\
&= \lim_{k\to\infty} \left\| \sum_{i=m(T_{n_k})}^{m(T_{n_k}+t)-1} (\alpha(i)H_{r_{n_k}}(\tilde{z}_{n_k}(t(i) - T_{n_k}), W_{i+1}), \beta(i)G_{r_{n_k}}(\tilde{z}_{n_k}(t(i) - T_{n_k}), W_{i+1})) \right. \\
&\qquad\qquad \left. - \int_0^t (h_{r_{n_k}}(z_{n_k}(s)), 0)ds \right\| \qquad\qquad\qquad\qquad\qquad\qquad \text{(by (21))} \\
&\leq \lim_{k\to\infty} \left\| \sum_{i=m(T_{n_k})}^{m(T_{n_k}+t)-1} \alpha(i)H_{r_{n_k}}(\tilde{z}_{n_k}(t(i) - T_{n_k}), W_{i+1}) - \int_0^t h_{r_{n_k}}(\tilde{z}^{\lim}(s))ds \right\| \\
&\quad + \lim_{k\to\infty} \left\| \int_0^t h_{r_{n_k}}(\tilde{z}^{\lim}(s))ds - \int_0^t h_{r_{n_k}}(z_{n_k}(s))ds \right\| \tag{24} \\
&\quad + \lim_{k\to\infty} \left\| \sum_{i=m(T_{n_k})}^{m(T_{n_k}+t)-1} \beta(i)G_{r_{n_k}}(\tilde{z}_{n_k}(t(i) - T_{n_k}), W_{i+1}) \right\|.
\end{aligned}
$$

Here, we will bound the last term. This term is a novelty of the two-timescale setting and so there is no analogous result in Liu et al. (2025b).

**Lemma C.5.** $\forall t \in [0, T+1)$,

$$\lim_{k \to \infty} \left\| \sum_{i=m(T_{n_k})}^{m(T_{n_k}+t)-1} \beta(i) G_{r_{n_k}}(\tilde{z}_{n_k}(t(i) - T_{n_k}), W_{i+1}) \right\| = 0.$$

Its proof is in Appendix G.5. We defer the rest of the argument to Appendix G.6 since it is quite similar to the relevant section in Liu et al. (2025b). At the end of it all, from Lemma G.11, we obtain

$$\lim_{k \to \infty} \|f_{n_k}(t)\| = 0,$$

for all $t \in [0, T+1)$ showing that the discretization error goes to 0 along $\{n_k\}$.

Now, since we had chosen an arbitrary subsequence $\{f_{n_{k,0}}\}$ and it has a subsequence $\{f_{n_k}\}$ that converges to 0, by Lemma A.6 we know that $\{f_n\}$ also converges to 0. Thus, the discretization error diminishes along the entire sequence. That is,

$$\lim_{k \to \infty} \|f_n(t)\| = 0 \tag{25}$$

for all $t \in [0, T+1)$.

## C.5. ODEs with External Inputs

This section contains some notation and results concerning ODEs with external inputs, which we need for the next section. First, we will define some new notation.

**Definition C.6.** We use the notation $\eta_c^{y(t)}(t, x)$ to denote the solution to the ODE

$$\frac{dx(t)}{dt} = h_c(x(t), y(t))$$

with the initial condition $x$.

Note that under this notation, $x_n(t)$ (C.3) can be written $\eta_{r_n}^{\tilde{y}_n(0)}(t, \tilde{x}_n(0))$. This notation (borrowed from Lakshminarayanan & Bhatnagar (2017)) is useful since it identifies a trajectory of an ODE parameterized by the rescaling factor $c$ and the external input $y(t)$.

The following lemma shows that within a certain amount of time, the trajectory of the ODE will be pulled close to the equilibrium determined by the external input.

**Lemma C.7 (Lemma 4.5 Restated).** *Let $K \subset \mathbb{R}^{d_1}$ be compact and fix $y \in \mathbb{R}^{d_2}$. Given any $\epsilon > 0$, there exists a $T_\epsilon > 0$ such that for all initial conditions $x \in K$, we have $\eta_\infty^y(t, x) \in B(\lambda_\infty(y), \epsilon)$ for all $t \geq T_\epsilon$.*

Its proof is in Appendix G.7. The next lemma shows that if the external input is close to some constant $y$, then the trajectory will remain close to the trajectory we would obtain if $y$ was the external input.

**Lemma C.8 (Lemma 4.6 Restated).** *Let $y \in \mathbb{R}^{d_2}$, $[0, T]$ be a given time interval, and $\rho$ be a small positive constant with $y'(t) \in B(y, \rho)$. Then,*

$$\left\| \eta_c^{y'(t)}(t, x) - \eta_\infty^y(t, x) \right\| \leq (L\rho + \epsilon(c)) T e^{LT}.$$

*for any initial $x \in \mathbb{R}^{d_1}$ and for all $t \in [0, T]$.*

Its proof is in G.8. Armed with these results, we are ready for the next section.

## C.6. Completing the Proof

We now proceed to complete the proof of Lemma 3.1. The most involved result in this section is Lemma C.9, which tells us that if the $x$ component is too much larger than the $y$ component, the norm of the iterate at $T_{n+1}$ cannot be larger than that of the iterate at $T_n$. We use the results about ODEs with external inputs to ensure that the $x$ component of the trajectory gets pulled to the equilibrium, which is close to 0.

**Lemma C.9** (**Lemma 4.7** Restated). *There exists a constant $C_1 > 1$ such that if $\|\bar{x}(T_n)\| > C_1(1 + \|\bar{y}(T_n)\|)$, then $r_{n+1} = r_n$.*

*Proof.* Suppose that for some $n$, $\|\bar{x}(T_n)\| > C_1(1 + \|\bar{y}(T_n)\|)$, which implies that $\|\tilde{x}_n(0)\| > C_1(\frac{1}{r_n} + \|\tilde{y}_n(0)\|)$. Since $1 \geq \|\tilde{x}_n(0)\|$ and $\frac{\|\tilde{x}_n(0)\|}{C_1} > \|\tilde{y}_n(0)\|$, we have

$$\|\tilde{y}_n(0)\| < \frac{1}{C_1}. \tag{26}$$

We know from Lemma 4.5 that there exists some $T_{\frac{1}{4}}$ such that for all $t \geq T_{\frac{1}{4}}$,

$$\left\|\eta_\infty^0(t, \tilde{x}_n(0))\right\| \leq \frac{1}{4}. \tag{27}$$

So if we set $T \doteq T_{\frac{1}{4}}$, then (27) holds for all $t \in [T, T+1)$.

From Lemma 4.6 we know that there exist $\rho_{\frac{1}{4}}$ and $c_{\frac{1}{4}}$ such that if $y(t) \in B(0, \rho_{\frac{1}{4}})$ for all $t \in [0, T+1)$ and $r_n > c_{\frac{1}{4}}$ then

$$\left\|\eta_{r_n}^{y(t)}(t, \tilde{x}_n(0)) - \eta_\infty^0(t, \tilde{x}_n(0))\right\| \leq \frac{1}{4} \tag{28}$$

for all $t \in [0, T+1)$. Since $\lim_{n \to \infty} r_n = \infty$, there is some finite $n_1$ such that if $n > n_1$, we can be sure that $r_n > c_{\frac{1}{4}}$. By (25), we know that there exists $n_2$ such that for $n > n_2$, the discretization error will diminish enough so that

$$\tilde{y}_n(t) \in B\left(\tilde{y}_n(0), \min\left(\frac{\rho_{\frac{1}{4}}}{2}, \frac{1}{8}\right)\right) \tag{29}$$

for all $t \in [0, T+1)$. So, if we choose $C_1$ large enough so that $C_1 > 16$ and $\frac{1}{C_1} < \frac{\rho_{\frac{1}{4}}}{2}$, then we have $\tilde{y}_n(t) \in B(0, \rho_{\frac{1}{4}})$ (by (26) and (29)) for all $t \in [0, T+1)$. So, we know that for all $n > \max(n_1, n_2)$, (28) holds.

From (25), we know that there is some finite $n_3$ such that for $n > n_3$,

$$\left\|\tilde{x}_n(t) - \eta_{r_n}^{\tilde{y}_n(0)}(t, \tilde{x}_n(0))\right\| \leq \frac{1}{4} \tag{30}$$

for all $t \in [0, T+1)$.

By (27), (28), and (30), since $\tilde{x}_n(T_{n+1} - T_n) = \frac{\bar{x}(T_{n+1})}{r_n}$, we have $\left\|\frac{\bar{x}(T_{n+1})}{r_n}\right\| \leq \frac{3}{4}$.

For $n > n_1$, since $\tilde{y}_n(T_{n+1} - T_n) = \frac{\bar{y}(T_{n+1})}{r_n}$ we also have $\left\|\frac{\bar{y}(T_{n+1})}{r_n}\right\| \leq \frac{1}{4}$ (by (26), $\frac{1}{C_1} < \frac{1}{16}$, and (29)).

So we conclude $\left\|\frac{\bar{z}(T_{n+1})}{r_n}\right\| \leq 1$, telling us that $r_{n+1} = r_n$ as desired. If we let $n_0 = \max(n_1, n_2, n_3)$ and ensure that $C_1 > \max_{i \leq n_0} \|\bar{x}(T_i)\|$, then the result holds for all $n$. $\square$

Lemma C.10 tells us that the iterate at each $T_n$ is bounded by the largest $y$ component seen so far in the sequence of times $T_n$. To do this, we use an inductive argument–intuitively, we combine the previous lemma, which prevents $z$ from growing if the $x$ component is too much larger than the $y$ component, with Lemma H.15, which prevents $z$ from growing too much within one period $T$, ensuring that $x$ can never get too much larger than $y$.

**Lemma C.10** (**Lemma 4.8** Restated). *There exists a constant $C_2$ such that for all $n$,*

$$\|\bar{z}(T_n)\| \leq C_1 C_2 (\max_{m \leq n} \|\bar{y}(T_m)\| + 1).$$

*Proof.* From Lemma H.15, we know that there are some constants $A, B$ such that for all $n$,

$$\|\bar{z}(T_{n+1})\| \leq A\|\bar{z}(T_n)\| + B.$$

Our argument is inductive in nature. Let $C_1, C_2$ and ensure that $C_1 > A + B$ and $C_2 > 2A + B + 2$. To make sure the base case $(n = 0)$, holds, we also ensure that $C_1 C_2 > \|z_0\|$. From Lemma C.9 we know that if $\|\bar{x}(T_n)\| \geq C_1(1 + \|\bar{y}(T_n)\|)$, then $r_{n+1} = r_n$. Thus, if the result holds for all $i$ less than some $n$, i.e., we have $\|\bar{z}(T_i)\| \leq C_1 C_2(\max_{m \leq i} \|\bar{y}(T_m)\| + 1)$ for all $i \leq n$, then we can conclude that $\|\bar{z}(T_{n+1})\| \leq C_1 C_2(\max_{m \leq n} \|\bar{y}(T_m)\| + 1)$ also.

To address the other case, assume for some $n$ that $\|\bar{x}(T_n)\| \leq C_1(\|\bar{y}(T_n)\| + 1)$. Then we have

$$
\begin{aligned}
\|\bar{z}(T_{n+1})\| &\leq A\|\bar{z}(T_n)\| + B \\
&\leq A\|\bar{x}(T_n)\| + A\|\bar{y}(T_n)\| + B \\
&\leq AC_1\|\bar{y}(T_n)\| + AC_1 + A\|\bar{y}(T_n)\| + B \\
&\leq C_1 C_2(\|\bar{y}(T_n)\| + 1) \\
&\leq C_1 C_2(\max_{m \leq n+1} \|\bar{y}(T_m)\| + 1).
\end{aligned}
$$

$\square$

Finally, while we've ensured that for all the $T_n$ that $\bar{z}(T_n)$ is not to much larger than $\bar{y}(T_n)$, we need the result to also hold for all the iterates that lie in between $T_n$ and $T_{n+1}$. Lemma C.11 takes care of this.

**Lemma C.11 (Lemma 4.9 Restated).** *There exists a constant $C_3$ such that for all $n$,*

$$\|x_n\| \leq C_1 C_2 C_3(\|y_n^{max}\| + 1),$$

*Proof.* From Lemma H.15, we know that for all $m$ such that $m(T_n) \leq m \leq m(T_{n+1})$, there exist constants $A, B$ such that

$$\|z_m\| \leq A\|\bar{z}(T_n)\| + B.$$

Let $C_3 > A + B$. This means that

$$
\begin{aligned}
\|x_m\| \leq \|z_m\| &\leq AC_1 C_2(\max_{l \leq n} \|\bar{y}(T_l)\| + 1) + B \\
&\leq C_1 C_2 C_3(\max_{l \leq n} \|\bar{y}(T_l)\| + 1) \\
&\leq C_1 C_2 C_3(\|y_n^{max}\| + 1).
\end{aligned}
$$

$\square$

By setting $K$ equal to $C_1 C_2 C_3$, this concludes the proof of Lemma 3.1.

### C.7. Connecting the Timescales

Now that we have proved the first major result, Lemma 3.1, we now have some final tasks in the fast timescale to show that the fast iterates will track the slow iterates in some way. The result at the end of this section will be used in showing the diminishing discretization error in the slow timescale analysis.

In the following result, we show that there is a time period within which we can be such that the ODE trajectory of the fast variable will fall into and stay within a ball around $\lambda_\infty(y)$.

**Lemma C.12.** *Let $K_y \subset \mathbb{R}^{d_2}$ where $K_y$ is compact. Given any $\epsilon > 0$, there exists a $T_\epsilon > 0$ such that for all constant external inputs $y \in K_y$, we have $\eta_\infty^y(t, x) \in B(\lambda_\infty(y), \epsilon)$ for any initial condition $x \in \lambda_\infty(K_y)$ and for all $t \geq T_\epsilon$.*

*Proof.* By Lyapunov stability, we know that there is some $\delta$ with $\frac{\epsilon}{2} > \delta > 0$ such that if $\eta_\infty^y(t, x) \in B(\lambda_\infty(y), \delta)$, then for all $t' > t$, $\eta_\infty^y(t', x) \in B(\lambda_\infty(y), \frac{\epsilon}{2})$. By Lemma 4.5 we know that there exists some time $T_\delta$ such that for all $x \in \lambda_\infty(K_y)$ (image of a compact set under a continuous map is compact) and $t \geq T_\delta$, we have

$$\eta_\infty^y(t, x) \in B(\lambda_\infty(y), \frac{\delta}{2}). \tag{31}$$

By Lemma 4.6, we can select $\rho_y$ small enough such that for all $y_1 \in B(y, \rho_y)$,

$$\|\eta_\infty^y(t, x) - \eta_\infty^{y_1}(t, x)\| \le \frac{\delta}{2} \tag{32}$$

for all $t \in [0, T_\delta]$, $x \in \lambda_\infty(K_y)$.

We can split up the timeline into chunks of size $T_\delta$. The insight we will rely on is that if $\eta_\infty^y(T_\delta, x) = x_1$, then $\eta_\infty^y(T_\delta + t, x) = \eta_\infty^y(t, x_1)$ for all $t \ge 0$. The way our logic proceeds is as follows: By (31) and (32) we have that

$$\eta_\infty^{y_1}(T_\delta, x) \in B(\lambda_\infty(y), \delta)$$

Let $x_1 \doteq \eta_\infty^{y_1}(T_\delta, x)$. For all $t \in [0, T_\delta]$, we know by Lyapunov stability that

$$\eta_\infty^y(t, x_1) \in B(\lambda_\infty(y), \frac{\epsilon}{2}),$$

so by (32) we know (since we can reuse the $\rho_y$ selected earlier) that

$$\eta_\infty^{y_1}(t, x_1) \in B(\lambda_\infty(y), \frac{3\epsilon}{4}).$$

for all $t \in [0, T_\delta]$, implying that

$$\eta_\infty^{y_1}(t, x) \in B(\lambda_\infty(y), \frac{3\epsilon}{4}) \tag{33}$$

for all $t \in [T_\delta, 2T_\delta]$.

By (31) and (32) we know that

$$\eta_\infty^{y_1}(T_\delta, x_1) \in B(\lambda_\infty(y), \delta).$$

We can then define $x_2 \doteq \eta_\infty^{y_1}(T_\delta, x_1)$ and repeat the above arguments to see that (33) holds for all $t \in [2T_\delta, 3T_\delta]$. We can continue repeating this argument for all $x_n$ to see that (33) holds for all $t \ge T_\delta$.

Let $L_\lambda$ be the Lipschitz constant for $\lambda_\infty$. Then for all $y_1 \in B(y, \frac{\epsilon}{4L_\lambda})$ we have

$$\|\lambda_\infty(y) - \lambda_\infty(y_1)\| \le \frac{\epsilon}{4}. \tag{34}$$

To summarize, by (33) and (34), we know that if $y_1 \in B(y, \min(\rho_y, \frac{\epsilon}{4L_\lambda}))$ then

$$\eta_\infty^{y_1}(t, x) \in B(\lambda_\infty(y_1), \epsilon).$$

Each $y$ gives us such a ball $B(y, \min(\rho_y, \frac{\epsilon}{4L_\lambda}))$ along with a $T_\delta$ and these balls cover $K$. By compactness, we can obtain a finite subcover and a finite number of times $T_1, \dots, T_n$. Taking $T_\epsilon$ to be the maximum of these times completes the proof. $\square$

The next lemma shows that if the fast variable starts close enough to the slow variable, it should stay close to it forever.

**Lemma C.13.** *Let $K_y \subset \mathbb{R}^{d_2}$ where $K_y$ is compact. Given any $\epsilon > 0$, there exists a $\delta$ such that for all constant external inputs $y \in K_y$, if the initial condition $x \in B(\lambda_\infty(y), \delta)$, then $\eta_\infty^y(t, x) \in B(\lambda_\infty(y), \epsilon)$ for all $t \ge 0$.*

*Proof.* Fix $y$ and $\epsilon$. From Lemma C.12 we know that the result holds for $t \ge T_\epsilon$, so we just need to show that it holds for $0 \le t < T_\epsilon$. By Lyapunov stability, there exists $\delta_y$ such that if $x \in B(\lambda_\infty(y), \delta_y)$, then for all $t \ge 0$, $\eta_\infty^y(t, x) \in B(\lambda_\infty(y), \frac{\epsilon}{3})$.

By Lemma 4.6, there exists $\rho$ small enough that for all $y' \in B(y, \rho)$,

$$\|\eta_\infty^y(t, x) - \eta_\infty^{y_1}(t, x)\| < \frac{\epsilon}{3}$$

for all $x \in B(\lambda_\infty(y), \delta_y)$, $t \in [0, T_\delta]$. Finally, since $\lambda_\infty$ is Lipschitz, if $y' \in B(y, \frac{\epsilon}{3L_\lambda})$,

$$\|\lambda_\infty(y) - \lambda_\infty(y_1)\| < \frac{\epsilon}{3}.$$

Combining these facts, we know that if $y_1 \in B(y, \min(\rho, \frac{\epsilon}{3L_\lambda}))$, then $\eta_\infty^y(t, x) \in B(\lambda_\infty(y), \epsilon)$ for all $t \geq 0$. Each $y$ comes with such a neighborhood and a distance $\delta$. By compactness, we can extract a finite subcover and a finite number of distances $\delta_1, \ldots, \delta_n$, and taking $\delta$ to be the smallest one gives us our result. $\qquad\square$

Finally, we repeatedly apply the previous two lemmas every period to show that the fast iterates track the slow iterates.

**Lemma C.14 (Lemma 4.10 Restated).** *For any $\epsilon > 0$, there is some $T_\epsilon$ and some $N_\epsilon$ such that for all $n > N_\epsilon$, $\|\tilde{x}_n(t) - \lambda_\infty(\tilde{y}_n(t))\| \leq \epsilon$ for all $t \in [0, T_\epsilon]$.*

*Proof.* By C.13, we know that for all $y \in [-2, 2]^{d_2}$ (chosen since $\|\tilde{y}_n(0)\| \leq 1$) there is some $\delta$ with $\frac{\epsilon}{2} > \delta > 0$ such that if $\eta_\infty^y(t, x) \in B(\lambda_\infty(y), \delta)$, then for all $t' > t$, $\eta_\infty^y(t', x) \in B(\lambda_\infty(y), \frac{\epsilon}{2})$. By Lemma C.12 we know that there exists some time $T_\delta$ such that for all $x \in \lambda_\infty(K_y)$ (image of a compact set under a continuous map is compact) and $t \geq T_\delta$, we have, for all $y \in [-2, 2]^{d_2}$,

$$\eta_\infty^y(t, x) \in B(\lambda_\infty(y), \frac{\delta}{4}). \tag{35}$$

By Lemma 4.6, there exists $n_1$ such that for all $n > n_1$,

$$\left\| \eta_{r_n}^y(t, x) - \eta_\infty^y(t, x) \right\| < \frac{\delta}{4} \tag{36}$$

for all $t \in [0, T_\delta + 1]$.

By (25), there exists $n_2$ such that for all $n > n_2$,

$$\left\| \tilde{x}_n(t) - \eta_{r_n}^{\tilde{y}_n(0)}(t, \tilde{x}_n(0)) \right\| < \frac{\delta}{4} \tag{37}$$

and

$$\|\lambda_\infty(\tilde{y}_n(t)) - \lambda_\infty(\tilde{y}_n(0))\| \leq L_\lambda \|\tilde{y}_n(t) - \tilde{y}_n(0)\| < \frac{\delta}{4} \tag{38}$$

for all $t \in [0, T_\delta + 1]$. Combining (35), (36), (37), (38), and the fact that for $n$ large enough, $T_{n+1} - T_n < T + 1$, we have

$$\|\tilde{x}_n(T_{n+1} - T_n) - \lambda_\infty(\tilde{y}_n(T_{n+1} - T_n))\| < \delta$$

for all $n > \max(n_1, n_2)$. This means that, by homogeneity of $\lambda_\infty$,

$$\|\tilde{x}_{n+1}(0) - \lambda_\infty(\tilde{y}_{n+1}(0))\| = \frac{r_n}{r_{n+1}} \|\tilde{x}_n(T_{n+1} - T_n) - \lambda_\infty(\tilde{y}_n(T_{n+1} - T_n))\| < \delta$$

when $n > \max(n_1, n_2)$, giving us

$$\|\tilde{x}_n(0) - \lambda_\infty(\tilde{y}_n(0))\| < \delta$$

for all $n > N_\epsilon = \max(n_1, n_2) + 1$. So,

$$\left\| \eta_{r_n}^{\tilde{y}_n(0)}(t, \tilde{x}_n(0)) - \lambda_\infty(\tilde{y}_n(0)) \right\| < \frac{\epsilon}{2}$$

for all $t \geq 0$. Combining this with (37) and (38) gives us

$$\|\tilde{x}_n(t) - \lambda_\infty(\tilde{y}_n(t))\| \leq \epsilon$$

for all $t \in [0, T_\delta]$, as desired. Taking $T_\epsilon \doteq T_\delta$ completes the proof.

$\qquad\square$

# D. Proof of Theorem 3.2

In this section we will prove Theorem 3.2.

## D.1. Slow Timescale Setup

Here we are working in the slow timescale. We reuse the notation from Section C.1 but redefine some things in terms of the slow timescale. We split the positive real axis $[0, \infty)$ into chunks of length $\{\beta(i)\}_{i=0,1,\dots}$. We then collect these segments together into larger intervals $\{[T_n, T_{n+1})\}_{n=0,1,\dots}$, where the sequence $\{T_n\}$ has the property that as $n$ grows large, we have $T_{n+1} - T_n \approx T$. We define

$$
t(0) \doteq 0,
$$
$$
t(n) \doteq \sum_{i=0}^{n-1} \beta(i) \quad n = 1, 2, \dots \quad .
$$

For all $T > 0$, define

$$
m(T) = \max\{i | T \geq t(i)\}
$$

to be the maximal $i$ where $t(i)$ is no greater than $T$.

Define

$$
T_0 = 0,
$$
$$
T_{n+1} = t(m(T_n + T) + 1).
$$

For any given function $f$ with domain $\mathcal{W}$, its asymptotic rate of change is defined as

$$
\limsup_{n} \sup_{-\tau \leq t_1 \leq t_2 \leq \tau} \left\| \sum_{i=m(t(n)+t_1)}^{m(t(n)+t_2)-1} \beta(i)[f(W_{i+1}) - \mathbb{E}_{w \sim d_{\mathcal{W}}}[f(w)]] \right\|.
$$

## D.2. Defining the Slow Iterates

Here, we take $\bar{z}(t)$ to be the piecewise constant interpolation of $z_n$ at points $\{t(n)\}_{n=0,1,\dots}$, i.e.,

$$
\bar{z}(t) \doteq (\bar{x}(t), \bar{y}(t)) \doteq \begin{cases} (x_0, y_0) & t \in [0, t(1)) \\ (x_1, y_1) & t \in [t(1), t(2)) \\ (x_2, y_2) & t \in [t(2), t(3)) \\ \vdots \end{cases}
$$

Now we describe our rescaling of $\bar{z}(t)$.

**Definition D.1.** $\forall n \in \mathbb{N}, t \in [0, T+1)$, define

$$
\tilde{z}_n(t) \doteq (\tilde{x}_n(t), \tilde{y}_n(t)) \doteq \frac{\bar{z}(T_n + t)}{r_n}.
$$

We will define $r_n$ differently, such that it will be at least as large as the largest iterate seen so far:

$$
r_n \doteq \max\left\{1, \left\| z_{m(T_n)}^{max} \right\| \right\}.
$$

where $z_n^{max} \doteq z_m$ such that $m = \arg\max_{i \leq n} \|z_i\|$ (with ties broken arbitrarily).

We can regard $\tilde{y}_n(t)$ as the Euler's discretization of $y_n(t)$ defined below.

**Definition D.2.** $\forall n \in \mathbb{N}, t \in [0, T+1)$, define $y_n(t)$ as the solution to the ODE

$$\frac{dy_n(t)}{dt} = g_{r_n}(\lambda_\infty(y_n(t)), y_n(t))$$

with initial condition

$$y_n(0) = \tilde{y}_n(0).$$

We can also write $y_n(t)$ as

$$y_n(t) = \tilde{y}_n(0) + \int_0^t g_{r_n}(\lambda_\infty(y_n(t)), y_n(t)) ds. \tag{39}$$

The discretization error is defined as

$$f_n(t) \doteq \tilde{y}_n(t) - y_n(t)$$

and we want to show that $f_n(t)$ diminishes to 0 uniformly in $t$ as $n \to \infty$. To accomplish this, we need to show that the following three sequences of functions are equicontinuous in the extended sense:

$$\{t \mapsto \tilde{y}_n(t)\}_{n=0}^\infty, \{y_n(t)\}_{n=0}^\infty, \{f_n(t)\}_{n=0}^\infty.$$

We deferred the relevant definitions to Appendix G.3 as they are quite similar to the analogous sections in Liu et al. (2025b), but we will prove it for one of the sequences of functions.

**Lemma D.3.** $\{\tilde{y}_n(t)\}_{n=0}^\infty$ *is equicontinuous in the extended sense on* $[0, T+1)$.

Its proof is in Appendix G.9. Equicontinuity for the other two sequences of functions follows more similarly to the fast timescale arguments.

### D.3. A Convergent Subsequence

The ultimate goal is to show that

$$\sup_n \|z_n\| < \infty.$$

Once again, observe the inequality

$$\forall n, \quad \|z_{m(T_n)}\| = \|\tilde{z}(T_n)\| \le r_n.$$

Therefore, if we had

$$\sup_n r_n < \infty,$$

then the result would come easily. So, we assume for contradiction from now on that $\sup_n r_n = \infty$.

According to the Arzelà-Ascoli theorem in the extended sense (Theorem A.4), a sequence of equicontinuous functions always has a subsequence of functions that uniformly converge to a continuous limit. In the following, we use this to identify a particular subsequence of interest.

**Lemma D.4.** *Suppose* $\sup_n r_n = \infty$. *Take an arbitrary subsequence* $\{n_{k,0}\}_{k=0}^\infty \subseteq \{0, 1, 2, \dots\}$. *Then there is a subsequence* $\{n_k\}_{k=0}^\infty \subseteq \{n_{k,0}\}_{k=0}^\infty$ *such that there exist some continuous functions* $f^{\text{lim}}(t)$ *and* $\tilde{y}^{\text{lim}}(t)$ *such that* $\forall t \in [0, T+1)$,

$$\lim_{k \to \infty} f_{n_k}(t) = f^{\text{lim}}(t),$$
$$\lim_{k \to \infty} \tilde{y}_{n_k}(t) = \tilde{y}^{\text{lim}}(t), \tag{40}$$

*where both convergences are uniform in t on $[0, T + 1)$. Furthermore, let $y^{\mathrm{lim}}(t)$ denote the unique solution to the ODE*

$$\frac{dy^{\mathrm{lim}}(t)}{dt} = g_\infty(\lambda_\infty(y^{\mathrm{lim}}(t)), y^{\mathrm{lim}}(t))$$

*with the initial condition*

$$y^{\mathrm{lim}}(0) = \tilde{y}_n^{\mathrm{lim}}(0),$$

*in other words,*

$$y^{\mathrm{lim}}(t) = \tilde{y}^{\mathrm{lim}}(0) + \int_0^t g_\infty(\lambda_\infty(y^{\mathrm{lim}}(s)), y^{\mathrm{lim}}(s))ds.$$

*Then $\forall t \in [0, T + 1)$, we have*

$$\lim_{k \to \infty} y_{n_k}(t) = y^{\mathrm{lim}}(t),$$

*where the convergence is uniform in t on $[0, T + 1)$.*

Its proof is in Appendix G.10. We use the subsequence $\{n_k\}$ intensively in the remaining proofs.

### D.4. Diminishing Discretization Error

Recall that $f_n(t)$ denotes the discretization error between $\tilde{y}_n(t)$ and $y_n(t)$. In this section, we will show that $\lim_{n \to \infty} \|f_n(t)\| = 0$ for all $t \in [0, T + 1)$. We start by first proving that the discretization error diminishes along the sequence $\{n_k\}$, i.e., that

$$\lim_{k \to \infty} \|f_{n_k}(t)\| = \|f^{\mathrm{lim}}(t)\| = 0.$$

This means $\tilde{y}_{n_k}(t)$ is close to $y_{n_k}(t)$ as $k \to \infty$. For any $t \in [0, T + 1)$, we have

$$\lim_{k \to \infty} \|f_{n_k}(t)\|$$

$$= \lim_{k \to \infty} \left\| \sum_{i=m(T_{n_k})}^{m(T_{n_k}+t)-1} \beta(i)G_{r_{n_k}}(\tilde{z}_{n_k}(t(i) - T_{n_k}), W_{i+1}) - \int_0^t g_{r_{n_k}}(\lambda_\infty(y_{n_k}(s)), y_{n_k}(s))ds \right\| \quad \text{(by (39))}$$

$$\leq \lim_{k \to \infty} \left\| \sum_{i=m(T_{n_k})}^{m(T_{n_k}+t)-1} \beta(i)[G_{r_{n_k}}(\tilde{z}_{n_k}(t(i) - T_{n_k}), W_{i+1}) - G_{r_{n_k}}(\lambda_\infty(\tilde{y}_{n_k}(t(i) - T_{n_k})), \tilde{y}_{n_k}(t(i) - T_{n_k}), W_{i+1})] \right\| (41)$$

$$+ \left\| \sum_{i=m(T_{n_k})}^{m(T_{n_k}+t)-1} \beta(i)G_{r_{n_k}}(\lambda_\infty(\tilde{y}_{n_k}(t(i) - T_{n_k})), \tilde{y}_{n_k}(t(i) - T_{n_k}), W_{i+1}) - \int_0^t g_{r_{n_k}}(\lambda_\infty(y_{n_k}(s)), y_{n_k}(s))ds \right\|.$$

**Lemma D.5 (Lemma 5.3 Restated).** *For all $\epsilon > 0$, there exists some $N$ such that for all $n > N$,*

$$\|\tilde{x}_n(t) - \lambda_\infty(\tilde{y}_n(t))\| < \epsilon$$

*for all $t \in [0, T]$.*

*Proof.* For clarity, since we will work with both fast and slow timescale objects, we will use $\alpha$ and $\beta$ in superscripts to indicate the fast and slow timescale variants respectively. We also define some notation to help in converting between timescales:

$$\tau^\alpha(t) \doteq t^\alpha(m^\beta(t))$$
$$\tau^\beta(t) \doteq t^\beta(m^\alpha(t))$$

By the difference in how the rescaling is defined between the timescales, we know that $\left\|\tilde{z}_n^\beta(0)\right\| \leq \left\|\tilde{z}_m^\alpha(\tau^\alpha(T_n^\beta) - T_m^\alpha)\right\|$, where $m$ is the largest $v$ such that $T_v^\alpha \leq \tau^\alpha(T_n^\beta)$. Essentially, whenever the slow timescale is rescaled, the iterate has a smaller norm than the fast timescale's iterate at that index. Now we need to show that for the rest of that period in the slow timescale (i.e. when $T_n^\beta \leq t \leq T_{n+1}^\beta$) the slow timescale iterates, despite not being rescaled during this period, never get too much larger than the fast timescale iterates which may be rescaled many times.

Thus, given $t \in [0, T^\beta]$, we are interested in the ratio $\frac{r_l^\alpha}{r_n^\beta}$, where $l$ is the largest $v$ where $T_v^\alpha \leq \tau^\alpha(T_n^\beta + t)$. By Lemma H.19 we know that

$$r_l^\alpha \leq \left\|z_{m^\beta(T_n^\beta+t)}^{max}\right\| \leq C\left\|z_{m^\beta(T_n^\beta)}^{max}\right\| + D = C \cdot r_n^\beta + D$$

implying that

$$\frac{r_l^\alpha}{r_n^\beta} \leq C + \frac{D}{r_n^\beta}.$$

There exists $n_1$ such that for all $n > n_1$, $\frac{D}{r_n^\beta} \leq C$, so for all $n > n_1$,

$$\frac{r_l^\alpha}{r_n^\beta} \leq 2C.$$

By Lemma C.14, there exist $n_2$ and $T_\epsilon$ such that for all $n > n_\epsilon$, we have

$$\|\tilde{x}_n^\alpha(t) - \lambda_\infty(\tilde{y}_n^\alpha(t))\| \leq \frac{\epsilon}{2C}$$

for all $t \in [0, T_\epsilon]$. Now, set $N = \max\{n_1, n_2\}$, where $n_2$ is the smallest $v$ such that $T_v^\beta \geq \tau^\beta(T_{n_\epsilon}^\alpha)$ and $T^\alpha$ is set to $T_\epsilon$. We then have, for all $t \in [0, T^\beta]$, by homogeneity of $\lambda_\infty$,

$$\left\|\tilde{x}_n^\beta(t) - \lambda_\infty(\tilde{y}_n^\beta(t))\right\| \leq 2C\left\|\tilde{x}_l^\alpha(\tau^\alpha(T_n^\beta + t)) - \lambda_\infty(\tilde{y}_l^\alpha(\tau^\alpha(T_n^\beta + t)))\right\| \leq \epsilon$$

for all $n > N$. Since we set $T^\alpha$ to $T_\epsilon$ without modifying $T^\beta$, the result holds for all $\epsilon$, regardless of what value we set $T^\beta$ to at the start. $\qquad\square$

**Lemma D.6.** *The first error term* (41) *converges to 0.*

*Proof.*

$$\lim_{k\to\infty} \left\|\sum_{i=m(T_{n_k})}^{m(T_{n_k}+t)-1} \beta(i)[G_{r_{n_k}}(\tilde{z}_{n_k}(t(i) - T_{n_k}), W_{i+1}) - G_{r_{n_k}}(\lambda_\infty(\tilde{y}_{n_k}(t(i) - T_{n_k})), \tilde{y}_{n_k}(t(i) - T_{n_k}), W_{i+1})]\right\|$$

$$\leq \lim_{k\to\infty} \sum_{i=m(T_{n_k})}^{m(T_{n_k}+t)-1} \beta(i)L(W_i)\|\tilde{x}_{n_k}(t(i) - T_{n_k}) - \lambda_\infty(\tilde{y}_{n_k}(t(i) - T_{n_k}))\|$$

$$\leq \lim_{k\to\infty} \epsilon(k) \sum_{i=m(T_{n_k})}^{m(T_{n_k}+t)-1} \beta(i)L(W_i) \tag{Lemma D.5}$$

$$\leq C \lim_{k\to\infty} \epsilon(k) \tag{95}$$

$$= 0.$$

$\qquad\square$

The rest of the proof, including the verification that the other term diminishes is the same as Liu et al. (2025b) and the fast timescale's analogous term.

Now, since we had chosen an arbitrary subsequence $\{f_{n_{k,0}}\}$ and it has a subsequence $\{f_{n_k}\}$ that converges to 0, by Lemma A.6 we know that $\{f_n\}$ also converges to 0. Thus, the discretization error diminishes along the entire sequence. That is,

$$\lim_{k\to\infty} \|f_n(t)\| = 0 \tag{42}$$

for all $t \in [0, T + 1)$.

## D.5. Completing the Proof

Now we finish the proof by first establishing two results about ODEs and then finding the contradiction.

**Definition D.7.** We use the notation $\chi_c(t, y)$ to denote the solution to the ODE

$$\frac{\mathrm{d}y(t)}{\mathrm{d}t} = g_c(\lambda_\infty(y(t)), y(t))$$

with the initial condition $y$.

Note that under this notation, $y_n(t)$ (D.2) can be written $\chi_{r_n}(t, \tilde{y}_n(0))$.

The next result shows us that there is a period within which the trajectory of the slow variable will fall to 0.

**Lemma D.8.** *Let $K \subset \mathbb{R}^{d_2}$ be compact. Given any $\epsilon > 0$, there exists a $T_\epsilon$ such that for all initial conditions $y \in K$, $\chi_\infty(t, y) \in B(0, \epsilon)$ for all $t > T_\epsilon$.*

*Proof.* By Lyapunov stability, there is a $\delta > 0$ such that any trajectory beginning within $B(0, \delta)$ stays within $\frac{\epsilon}{2}$ of the equilibrium 0.

For an initial condition $y$, let $T_y$ be a time at which the trajectory is within $\frac{\delta}{2}$ of the equilibrium. Let $y_1$ be some other initial condition. By definition, Lipschitzness, and the Gronwall inequality, we have

$$\|\chi_\infty(t, y) - \chi_\infty(t, y_1)\| \leq \|y - y_1\| + L(L_\lambda + 1) \int_0^t \|\chi_\infty(s, y) - \chi_\infty(s, y_1)\| ds$$

$$\leq \|y - y_1\| e^{L(L_\lambda+1)T_y}$$

for all $t \leq T_y$.

So there is a neighborhood $V_y$ such that for all $y_1 \in V_y$, $\chi_\infty(T_y, y_1)$ is within $\delta$ of the equilibrium, which by Lyapunov stability implies that it will always be within $\epsilon$ of the equilibrium after $T_y$.

By compactness, we can cover the set $K$ by a finite number of such intervals and obtain a finite number of times $T_{y_1}, \ldots, T_{y_n}$ and then take the maximum time to be the value of $T_\epsilon$. $\qquad\square$

The next lemma shows that as $c$ approaches infinity, the two trajectories remain close within a certain time interval.

**Lemma D.9.** *Let $[0, T]$ be a given time interval. Then,*

$$\|\chi_c(t, y) - \chi_\infty(t, y)\| \leq \epsilon(c) \cdot T e^{L(L_\lambda+1)T}.$$

*for any initial $y \in \mathbb{R}^{d_2}$ and for all $t \in [0, T]$.*

*Proof.* We have

$$\chi_c(t, y) = y + \int_0^t g_c(\lambda_\infty(\chi_c(s, y)), \chi_c(s, y)) ds,$$

$$\chi_\infty(t, y) = y + \int_0^t g_\infty(\lambda_\infty(\chi_\infty(s, y)), \chi_\infty(s, y)) ds.$$

Let us define the error term:

$$e(t) = \|\chi_c(t, y) - \chi_\infty(t, y)\|.$$

We can bound $e(t)$ by two terms:

$$e(t) \leq \int_0^t \|g_c(\lambda_\infty(\chi_c(s, y)), \chi_c(s, y)) - g_c(\lambda_\infty(\chi_\infty(s, y)), \chi_\infty(s, y))\| ds$$

$$+ \int_0^t \|g_c(\lambda_\infty(\chi_\infty(s, y)), \chi_\infty(s, y)) - g_\infty(\lambda_\infty(\chi_\infty(s, y)), \chi_\infty(s, y))\| ds$$

To bound the first term, we use Lipschitzness:

$$\int_0^t \|g_c(\lambda_\infty(\chi_c(s,y)), \chi_c(s,y)) - g_c(\lambda_\infty(\chi_\infty(s,y)), \chi_\infty(s,y))\| ds$$

$$\leq L(L_\lambda + 1) \int_0^t \|\chi_c(t,y) - \chi_\infty(t,y)\| ds$$

$$= L \int_0^t e(s) ds.$$

To bound the second term, we use Lipschitzness and Assumption B.4:

$$\int_0^t \|g_c(\lambda_\infty(\chi_\infty(s,y)), \chi_\infty(s,y)) - g_\infty(\lambda_\infty(\chi_\infty(s,y)), \chi_\infty(s,y))\| ds$$

$$\leq \int_0^t \epsilon(c) ds$$

$$\leq T\epsilon(c).$$

To conclude, we will use the Gronwall Inequality (Appendix A.1):

$$e(t) \leq T\epsilon(c) + L(L_\lambda + 1) \int_0^t e(s) ds$$

$$\leq \epsilon(c) \cdot Te^{L(L_\lambda+1)T}.$$

$\square$

Since in this section, $r_n$ is defined as the largest iterate norm seen so far, if it is bounded, then the iterate norms are bounded, verifying Theorem 3.2.

**Lemma D.10.** *The sequence $r_n$ is bounded, creating a contradiction.*

*Proof.* By Lyapunov stability there is some $\delta$ with $\frac{1}{4(L_\lambda+1)} > \delta > 0$ such that if $\chi_\infty(t,y) \in B(0,\delta)$ at some time $t$, then for all times $t' > t$, we have

$$\chi_\infty(t',y) \in B\left(0, \frac{1}{4(L_\lambda + 1)}\right). \tag{43}$$

By Lemma D.8 we know that there is a $T_\delta$ such that when we set $T = T_\delta$, for all $t > T$,

$$\chi_\infty(t,y) \in B(0, \frac{\delta}{3}) \tag{44}$$

for all $y \in [-2, 2]^{d_2}$.

By Lemma D.9 we know that there is an $n_1$ such that for all $n > n_1$,

$$\|\chi_{r_n}(t,y) - \chi_\infty(t,y)\| < \frac{\delta}{3} \tag{45}$$

for all $y \in [-2, 2]^{d_2}$ and $t \in [0, T+1]$.

By (42), we know that there is an $n_2$ such that for all $n > n_2$, we have

$$\|\tilde{z}_n(t) - (\lambda_\infty(\chi_{r_n}(t, \tilde{y}_n(0))), \chi_{r_n}(t, \tilde{y}_n(0)))\| < \frac{\delta}{3} \tag{46}$$

for all $t \in [0, T+1]$.

From (44), (45), and (46), and the fact that when $n$ is large enough, $T_{n+1} - T_n < T+1$, we have

$$\tilde{y}_n(0) \in B(0, \delta)$$

for all $n \geq \max\{n_1, n_2\} + 1$. Thus, from (43), we have

$$\chi_\infty(t, \tilde{y}_n(0)) \in B\left(0, \frac{1}{4(L_\lambda + 1)}\right)$$

for all $t \in [0, T+1]$, which implies that

$$(\lambda_\infty(\chi_\infty(t, \tilde{y}_n(0))), \chi_\infty(t, \tilde{y}_n(0))) \in B\left(0, \frac{1}{4}\right).$$

Combining this with (46) gives us

$$\tilde{z}_n(t) \in B(0, \frac{1}{2}),$$

for all $n \geq \max\{n_1, n_2\} + 1$ and $t \in [0, T+1]$, which, by definition, prevents the sequence $r_n$ from increasing, contradicting the assumption that $r_n$ increases to infinity. □

# E. Convergence

## E.1. Fast Timescale Convergence

In this section, we return to the fast timescale definitions.

**Corollary E.1.** *Let Assumptions B.1 - B.7 hold. Then the iterates $\{z_n\}$ generated by (1) converge almost surely to a (sample path dependent) bounded invariant set [4] of the ODE system [5]*

$$\frac{\mathrm{d}x(t)}{\mathrm{d}t} = h(x(t), y(t)) \tag{47}$$

$$\frac{\mathrm{d}y(t)}{\mathrm{d}t} = 0.$$

*Proof.* To prove convergence results on $t \in (-\infty, \infty)$ in Corollary E.1, we fix an arbitrary sample path $\{z_0, \{W_i\}_{i=1}^{\infty}\}$. The stability results from Theorem 3.2 hold. To prove properties on $t \in (-\infty, \infty)$, we first fix an arbitrary $\tau > 0$ and show properties on $\forall t \in [-\tau, \tau]$.

**Definition E.2.** $\forall n \in \mathbb{N}$, define $\bar{z}_n(t)$ as the solution to the ODE (47) in $(-\infty, \infty)$ with an initial condition

$$\bar{z}_n(0) = \bar{z}(t(n)).$$

$\bar{z}_n(t)$ can also be written as

$$\bar{z}_n(t) = \bar{z}(t(n)) + \int_0^t (h(\bar{z}_n(s)), 0)ds, \quad \forall t \in (-\infty, \infty). \tag{48}$$

We need to show that the error of Euler's discretization diminishes asymptotically. With (16) and (17), $\forall \tau > 0, \forall t \in [-\tau, \tau]$,

$$\bar{z}(t(n) + t) = z_{m(t(n)+t)} \tag{49}$$

$$= \begin{cases} \bar{z}(t(n)) + \sum_{i=n}^{m(t(n)+t)-1}(\alpha(i)H(\bar{z}(t(i)), W_{i+1}), \beta(i)G(\bar{z}(t(i)), W_{i+1})) & \text{if } t \geq 0 \\ \bar{z}(t(n)) - \sum_{i=m(t(n)+t)}^{n-1}(\alpha(i)H(\bar{z}(t(i)), W_{i+1}), \beta(i)G(\bar{z}(t(i)), W_{i+1})) & \text{if } t < 0. \end{cases}$$

Notably, the property (16) that $\forall t < 0, m(t) = 0$ in (49) ensures $\bar{z}(t(n) + t)$ is well-defined when $t(n) + t < 0$. Precisely speaking, $\forall \tau > 0, \forall t \in [-\tau, \tau]$, the discretization error is defined as

$$\bar{f}_n(t) \doteq \bar{z}(t(n) + t) - \bar{z}_n(t). \tag{50}$$

and we need $\bar{f}_n(t)$ to diminish to 0 as $n \to \infty$. To this end, we study the following three sequences of functions

$$\{\bar{z}(t(n) + t)\}_{n=0}^{\infty}, \{\bar{z}_n(t)\}_{n=0}^{\infty}, \{\bar{f}_n(t)\}_{n=0}^{\infty}.$$

Equicontinuity in the extended sense on the domain $(-\infty, \infty)$ is defined as following (Section 4.2.1 in Kushner & Yin (2003)).

**Definition E.3.** A sequence of functions $\{\gamma_n : (-\infty, \infty) \to \mathbb{R}^K\}$ is equicontinuous in the extended sense on $(-\infty, \infty)$ if $\sup_n \|\gamma_n(0)\| < \infty$ and $\forall \tau > 0, \forall \epsilon > 0, \exists \delta > 0$ such that

$$\limsup_n \sup_{0 \leq |t_1 - t_2| \leq \delta, |t_1| \leq \tau, |t_2| \leq \tau} \|\gamma_n(t_1) - \gamma_n(t_2)\| \leq \epsilon.$$

We show $\{\bar{z}(t(n) + t)\}$, $\{\bar{z}_n(t)\}$ and $\{\bar{f}_n(t)\}$ are all equicontinuous in the extended sense.

---

[4]A set $X$ is an invariant set of the ODE system (47) if and only if for every $z \in Z$, there exists a solution $z(t)$ to the ODE (47) such that $z(0) = z$ and $z(t) \in Z$ for all $t \in (-\infty, \infty)$. If the ODE (47) is globally asymptotically stable, the only bounded invariant set is $\{(\lambda(y), y) : y \in \mathbb{R}^{d_2}\}$.

[5]By $\{z_n\}$ converges to a set $Z$, we mean $\lim_{n \to \infty} \inf_{z \in Z} \|z_n - z\| = 0$.

**Lemma E.4.** *The three sequences of functions $\{\bar{z}(t(n)+t)\}_{n=0}^{\infty}$, $\{\bar{z}_n(t)\}_{n=0}^{\infty}$, and $\{\bar{f}_n(t)\}_{n=0}^{\infty}$ are all equicontinuous in the extended sense on $t \in (-\infty, \infty)$.*

To prove those lemmas, we need the Gronwall inequality in the reverse time in Appendix A.2. Compared to lemmas in the main text which have the domain $t \in [0, T+1)$, lemmas in this section have similar proofs because we first fix an arbitrary $\tau$ and prove properties on the domain $t \in [-\tau, \tau]$. We omit proofs for Lemma E.4 because they are extremely similar to the proofs of equicontinuity in the fast timescale. Similar to Lemma 4.3, we now construct a particular subsequence of interest.

**Lemma E.5.** *There exists a subsequence $\{n_k\}_{k=0}^{\infty} \subseteq \{0, 1, 2, \dots\}$ and some continuous functions $\bar{f}^{\text{lim}}(t)$ and $\bar{z}^{\text{lim}}(t)$ such that $\forall \tau$, $\forall t \in [-\tau, \tau]$,*

$$\lim_{k \to \infty} \bar{f}_{n_k}(t) = \bar{f}^{\text{lim}}(t),$$

$$\lim_{k \to \infty} \bar{z}(t(n_k) + t) = \bar{z}^{\text{lim}}(t),$$

*where both convergences are uniform in $t$ on $[-\tau, \tau]$. Furthermore, let $z^{\text{lim}}(t)$ denote the unique solution to the ODE (47) with the initial condition*

$$z^{\text{lim}}(0) = \bar{z}^{\text{lim}}(0),$$

*in other words,*

$$z^{\text{lim}}(t) = \bar{z}^{\text{lim}}(0) + \int_0^t (h(\bar{z}^{\text{lim}}(s)), 0)ds.$$

*Then $\forall \tau$, $\forall t \in [-\tau, \tau]$, we have*

$$\lim_{k \to \infty} \bar{z}_{n_k}(t) = z^{\text{lim}}(t),$$

*where the convergence is uniform in $t$ on $[-\tau, \tau]$.*

Its proof is very similar to the proof of Lemma 4.3 and is omitted. We use the subsequence $\{n_k\}$ intensively in the remaining proofs. Recall that $\bar{f}_n(t)$ denotes the discretization error between $\bar{z}(t(n) + t)$ and $\bar{z}_n(t)$. We now proceed to prove that this discretization error diminishes along $\{n_k\}$. In particular, we aim to prove that $\forall \tau$, $\forall t \in [-\tau, \tau]$,

$$\lim_{k \to \infty} \left\| \bar{f}_{n_k}(t) \right\| = \left\| \bar{f}^{\text{lim}}(t) \right\| = 0.$$

This means $\bar{z}(t(n_k) + t)$ is close to $\bar{z}_{n_k}(t)$ as $k \to \infty$. For $t \in (0, \tau]$, the proof for this part is the same as the proof we have done in Section C.4. Thus, we only discuss the proof for $t \in [-\tau, 0]$. $\forall \tau$, $\forall t \in [-\tau, 0]$,

$$
\begin{aligned}
&\lim_{k \to \infty} \left\| \bar{f}_{n_k}(t) \right\| \\
&= \lim_{k \to \infty} \left\| \bar{z}(t(n_k)) - \sum_{i=m(t(n_k)+t)}^{n_k-1} (\alpha(i)H(\bar{z}(t(i)), W_{i+1}), \beta(i)G(\bar{z}(t(i)), W_{i+1})) - \bar{z}_{n_k}(t) \right\| \quad \text{(by (49) and (50))} \\
&= \lim_{k \to \infty} \left\| -\sum_{i=m(t(n_k)+t)}^{n_k-1} (\alpha(i)H(\bar{z}(t(i)), W_{i+1}), \beta(i)G(\bar{z}(t(i)), W_{i+1})) - \int_0^t (h(\bar{z}_{n_k}(s)), 0)ds \right\| \quad \text{(by (48))} \\
&\leq \lim_{k \to \infty} \left\| -\sum_{i=m(t(n_k)+t)}^{n_k-1} \alpha(i)H(\bar{z}(t(i)), W_{i+1}) - \int_0^t h(\bar{z}^{\text{lim}}(s))ds \right\| \\
&\quad + \lim_{k \to \infty} \left\| -\sum_{i=m(t(n_k)+t)}^{n_k-1} \beta(i)G(\bar{z}(t(i)), W_{i+1}) \right\| \\
&\quad + \lim_{k \to \infty} \left\| \int_0^t h(\bar{z}^{\text{lim}}(s))ds - \int_0^t h(\bar{z}_{n_k}(s))ds \right\|. \quad (51)
\end{aligned}
$$

The second term in the RHS of (51) is 0.

**Lemma E.6.** $\forall \tau$, $\forall t \in [-\tau, 0]$,

$$\lim_{k \to \infty} \left\| - \sum_{i=m(t(n_k)+t)}^{n_k-1} \beta(i)G(\bar{z}(t(i)), W_{i+1}) \right\| = 0.$$

The proof is very similar to the proof of Lemma C.5 so is omitted.

The first term in the RHS of (51) is also 0.

**Lemma E.7.** $\forall \tau$, $\forall t \in [-\tau, 0]$,

$$\lim_{k \to \infty} \left\| - \sum_{i=m(t(n_k)+t)}^{n_k-1} \alpha(i)H(\bar{z}(t(i)), W_{i+1}) - \int_0^t h(\bar{z}^{\lim}(s))ds \right\| = 0.$$

Its proof is very similar to the proof of Lemma G.9 and is omitted. This convergence is also simpler than (68) because here we have only a single $(H, h)$. But in (68), we have a sequence $\{(H_{n_k}, h_{n_k})\}$, for which we have to split it to a double limit (69) and then invoke the Moore-Osgood theorem to reduce it to the single $(H, h)$ case.

Lemma E.7 confirms that the first term in the RHS of (51) is 0. Moreover, it also enables us to rewrite $\bar{z}^{\lim}(t)$ from a summation form to an integral form. $\forall \tau$, $\forall t \in [-\tau, 0]$

$$\bar{z}^{\lim}(t)$$

$$= \lim_{k \to \infty} \bar{z}(t(n_k)) - \sum_{i=m(t(n_k)+t)}^{n_k-1} (\alpha(i)H(\bar{z}(t(i)), W_{i+1}), \beta(i)G(\bar{z}(t(i)), W_{i+1}))$$

$$= \lim_{k \to \infty} \bar{z}(t(n_k)) + \int_0^t (h(\bar{z}^{\lim}(s)), 0)ds. \qquad \text{(by Lemma E.7)}$$

Thus, we can show the following diminishing discretization error.

**Lemma E.8.** $\forall \tau$, $\forall t \in [-\tau, \tau]$,

$$\lim_{k \to \infty} \left\| \bar{f}_{n_k}(t) \right\| = 0.$$

*Moreover, the convergence is uniform in $t$ on $[-\tau, \tau]$.*

Its proof is very similar to the proof of Lemma G.11 and is omitted. This immediately implies that for any $t \in (-\infty, \infty)$

$$\lim_{k \to \infty} \bar{z}(t(n_k) + t) = z^{\lim}(t). \tag{52}$$

Theorem 3.2 then yields that

$$\sup_{t \in (-\infty, \infty)} \left\| z^{\lim}(t) \right\| < \infty.$$

Let $Z$ be the set of limit points of $\{z_n\}$. By Theorem 3.2, $\sup_n \|z_n\| < \infty$, so $Z$ is bounded and nonempty. We now prove $Z$ is an invariant set of the ODE (47). For any $z \in Z$, there exists a subsequence $\{z_{n_k}\}$ such that

$$\lim_{k \to \infty} z_{n_k} = z.$$

Since $\{\bar{z}(t(n_k) + t)\}$ is equicontinuous in the extended sense, following the way we arrive at (52), we can construct a subsequence $\{n'_k\} \subseteq \{n_k\}$ such that

$$\lim_{k \to \infty} \bar{z}(t(n'_k) + t) = z^{\lim}_*(t), \tag{53}$$

where $z_*^{\lim}(t)$ is a solution to the ODE (47) and $z_*^{\lim}(0) = z$. The remaining is to show that $z_*^{\lim}(t)$ lies entirely in $Z$. For any $t \in (-\infty, \infty)$, by the piecewise constant nature of $\bar{z}$ in (49), the above limit (53) implies that there exists a subsequence of $\{z_n\}$ that converges to $z_*^{\lim}(t)$, indicating $z_*^{\lim}(t) \in Z$ by the definition of the limit set. We now have proved $\forall z \in Z$, there exists a solution $z_*^{\lim}(t)$ to the ODE (47) such that $z_*^{\lim}(0) = z$ and $\forall t \in (-\infty, \infty), z_*^{\lim}(t) \in Z$. This means $Z$ is an invariant set, by definition. In particular, $Z$ is a bounded invariant set.

We now prove that $\{z_n\}$ converges to $Z$. Let $\{z_{n_k}\}$ be any convergent subsequence of $\{z_n\}$ with its limit denoted by $z$. We must have $z \in Z$ by the definition of the limit set. So we have proved that all convergent subsequences of $\{z_n\}$ converge to a point in the bounded invariant set $Z$. If $\{z_n\}$ does not converge to $Z$, there must exist a subsequence $\left\{z_{n'_k}\right\}$ such that $\left\{z_{n'_k}\right\}$ is always away from $Z$ by some small $\epsilon_0 > 0$, i.e., $\forall k$,

$$\inf_{z \in Z} \left\| z_{n'_k} - z \right\| \geq \epsilon_0. \tag{54}$$

But $\left\{z_{n'_k}\right\}$ is bounded, so by the Bolzano-Weierstrass Theorem, it must have a convergent subsequence, which, by the definition of the limit set, converges to some point in $Z$. This contradicts (54). So we must have $\{z_n\}$ converge to $Z$, which is a bounded invariant set of the ODE (47). This completes the proof. $\qquad\square$

Since the invariant set of (47) is $\left\{ (\lambda(y), y) : y \in \mathbb{R}^{d_2} \right\}$, we have

$$\lim_{n \to \infty} \|x_n - \lambda(y_n)\| = 0. \tag{55}$$

We will use this fact to now establish convergence to the unique equilibrium.

### E.2. Slow Timescale Convergence

Returning to slow timescale definitions here.

**Corollary E.9.** *Let Assumptions B.1 - B.7 hold. Then the iterates $\{y_n\}$ generated by (1) converge almost surely to a (sample path dependent) bounded invariant set of the ODE*

$$\frac{\mathrm{d}y(t)}{\mathrm{d}t} = g(\lambda(y(t)), y(t)) \tag{56}$$

*Proof.* To prove convergence results on $t \in (-\infty, \infty)$ in Corollary E.9, we fix an arbitrary sample path $\{z_0, \{W_i\}_{i=1}^{\infty}\}$. The stability results from Theorem 3.2 hold. To prove properties on $t \in (-\infty, \infty)$, we first fix an arbitrary $\tau > 0$ and show properties on $\forall t \in [-\tau, \tau]$.

**Definition E.10.** $\forall n \in \mathbb{N}$, define $\bar{y}_n(t)$ as the solution to the ODE (56) in $(-\infty, \infty)$ with an initial condition

$$\bar{y}_n(0) = \bar{y}(t(n)).$$

$\bar{y}_n(t)$ can also be written as

$$\bar{y}_n(t) = \bar{y}(t(n)) + \int_0^t g(\lambda(\bar{y}_n(s)), \bar{y}_n(s))ds, \quad \forall t \in (-\infty, \infty). \tag{57}$$

We need to show that the error of Euler's discretization diminishes asymptotically. With (16) and (17), $\forall \tau > 0, \forall t \in [-\tau, \tau]$,

$$\bar{z}(t(n) + t) = z_{m(t(n)+t)} \tag{58}$$

$$= \begin{cases} \bar{z}(t(n)) + \sum_{i=n}^{m(t(n)+t)-1}(\alpha(i)H(\bar{z}(t(i)), W_{i+1}), \beta(i)G(\bar{z}(t(i)), W_{i+1})) & \text{if } t \geq 0 \\ \bar{z}(t(n)) - \sum_{i=m(t(n)+t)}^{n-1}(\alpha(i)H(\bar{z}(t(i)), W_{i+1}), \beta(i)G(\bar{z}(t(i)), W_{i+1})) & \text{if } t < 0. \end{cases}$$

Notably, the property (16) that $\forall t < 0, m(t) = 0$ in (58) ensures $\bar{z}(t(n) + t)$ is well-defined when $t(n) + t < 0$. Precisely speaking, $\forall \tau > 0, \forall t \in [-\tau, \tau]$, the discretization error is defined as

$$\bar{f}_n(t) \doteq \bar{y}(t(n) + t) - \bar{y}_n(t). \tag{59}$$

and we need $\bar{f}_n(t)$ to diminish to 0 as $n \to \infty$. To this end, we study the following three sequences of functions

$$\{\bar{y}(t(n) + t)\}_{n=0}^{\infty}, \{\bar{y}_n(t)\}_{n=0}^{\infty}, \{\bar{f}_n(t)\}_{n=0}^{\infty}.$$

We show $\{\bar{y}(t(n) + t)\}$, $\{\bar{y}_n(t)\}$ and $\{\bar{f}_n(t)\}$ are all equicontinuous in the extended sense on the domain $(-\infty, \infty)$.

**Lemma E.11.** *The three sequences of functions $\{\bar{y}(t(n) + t)\}_{n=0}^{\infty}$, $\{\bar{y}_n(t)\}_{n=0}^{\infty}$, and $\{\bar{f}_n(t)\}_{n=0}^{\infty}$ are all equicontinuous in the extended sense on $t \in (-\infty, \infty)$.*

To prove those lemmas, we need the Gronwall inequality in the reverse time in Appendix A.2. Compared to lemmas in the main text which have the domain $t \in [0, T + 1)$, lemmas in this section have similar proofs because we first fix an arbitrary $\tau$ and prove properties on the domain $t \in [-\tau, \tau]$. We omit proofs for Lemma E.11 because they are extremely similar to the proofs of equicontinuity in the slow timescale. Similar to Lemma D.4, we now construct a particular subsequence of interest.

**Lemma E.12.** *There exists a subsequence $\{n_k\}_{k=0}^{\infty} \subseteq \{0, 1, 2, \dots\}$ and some continuous functions $\bar{f}^{\lim}(t)$ and $\bar{y}^{\lim}(t)$ such that $\forall \tau$, $\forall t \in [-\tau, \tau]$,*

$$\lim_{k \to \infty} \bar{f}_{n_k}(t) = \bar{f}^{\lim}(t),$$
$$\lim_{k \to \infty} \bar{y}(t(n_k) + t) = \bar{y}^{\lim}(t),$$

*where both convergences are uniform in $t$ on $[-\tau, \tau]$. Furthermore, let $y^{\lim}(t)$ denote the unique solution to the ODE (56) with the initial condition*

$$y^{\lim}(0) = \bar{y}^{\lim}(0),$$

*in other words,*

$$y^{\lim}(t) = \bar{y}^{\lim}(0) + \int_0^t g(\lambda(\bar{y}^{\lim}), \bar{y}^{\lim}(s))ds.$$

*Then $\forall \tau$, $\forall t \in [-\tau, \tau]$, we have*

$$\lim_{k \to \infty} \bar{y}_{n_k}(t) = y^{\lim}(t),$$

*where the convergence is uniform in $t$ on $[-\tau, \tau]$.*

Its proof is very similar to the proof of Lemma D.4 and is omitted. We use the subsequence $\{n_k\}$ intensively in the remaining proofs. Recall that $\bar{f}_n(t)$ denotes the discretization error between $\bar{y}(t(n) + t)$ and $\bar{y}_n(t)$. We now proceed to prove that this discretization error diminishes along $\{n_k\}$. In particular, we aim to prove that $\forall \tau$, $\forall t \in [-\tau, \tau]$,

$$\lim_{k \to \infty} \left\| \bar{f}_{n_k}(t) \right\| = \left\| \bar{f}^{\lim}(t) \right\| = 0.$$

This means $\bar{y}(t(n_k) + t)$ is close to $\bar{y}_{n_k}(t)$ as $k \to \infty$. For $t \in (0, \tau]$, the proof for this part is the same as the proof we have

done in Section D.4. Thus, we only discuss the proof for $t \in [-\tau, 0]$. $\forall \tau$, $\forall t \in [-\tau, 0]$,

$$
\begin{aligned}
&\lim_{k \to \infty} \left\| \bar{f}_{n_k}(t) \right\| \\
&= \lim_{k \to \infty} \left\| \bar{y}(t(n_k)) - \sum_{i=m(t(n_k)+t)}^{n_k-1} \beta(i) G(\bar{z}(t(i)), W_{i+1}) - \bar{y}_{n_k}(t) \right\| \quad \text{(by (58) and (59))} \\
&= \lim_{k \to \infty} \left\| - \sum_{i=m(t(n_k)+t)}^{n_k-1} \beta(i) G(\bar{z}(t(i)), W_{i+1}) - \int_0^t g(\lambda(\bar{y}_{n_k}), \bar{y}_{n_k}) ds \right\| \quad \text{(by (57))} \\
&\leq \lim_{k \to \infty} \left\| - \sum_{i=m(t(n_k)+t)}^{n_k-1} \beta(i) G(\bar{z}(t(i)), W_{i+1}) + \sum_{i=m(t(n_k)+t)}^{n_k-1} \beta(i) G(\lambda(\bar{y}(t(i))), \bar{y}(t(i)), W_{i+1}) \right\| \\
&\quad + \lim_{k \to \infty} \left\| - \sum_{i=m(t(n_k)+t)}^{n_k-1} \beta(i) G(\lambda(\bar{y}(t(i))), \bar{y}(t(i)), W_{i+1}) - \int_0^t g(\lambda(\bar{y}_{n_k}), \bar{y}_{n_k}) ds \right\|. \quad (60)
\end{aligned}
$$

The first term in the RHS of (60) is 0.

**Lemma E.13.** $\forall \tau$, $\forall t \in [-\tau, 0]$,

$$
\lim_{k \to \infty} \left\| - \sum_{i=m(t(n_k)+t)}^{n_k-1} \beta(i) G(\bar{z}(t(i)), W_{i+1}) + \sum_{i=m(t(n_k)+t)}^{n_k-1} \beta(i) G(\lambda(\bar{y}(t(i))), \bar{y}(t(i)), W_{i+1}) \right\| = 0.
$$

Its proof is very similar to the proof of Lemma D.6 (except that we use (55) instead of Lemma D.5) and is omitted. This convergence is also simpler than (68) because here we have only a single $(G, g)$.

**Lemma E.14.** $\forall \tau$, $\forall t \in [-\tau, \tau]$,

$$
\lim_{k \to \infty} \left\| \bar{f}_{n_k}(t) \right\| = 0.
$$

*Moreover, the convergence is uniform in $t$ on $[-\tau, \tau]$.*

Its proof is very similar to the proof of the diminishing discretization error in the slow timescale. This immediately implies that for any $t \in (-\infty, \infty)$

$$
\lim_{k \to \infty} \bar{z}(t(n_k) + t) = (\lambda(y^{\text{lim}}(t)), y^{\text{lim}}(t)). \quad (61)
$$

Theorem 3.2 then yields that

$$
\sup_{t \in (-\infty, \infty)} \left\| (\lambda(y^{\text{lim}}(t)), y^{\text{lim}}(t)) \right\| < \infty.
$$

Let $Y$ be the set of limit points of $\{y_n\}$. By Theorem 3.2, $\sup_n \|y_n\| < \infty$, so $Y$ is bounded and nonempty. We now prove $Y$ is an invariant set of the ODE (56). For any $y \in Y$, there exists a subsequence $\{y_{n_k}\}$ such that

$$
\lim_{k \to \infty} y_{n_k} = y.
$$

Since $\{\bar{y}(t(n_k) + t)\}$ is equicontinuous in the extended sense, following the way we arrive at (61), we can construct a subsequence $\{n_k'\} \subseteq \{n_k\}$ such that

$$
\lim_{k \to \infty} \bar{y}(t(n_k') + t) = y_*^{\text{lim}}(t), \quad (62)
$$

where $y_*^{\lim}(t)$ is a solution to the ODE (56) and $y_*^{\lim}(0) = y$. The remaining is to show that $y_*^{\lim}(t)$ lies entirely in $Y$. For any $t \in (-\infty, \infty)$, by the piecewise constant nature of $\bar{y}$ in (58), the above limit (62) implies that there exists a subsequence of $\{y_n\}$ that converges to $y_*^{\lim}(t)$, indicating $y_*^{\lim}(t) \in Y$ by the definition of the limit set. We now have proved $\forall y \in Y$, there exists a solution $y_*^{\lim}(t)$ to the ODE (56) such that $y_*^{\lim}(0) = y$ and $\forall t \in (-\infty, \infty), y_*^{\lim}(t) \in Y$. This means $Y$ is an invariant set, by definition. Additionally, $Y$ is a bounded invariant set.

We now prove that $\{y_n\}$ converges to $Y$. Let $\{y_{n_k}\}$ be any convergent subsequence of $\{y_n\}$ with its limit denoted by $y$. We must have $y \in Y$ by the definition of the limit set. So we have proved that all convergent subsequences of $\{y_n\}$ converge to a point in the bounded invariant set $Y$. If $\{y_n\}$ does not converge to $Y$, there must exist a subsequence $\left\{y_{n'_k}\right\}$ such that $\left\{y_{n'_k}\right\}$ is always away from $Y$ by some small $\epsilon_0 > 0$, i.e., $\forall k$,

$$\inf_{y \in Y} \left\| y_{n'_k} - y \right\| \geq \epsilon_0. \tag{63}$$

But $\left\{y_{n'_k}\right\}$ is bounded, so by the Bolzano-Weierstrass Theorem, it must have a convergent subsequence, which, by the definition of the limit set, converges to some point in $Y$. This contradicts (63). So we must have $\{y_n\}$ converge to $Y$, which is a bounded invariant set of the ODE (56). This completes the proof. $\square$

Since the invariant set of (56) is the singleton containing $y^*$, the equilibrium of the ODE, we have

$$\lim_{n \to \infty} \|y_n - y^*\| = 0.$$

Combined with the fast convergence, we have

$$\lim_{n \to \infty} \|z_n - (\lambda(y^*), y^*)\| = 0.$$

# F. Convergence of TDC with Eligibility Traces

TDC was first proposed in Sutton et al. (2008) as a modification of gradient temporal difference learning (GTD) (Sutton et al., 2008). GTD was developed to break the deadly triad, divergence that can arise when combining off-policy learning, function approximation, and bootstrapping, each of which are critical components in successful RL algorithms. While GTD mitigates the deadly triad, it is slow. TDC, on the other hand, is nearly as fast as regular TD learning and converges. It is also a two-timescale algorithm, as the gradient correction runs on a faster timescale. Although vanilla TDC is known to converge, the best prior work was only able to establish the convergence of projected variants of TDC with eligibility traces (Yu, 2017).

We must also explain the important of eligibility traces: they are a powerful tool for credit assignment, a critical challenge in RL, and have been a fundamental part of RL since the inception of the field (Barto & Sutton, 1981). Although eligibility traces are useful, they introduce difficulties in analysis. Even if the state space of the Markov chain $\{(S_t, A_t)\}$ is finite, with eligibility traces, we have to instead consider the chain $\{(S_t, A_t, e_t)\}$, which now evolves in an uncountable space. And, in the case of off-policy learning, the importance sampling ratio can cause the state space to be unbounded as well. Our results, therefore, are the first to be able to handle the important case of off-policy RL algorithms with eligibility traces. We demonstrate this with TDC.

**Assumption F.1.** Both the state space $\mathcal{S}$ and the action space $\mathcal{A}$ are finite. The Markov chain $\{S_t\}$ induced by the behavior policy $\mu$ is irreducible, and $\mu(a|s) > 0$ for all $s, a$.

**Assumption F.2.** The feature matrix $\Phi$ is of full rank.

Assumption F.2 is standard in RL with linear function approximations to guarantee existence and uniqueness of the solution. Additionally, any feature matrix that is not full rank can be converted into one that is by finding a linearly independent basis for the space spanned by the column vectors, so in practice, this is a very weak assumption (Tsitsiklis & Van Roy, 1997).

TDC with eligibility traces is defined as follows:

$$
\begin{aligned}
e_t =& \lambda \gamma \rho_{t-1} e_{t-1} + \phi_t, \\
\delta_t =& R_{t+1} + \gamma \phi_{t+1}^\top \theta_t - \phi_t^\top \theta_t, \\
\nu_{t+1} =& \nu_t + \alpha_t \left( \rho_t \delta_t e_t - \phi_t \phi_t^\top \nu_t \right), \\
\theta_{t+1} =& \theta_t + \beta_t (\rho_t \delta_t e_t - \rho_t (1 - \lambda) \gamma \phi_{t+1} e_t^\top \nu_t).
\end{aligned}
$$

We can more compactly express the updates with the following equations:

$$
\begin{aligned}
\nu_{t+1} &= \nu_t + \alpha_t \left( \begin{bmatrix} -\phi_t \phi_t^\top & \rho_t e_t (\gamma \phi_{t+1} - \phi_t)^\top \end{bmatrix} \begin{bmatrix} \nu_t \\ \theta_t \end{bmatrix} + \begin{bmatrix} \rho_t R_{t+1} e_t \end{bmatrix} \right) \\
\theta_{t+1} &= \theta_t + \beta_t \left( \begin{bmatrix} -\rho_t (1 - \lambda) \gamma \phi_{t+1} e_t^\top & \rho_t e_t (\gamma \phi_{t+1} - \phi_t)^\top \end{bmatrix} \begin{bmatrix} \nu_t \\ \theta_t \end{bmatrix} + \begin{bmatrix} \rho_t R_{t+1} e_t \end{bmatrix} \right).
\end{aligned}
$$

Now we define the augmented Markov chain $\{W_t\}$ as

$$
W_{t+1} \doteq (S_t, A_t, S_{t+1}, e_t), \quad t = 0, 1, \dots.
$$

We also define shorthands

$$x \doteq \nu, x_t \doteq \nu_t, y \doteq \theta, y_t \doteq \theta_t$$
$$w \doteq (s, a, s', e),$$
$$A(w) \doteq \rho(s,a)e(\gamma\phi(s') - \phi(s))^\top,$$
$$b(w) \doteq \rho(s,a)r(s,a)e,$$
$$C(w) \doteq \phi(s)\phi(s)^\top,$$
$$D(w) \doteq e\rho(s,a)(1-\lambda)\gamma\phi(s')^\top,$$
$$H(x,y,w) \doteq \begin{bmatrix} -C(w) & A(w) \end{bmatrix} \begin{bmatrix} x \\ y \end{bmatrix} + b(w),$$
$$G(x,y,w) \doteq \begin{bmatrix} -D^\top(w) & A(w) \end{bmatrix} \begin{bmatrix} x \\ y \end{bmatrix} + b(w).$$

Then TDC can be expressed as

$$x_{t+1} = x_t + \alpha_t H(x_t, y_t, W_{t+1}),$$
$$y_{t+1} = y_t + \beta_t G(x_t, y_t, W_{t+1}),$$

which reduces to the form of (1) and (2).

**Lemma F.1.** *(Theorem 2.1 from Yu (2017)) If Assumption F.1 holds, then*

(i) $\{W_t\}$ *has a unique invariant probability measure,* $d_\mathcal{W}$.

(ii) *The expectation with respect to the stationary distribution,* $\mathbb{E}_{w \sim d_\mathcal{W}}[\gamma(w)] < \infty$ *when* $\gamma(w)$ *is a function which is Lipschitz in the trace variable* $e$. *Additionally,* (LLN) *holds for* $\gamma$.

Yu (2017) also shows that

$$A \doteq \mathbb{E}_{w \sim d_\mathcal{W}}[A(w)] = \Phi^\top D_\mu (P_{\pi,\lambda} - I)\Phi,$$
$$b \doteq \mathbb{E}_{w \sim d_\mathcal{W}}[b(w)] = \Phi^\top D_\mu r_{\pi,\lambda},$$
$$C \doteq \mathbb{E}_{w \sim d_\mathcal{W}}[C(w)] = \Phi^\top D_\mu \Phi,$$
$$D \doteq \mathbb{E}_{w \sim d_\mathcal{W}}[D(w)] = \Phi^\top D_\mu P_{\pi,\lambda}\Phi,$$

where we use $D_\mu$ to denote the diagonal matrix whose diagonal entry is $d_\mu$, the invariant distribution of the underlying Markov chain $\{S_t\}$.

**Theorem (Theorem 7.2 Restated).** *Assume $A$ is invertible (without this assumption, there is no solution so the algorithm itself would be ill-posed as the solution point involves $A^{-1}$). Then, TDC with eligibility traces converges.*

*Proof.* We show that all the assumptions are satisfied such that Theorem 3.3 applies.

Assumption B.1 follows immediately from Lemma F.1.

The assumption B.2 follows immediately from the choice of appropriate learning rates.

For Assumption B.4, define

$$H_\infty(x,y,w) \doteq \begin{bmatrix} -C(w) & A(w) \end{bmatrix} \begin{bmatrix} x \\ y \end{bmatrix},$$
$$G_\infty(x,y,w) \doteq \begin{bmatrix} -D^\top(w) & A(w) \end{bmatrix} \begin{bmatrix} x \\ y \end{bmatrix}.$$

Then we have

$$H_c(x, y, w) - H_\infty(x, y, w) = \frac{b(w)}{c},$$

$$G_c(x, y, w) - G_\infty(x, y, w) = \frac{b(w)}{c}$$

After noticing

$$\|b((s, a, s', e)) - b((s, a, s', e'))\| = \rho(s, a)|r(s, a)|\|e - e'\|, \quad \forall s, a, s', e, e',$$

Assumption B.4 follows immediately from Lemma F.1.

For Assumption B.5, it can be easily verified that $H(x, y, w), G(x, y, w), H_\infty(x, y, w)$, and $G_\infty(x, y, w)$ are Lipschitz continuous in $(x, y)$ for each $w$, with the Lipschitz constant being

$$L(w) = \max \left\{ \left\| \begin{bmatrix} -C(w) & A(w) \end{bmatrix} \right\|, \left\| \begin{bmatrix} -D^\top(w) & A(w) \end{bmatrix} \right\| \right\}$$

Since $A(w), b(w), C(w), D(w)$ are Lipschitz continuous in $e$ for each $(s, a, s')$, Lemma F.1 implies that

$$h(x, y) = \begin{bmatrix} -C & A \end{bmatrix} \begin{bmatrix} x \\ y \end{bmatrix} + b,$$

$$h_\infty(x, y) = \begin{bmatrix} -C & A \end{bmatrix} \begin{bmatrix} x \\ y \end{bmatrix},$$

$$g(x, y) = \begin{bmatrix} -D^\top & A \end{bmatrix} \begin{bmatrix} x \\ y \end{bmatrix} + b$$

$$g_\infty(x, y) = \begin{bmatrix} -D^\top & A \end{bmatrix} \begin{bmatrix} x \\ y \end{bmatrix},$$

$$L = \max \left\{ \left\| \begin{bmatrix} -C & A \end{bmatrix} \right\|, \left\| \begin{bmatrix} -D^\top & A \end{bmatrix} \right\| \right\}.$$

Assumption B.5 then follows.

For Assumption B.6, we have

$$\|h_c(x, y) - h_\infty(x, y)\| = \frac{1}{c}\|b\|,$$

$$\|g_c(x, y) - g_\infty(x, y)\| = \frac{1}{c}\|b\|,$$

so the uniform convergence of $h_c$ to $h_\infty$ and $g_c$ to $g_\infty$ follow immediately.

Since $\Phi$ has full column rank and $D_\mu$ is positive definite (due to irreducibility of $\{S_t\}$), $C = \Phi^\top D_\mu \Phi$ is positive definite, so $-C$ is negative definite. This implies the global asymptotic stability of these two ODEs:

$$\frac{\mathrm{d}x(t)}{\mathrm{d}t} = h(x(t), y) = -Cx(t) + Ay + b, \quad \frac{\mathrm{d}x(t)}{\mathrm{d}t} = h_\infty(x(t), y) = -Cx(t) + Ay.$$

The unique globally asymptotically stable equilibrium of the first is $C^{-1}(Ay + b)$, and of the second is $C^{-1}Ay$. This means that we can define $\lambda(y) = C^{-1}(Ay + b)$ and $\lambda_\infty(y) = C^{-1}Ay$. Note that $\lambda_\infty(cy) = c\lambda_\infty(y)$ as required.

Now we will analyze two ODEs:

$$
\begin{aligned}
\frac{\mathrm{d}y(t)}{\mathrm{d}t} &= g(\lambda(y(t)), y(t)) \\
&= -D^\top \lambda(y(t)) + Ay(t) + b \\
&= -D^\top (C^{-1}(Ay(t) + b)) + Ay(t) + b \\
&= (I - D^\top C^{-1})(Ay(t) + b) \\
&= (I - (A^\top + C)C^{-1})(Ay(t) + b) \qquad\qquad \text{(by } A + C = D, C = C^\top) \\
&= -A^\top C^{-1}(Ay(t) + b),
\end{aligned}
$$

$$
\begin{aligned}
\frac{\mathrm{d}y(t)}{\mathrm{d}t} &= g_\infty(\lambda_\infty(y(t)), y(t)) \\
&= -D^\top \lambda_\infty(y(t)) + Ay(t) \\
&= -D^\top C^{-1} Ay(t) + Ay(t) \\
&= (I - D^\top C^{-1}) Ay(t) \\
&= (I - (A^\top + C) C^{-1}) Ay(t) \qquad\qquad\qquad \text{(by } A + C = D, C = C^\top) \\
&= -A^\top C^{-1} Ay(t),
\end{aligned}
$$

Since $A$ is nonsingular and $C$ is positive definite, $-A^\top C^{-1} A$ is negative definite. This implies the global asymptotic stability of the previous two ODEs. The unique globally asymptotically stable equilibrium of the first is $-A^{-1}b$, and of the second is $0$. Assumption B.6 then follows.

Assumption B.7 follows immediately from Lemma F.1 and Assumption B.2.

Theorem 3.3 then implies that

$$
\lim_{t\to\infty} x_t = 0 \quad \text{a.s.}
$$
$$
\lim_{t\to\infty} y_t = -A^{-1}b \quad \text{a.s.}
$$

which completes the proof. $\qquad\qquad\qquad\qquad\qquad\qquad\qquad\qquad\qquad\qquad\qquad\qquad\square$

# G. Technical Lemmas

## G.1. Asymptotic Rate of Change of Functions in Assumption B.7

**Lemma G.1.** *Let Assumptions B.1, B.2, B.5, and B.7 hold. Then the asymptotic rate of change of the functions that could be represented by $\gamma$ in Assumption B.7 is 0, i.e., for any fixed $\tau > 0$ and $x$, it holds that*

$$\limsup_n \sup_{-\tau \leq t_1 \leq t_2 \leq \tau} \left\| \sum_{i=m(t(n)+t_1)}^{m(t(n)+t_2)-1} \alpha(i) \left[ H(x,y,W_{i+1}) - h(x,y) \right] \right\| = 0 \quad a.s.,$$

$$\limsup_n \sup_{-\tau \leq t_1 \leq t_2 \leq \tau} \left\| \sum_{i=m(t(n)+t_1)}^{m(t(n)+t_2)-1} \alpha(i) \left[ G(x,y,W_{i+1}) - g(x,y) \right] \right\| = 0 \quad a.s.,$$

$$\limsup_n \sup_{-\tau \leq t_1 \leq t_2 \leq \tau} \left\| \sum_{i=m(t(n)+t_1)}^{m(t(n)+t_2)-1} \alpha(i) [L_b(W_{i+1}) - L_b] \right\| = 0 \quad a.s.,$$

$$\limsup_n \sup_{-\tau \leq t_1 \leq t_2 \leq \tau} \left\| \sum_{i=m(t(n)+t_1)}^{m(t(n)+t_2)-1} \alpha(i) [L(W_{i+1}) - L] \right\| = 0 \quad a.s.$$

*and we can replace $\alpha(i)$ with $\beta(i)$ in any of the above.*

The proofs of these results are very similar to the proof of Lemma 9 in Liu et al. (2025b) and so are omitted due to their length.

## G.2. A Uniform Convergence of $H_c, G_c$ to $H_\infty, G_\infty$

**Lemma G.2.** *Let Assumptions B.1, B.2, B.5, and B.7 hold. It then holds that*

$$\lim_{c \to \infty} \sup_{z \in \mathcal{B}} \sup_n \sup_{t \in [0,T]} \left\| \sum_{i=m(T_n)}^{m(T_n+t)-1} \alpha(i) \left[ H_c(z, W_{i+1}) - H_\infty(z, W_{i+1}) \right] \right\| = 0 \quad a.s.,$$

$$\lim_{c \to \infty} \sup_{z \in \mathcal{B}} \sup_n \sup_{t \in [0,T]} \left\| \sum_{i=m(T_n)}^{m(T_n+t)-1} \alpha(i) \left[ G_c(z, W_{i+1}) - G_\infty(z, W_{i+1}) \right] \right\| = 0 \quad a.s.,$$

*where $\mathcal{B}$ denotes an arbitrary compact set of $\mathbb{R}^{d_1+d_2}$. Again, $\alpha(i)$ can be replaced with $\beta(i)$.*

*Proof.* Fix an arbitrary sample path $\{x_0, y_0, \{W_i\}_{i=1}^\infty\}$. Use $\mathcal{B}$ to denote an arbitrary compact subset of $\mathbb{R}^{d_1+d_2}$.

$$\lim_{c \to \infty} \sup_{z \in \mathcal{B}} \sup_n \sup_{t \in [0,T]} \left\| \sum_{i=m(T_n)}^{m(T_n+t)-1} \alpha(i) \left[ H_c(z, W_{i+1}) - H_\infty(z, W_{i+1}) \right] \right\|$$

$$= \lim_{c \to \infty} \sup_{z \in \mathcal{B}} \sup_n \sup_{t \in [0,T]} \left\| \sum_{i=m(T_n)}^{m(T_n+t)-1} \alpha(i)\kappa(c)b(z, W_{i+1}) \right\| \qquad \text{(by (11))}$$

$$= \lim_{c \to \infty} \kappa(c) \sup_{z \in \mathcal{B}} \sup_n \sup_{t \in [0,T]} \left\| \sum_{i=m(T_n)}^{m(T_n+t)-1} \alpha(i)b(z, W_{i+1}) \right\|$$

$$= 0 \sup_{z \in \mathcal{B}} \sup_n \sup_{t \in [0,T]} \left\| \sum_{i=m(T_n)}^{m(T_n+t)-1} \alpha(i)b(z, W_{i+1}) \right\| \qquad (64)$$

We now show that the function

$$z \mapsto \sup_n \sup_{t \in [0,T]} \left\| \sum_{i=m(T_n)}^{m(T_n+t)-1} \alpha(i) b(z, W_{i+1}) \right\| \tag{65}$$

is Lipschitz continuous. $\forall z, z'$,

$$\left| \sup_n \sup_{t \in [0,T]} \left\| \sum_{i=m(T_n)}^{m(T_n+t)-1} \alpha(i) b(z, W_{i+1}) \right\| - \sup_n \sup_{t \in [0,T]} \left\| \sum_{i=m(T_n)}^{m(T_n+t)-1} \alpha(i) b(z', W_{i+1}) \right\| \right|$$

$$\leq \sup_n \sup_{t \in [0,T]} \left| \left\| \sum_{i=m(T_n)}^{m(T_n+t)-1} \alpha(i) b(z, W_{i+1}) \right\| - \left\| \sum_{i=m(T_n)}^{m(T_n+t)-1} \alpha(i) b(z', W_{i+1}) \right\| \right|$$

$$\text{(by } |\sup_z f(z) - \sup_z g(z)| \leq \sup_z |f(z) - g(z)|)$$

$$\leq \sup_n \sup_{t \in [0,T]} \left\| \sum_{i=m(T_n)}^{m(T_n+t)-1} \alpha(i) b(z, W_{i+1}) - \sum_{i=m(T_n)}^{m(T_n+t)-1} \alpha(i) b(z', W_{i+1}) \right\|$$

$$\leq \sup_n \sup_{t \in [0,T]} \sum_{i=m(T_n)}^{m(T_n+t)-1} \alpha(i) \| b(z, W_{i+1}) - b(z', W_{i+1}) \|$$

$$\leq \sup_n \sup_{t \in [0,T]} \left( \sum_{i=m(T_n)}^{m(T_n+t)-1} \alpha(i) L_b(W_{i+1}) \right) \| z - z' \|$$

By Lemma G.1 and (98),

$$\sup_n \sup_{t \in [0,T]} \left( \sum_{i=m(T_n)}^{m(T_n+t)-1} \alpha(i) L_b(W_{i+1}) \right) < \infty$$

can be viewed as the Lipschitz constant. Thus, (65) is a continuous function. Since $\mathcal{B}$ is compact, the extreme value theorems asserts that the supremum of (65) in $\mathcal{B}$ is attainable at some $z_{\mathcal{B}}$ and is finite. This means the RHS of (64) is 0, so

$$\lim_{c \to \infty} \sup_{z \in \mathcal{B}} \sup_n \sup_{t \in [0,T]} \left\| \sum_{i=m(T_n)}^{m(T_n+t)-1} \alpha(i) \left[ H_c(z, W_{i+1}) - H_\infty(z, W_{i+1}) \right] \right\| = 0.$$

The proofs for the other statements follow similar arguments and so are omitted. $\qquad \square$

### G.3. Definitions and Proofs Relating to Equicontinuity

**Definition G.3.** A sequence of functions $\left\{ \gamma_n : [0,T) \to \mathbb{R}^K \right\}$ is equicontinuous on $[0,T)$ if $\sup_n \| \gamma_n(0) \| < \infty$ and $\forall \epsilon > 0, \exists \delta > 0$ such that

$$\sup_n \sup_{0 \leq |t_1 - t_2| \leq \delta, \, 0 \leq t_1 \leq t_2 < T} \| \gamma_n(t_1) - \gamma_n(t_2) \| \leq \epsilon.$$

A standard example of a family of equicontinuous functions is a sequence of bounded Lipschitz continuous functions with a common Lipschitz constant. Clearly, if $\{ \gamma_n \}$ is equicontinuous, each $\gamma_n$ must be continuous. However, the functions of interest in this work, i.e., $\tilde{z}_n(t), f_n(t)$, are not continuous, so equicontinuity cannot apply. We, therefore, define the following equicontinuity in the extended sense.

**Definition G.4.** A sequence of functions $\left\{\gamma_n : [0,T) \to \mathbb{R}^K\right\}$ is equicontinuous in the extended sense on $[0,T)$ if $\sup_n \|\gamma_n(0)\| < \infty$ and $\forall \epsilon > 0, \exists \delta > 0$ such that

$$\limsup_n \sup_{0 \leq |t_1 - t_2| \leq \delta,\, 0 \leq t_1 \leq t_2 < T} \|\gamma_n(t_1) - \gamma_n(t_2)\| \leq \epsilon.$$

The following lemmas establish the desired equicontinuity, where Lemma G.1 plays a key role.

**Lemma G.5.** $\{\tilde{z}_n(t)\}_{n=0}^\infty$ *is equicontinuous in the extended sense on* $[0, T+1)$.

*Proof.* By (19),

$$\sup_n \|\tilde{z}_n(0)\| \leq 1.$$

Without loss of generality, let $t_1 \leq t_2$.

$$\limsup_n \sup_{0 \leq t_2 - t_1 \leq \delta} \|\tilde{z}_n(t_1) - \tilde{z}_n(t_2)\|$$
$$= \limsup_n \sup_{0 \leq t_2 - t_1 \leq \delta} \left\| \sum_{i=m(T_n+t_1)}^{m(T_n+t_2)-1} \left( \alpha(i) H_{r_n}(\tilde{z}_n(t(i) - T_n), W_{i+1}), \beta(i) G_{r_n}(\tilde{z}_n(t(i) - T_n), W_{i+1}) \right) \right\|$$
$$\leq \limsup_n \sup_{0 \leq t_2 - t_1 \leq \delta} \left\| \sum_{i=m(T_n+t_1)}^{m(T_n+t_2)-1} \alpha(i) H_{r_n}(\tilde{z}_n(t(i) - T_n), W_{i+1}) \right\|$$
$$+ \limsup_n \sup_{0 \leq t_2 - t_1 \leq \delta} \left\| \sum_{i=m(T_n+t_1)}^{m(T_n+t_2)-1} \beta(i) G_{r_n}(\tilde{z}_n(t(i) - T_n), W_{i+1}) \right\|.$$

We bound each term individually. We start with the first term.

$\forall \xi > 0$, by (91), $\exists \delta_0$, such that $\forall 0 < \delta \leq \delta_0$,

$$\sup_{c \geq 1} \limsup_n \sup_{0 \leq t_2 - t_1 \leq \delta} \left\| \sum_{i=m(T_n+t_1)}^{m(T_n+t_2)-1} \alpha(i) H_c(0, W_{i+1}) \right\| \leq \xi. \tag{66}$$

By (97), $\exists \delta_1$, such that $\forall 0 < \delta \leq \delta_1$,

$$\limsup_n \sup_{0 \leq t_2 - t_1 \leq \delta} \sum_{i=m(T_n+t_1)}^{m(T_n+t_2)-1} \alpha(i) L(W_{i+1}) \leq \xi. \tag{67}$$

Without loss of generality, let $t_1 \leq t_2$. Then $\forall \delta \leq \min\{\delta_0, \delta_1\}$, we have

$$\limsup_n \sup_{0 \leq t_2 - t_1 \leq \delta} \left\| \sum_{i=m(T_n+t_1)}^{m(T_n+t_2)-1} \alpha(i) H_{r_n}(\tilde{z}_n(t(i) - T_n), W_{i+1}) \right\|$$

$$\leq \limsup_n \sup_{0 \leq t_2 - t_1 \leq \delta} \left\| \left\| \sum_{i=m(T_n+t_1)}^{m(T_n+t_2)-1} \alpha(i) H_{r_n}(\tilde{z}_n(t(i) - T_n), W_{i+1}) \right\| - \left\| \sum_{i=m(T_n+t_1)}^{m(T_n+t_2)-1} \alpha(i) H_{r_n}(0, W_{i+1}) \right\| \right\|$$

$$+ \limsup_n \sup_{0 \leq t_2 - t_1 \leq \delta} \left\| \sum_{i=m(T_n+t_1)}^{m(T_n+t_2)-1} \alpha(i) H_{r_n}(0, W_{i+1}) \right\|$$

$$\leq \limsup_n \sup_{0 \leq t_2 - t_1 \leq \delta} \left\| \left\| \sum_{i=m(T_n+t_1)}^{m(T_n+t_2)-1} \alpha(i) H_{r_n}(\tilde{z}_n(t(i) - T_n), W_{i+1}) \right\| - \left\| \sum_{i=m(T_n+t_1)}^{m(T_n+t_2)-1} \alpha(i) H_{r_n}(0, W_{i+1}) \right\| \right\|$$

$$+ \sup_{c \geq 1} \limsup_n \sup_{0 \leq t_2 - t_1 \leq \delta} \left\| \sum_{i=m(T_n+t_1)}^{m(T_n+t_2)-1} \alpha(i) H_c(0, W_{i+1}) \right\|$$

$$\leq \limsup_n \sup_{0 \leq t_2 - t_1 \leq \delta} \left\| \left\| \sum_{i=m(T_n+t_1)}^{m(T_n+t_2)-1} \alpha(i) H_{r_n}(\tilde{z}_n(t(i) - T_n), W_{i+1}) \right\| - \left\| \sum_{i=m(T_n+t_1)}^{m(T_n+t_2)-1} \alpha(i) H_{r_n}(0, W_{i+1}) \right\| \right\|$$

$$+ \xi \qquad\qquad\qquad \text{(by (66))}$$

$$\leq \limsup_n \sup_{0 \leq t_2 - t_1 \leq \delta} \left\| \sum_{i=m(T_n+t_1)}^{m(T_n+t_2)-1} \alpha(i) H_{r_n}(\tilde{z}_n(t(i) - T_n), W_{i+1}) - \sum_{i=m(T_n+t_1)}^{m(T_n+t_2)-1} \alpha(i) H_{r_n}(0, W_{i+1}) \right\|$$

$$+ \xi$$

$$\leq \limsup_n \sup_{0 \leq t_2 - t_1 \leq \delta} \sum_{i=m(T_n+t_1)}^{m(T_n+t_2)-1} \alpha(i) \| H_{r_n}(\tilde{z}_n(t(i) - T_n), W_{i+1}) - H_{r_n}(0, W_{i+1}) \| + \xi$$

$$\leq \limsup_n \sup_{0 \leq t_2 - t_1 \leq \delta} \sum_{i=m(T_n+t_1)}^{m(T_n+t_2)-1} \alpha(i) L(W_{i+1}) \| \tilde{z}_n(t(i) - T_n) \| + \xi$$

$$\leq (C_{\hat{x}} + C_{\hat{y}}) \limsup_n \sup_{0 \leq t_2 - t_1 \leq \delta} \sum_{i=m(T_n+t_1)}^{m(T_n+t_2)-1} \alpha(i) L(W_{i+1}) + \xi \qquad\qquad \text{(by Lemma H.6)}$$

$$\leq (C_{\hat{x}} + C_{\hat{y}}) \xi + \xi. \qquad\qquad\qquad\qquad\qquad\qquad\qquad\qquad\qquad \text{(by (67))}$$

A similar argument bounds the other term, implying that $\{\tilde{z}_n(t)\}$ is equicontinuous in the extended sense. $\qquad\square$

**Lemma G.6.** $\{z_n(t)\}$ *is equicontinuous on* $[0, T+1)$.

*Proof.* By (19) and (20),

$$\sup_n \| z_n(0) \| \leq 1.$$

Without loss of generality, let $t_1 \leq t_2$. Then $\forall \delta > 0$, we have

$$\sup_n \sup_{0 \leq |t_1 - t_2| \leq \delta,\, 0 \leq t_1 \leq t_2 < T} \|z_n(t_1) - z_n(t_2)\|$$

$$= \sup_n \sup_{0 \leq |t_1 - t_2| \leq \delta,\, 0 \leq t_1 \leq t_2 < T} \left\| \int_{t_1}^{t_2} h_{r_n}(z_n(s)) ds \right\|$$

$$= \sup_n \sup_{0 \leq |t_1 - t_2| \leq \delta,\, 0 \leq t_1 \leq t_2 < T} \left\| \int_{t_1}^{t_2} \left[ h_{r_n}(z_n(s)) - h_{r_n}(0) \right] ds + \int_{t_1}^{t_2} h_{r_n}(0) ds \right\|$$

$$\leq \sup_n \sup_{0 \leq |t_1 - t_2| \leq \delta,\, 0 \leq t_1 \leq t_2 < T} \int_{t_1}^{t_2} \|h_{r_n}(z_n(s)) - h_{r_n}(0)\| ds + \sup_n \sup_{0 \leq |t_1 - t_2| \leq \delta,\, 0 \leq t_1 \leq t_2 < T} \int_{t_1}^{t_2} \|h_{r_n}(0)\| ds$$

$$\leq \sup_n \sup_{0 \leq |t_1 - t_2| \leq \delta,\, 0 \leq t_1 \leq t_2 < T} \int_{t_1}^{t_2} L\|z_n(s)\| ds + \sup_n \sup_{0 \leq |t_1 - t_2| \leq \delta,\, 0 \leq t_1 \leq t_2 < T} \int_{t_1}^{t_2} \|h_{r_n}(0)\| ds \qquad \text{(by Lemma H.2)}$$

$$\leq \delta L C_{\hat{x}} + \sup_n \sup_{0 \leq |t_1 - t_2| \leq \delta,\, 0 \leq t_1 \leq t_2 < T} \int_{t_1}^{t_2} \|h_{r_n}(0)\| ds \qquad \text{(by Lemma H.7)}$$

$$\leq \delta(L C_{\hat{x}} + C_H), \qquad \text{(by (100))}$$

which implies that $\{z_n\}$ is equicontinuous. □

**Lemma G.7.** $\{f_n(t)\}$ *is equicontinuous in the extended sense on* $[0, T+1)$.

*Proof.*

$$\sup_n f_n(0) = \sup_n \tilde{z}_n(0) - z_n(0) = \sup_n \tilde{z}_n(0) - \tilde{z}_n(0) = 0 < \infty.$$

By Lemma G.5 and Lemma G.6, $\forall \epsilon > 0$, $\exists \delta$ such that

$$\limsup_n \sup_{0 \leq t_2 - t_1 \leq \delta} \|\tilde{z}_n(t_1) - \tilde{z}_n(t_2)\| \leq \frac{\epsilon}{2},$$

$$\sup_n \sup_{0 \leq t_2 - t_1 \leq \delta} \|z_n(t_1) - z_n(t_2)\| \leq \frac{\epsilon}{2}.$$

Without loss of generality let $t_1 \leq t_2$. Then $\forall \epsilon$, $\exists \delta$ such that

$$\limsup_n \sup_{0 \leq t_2 - t_1 \leq \delta} \|f_n(t_1) - f_n(t_2)\|$$

$$= \limsup_n \sup_{0 \leq t_2 - t_1 \leq \delta} \|\tilde{z}_n(t_1) - \tilde{z}_n(t_2) - (z_n(t_1) - z_n(t_2))\|$$

$$\leq \limsup_n \sup_{0 \leq t_2 - t_1 \leq \delta} \|\tilde{z}_n(t_1) - \tilde{z}_n(t_2)\| + \limsup_n \sup_{0 \leq t_2 - t_1 \leq \delta} \|z_n(t_1) - z_n(t_2)\|$$

$$\leq \limsup_n \sup_{0 \leq t_2 - t_1 \leq \delta} \|\tilde{z}_n(t_1) - \tilde{z}_n(t_2)\| + \sup_n \sup_{0 \leq t_2 - t_1 \leq \delta} \|z_n(t_1) - z_n(t_2)\|$$

$$\leq \epsilon,$$

which implies that $\{f_n\}$ is equicontinuous in the extended sense. □

### G.4. Proof of Lemma 4.3

*Proof.* Since $\sup_n r_n = \infty$ and $r_n$ is monotonic, $\lim_{n \to \infty} r_n = \infty$, and every subsequence also converges to infinity.

Since $\{f_{n_{k,0}}\}$ is equicontinuous, by the Arzelà-Ascoli theorem (see Appendix A.4), there exists a subsequence $n_{k,1} \subseteq n_{k,0}$ such that $\{f_{n_{k,1}}\}$ converges uniformly to a continuous limit $f^{\lim}$. Similarly, since $\{\tilde{z}_{n_{k,1}}(t)\}$ is equicontinuous, there is a subsequence $n_k \subseteq n_{k,1}$ such that $\{\tilde{z}_{n_k}(t)\}$ converges uniformly in $t$ to a continuous limit $\tilde{z}^{\lim}(t)$.

The proof that $\lim_{n \to \infty} z_{n_k}(t) = z^{\lim}(t)$ uniformly is by lemma H.11. □

## G.5. Proof of Lemma C.5

*Proof.*

$$
\begin{aligned}
&\lim_{k\to\infty} \left\| \sum_{i=m(T_{n_k})}^{m(T_{n_k}+t)-1} \beta(i) G_{r_{n_k}}(\tilde{z}_{n_k}(t(i)-T_{n_k}), W_{i+1}) \right\| \\
&\leq \lim_{k\to\infty} \sum_{i=m(T_{n_k})}^{m(T_{n_k}+t)-1} \beta(i) \left\| G_{r_{n_k}}(\tilde{z}_{n_k}(t(i)-T_{n_k}), W_{i+1}) \right\| \\
&\leq \lim_{k\to\infty} \sum_{i=m(T_{n_k})}^{m(T_{n_k}+t)-1} \beta(i) L(W_{i+1}) \| \tilde{z}_{n_k}(t(i)-T_{n_k}) \| \\
&\leq C_{\hat{z}} \lim_{k\to\infty} \sum_{i=m(T_{n_k})}^{m(T_{n_k}+t)-1} \beta(i) L(W_{i+1}) && \text{(by Lemma H.6)} \\
&\leq 0. && (96)
\end{aligned}
$$

$\square$

## G.6. Bounding the Discretization Error

We will now prove that the first term in the RHS of (24) is 0. We need to show that $\forall t \in [0, T+1)$,

$$
\lim_{k\to\infty} \left\| \sum_{i=m(T_{n_k})}^{m(T_{n_k}+t)-1} \alpha(i) H_{r_{n_k}}(\tilde{z}_{n_k}(t(i)-T_{n_k}), W_{i+1}) - \int_0^t h_{r_{n_k}}(\tilde{z}^{\lim}(s)) ds \right\| = 0. \tag{68}
$$

To evaluate the above, we first fix any $t \in [0, T+1)$ and then compute the following stronger double limit, which implies that the above limit holds.

$$
\lim_{\substack{j\to\infty \\ k\to\infty}} \left\| \sum_{i=m(T_{n_k})}^{m(T_{n_k}+t)-1} \alpha(i) H_{r_{n_j}}(\tilde{z}_{n_k}(t(i)-T_{n_k}), W_{i+1}) - \int_0^t h_{r_{n_j}}(\tilde{z}^{\lim}(s)) ds \right\|. \tag{69}
$$

The Moore-Osgood theorem (Appendix A.5) will help us compute this double limit by turning it into iterated limits. To invoke the Moore-Osgood theorem, we first prove the uniform convergence in $k$ when $j \to \infty$.

**Lemma G.8.** $\forall t \in [0, T+1)$,

$$
\begin{aligned}
&\lim_{j\to\infty} \left\| \sum_{i=m(T_{n_k})}^{m(T_{n_k}+t)-1} \alpha(i) H_{r_{n_j}}(\tilde{z}_{n_k}(t(i)-T_{n_k}), W_{i+1}) - \int_0^t h_{r_{n_k}}(\tilde{z}^{\lim}(s)) ds \right\| \\
&= \left\| \sum_{i=m(T_{n_k})}^{m(T_{n_k}+t)-1} \alpha(i) H_{\infty}(\tilde{z}_{n_k}(t(i)-T_{n_k}), W_{i+1}) - \int_0^t h_{\infty}(\tilde{z}^{\lim}(s)) ds \right\|
\end{aligned}
$$

*uniformly in $k$.*

*Proof.* $\forall j, \forall k, \forall t \in [0, T+1)$,

$$
\left\| \left\| \sum_{i=m(T_{n_k})}^{m(T_{n_k}+t)-1} \alpha(i) H_{r_{n_j}}(\tilde{z}_{n_k}(t(i)-T_{n_k}), W_{i+1}) - \int_0^t h_{r_{n_j}}(\tilde{z}^{\lim}(s))ds \right\| \right.
$$

$$
\left. - \left\| \sum_{i=m(T_{n_k})}^{m(T_{n_k}+t)-1} \alpha(i) H_\infty(\tilde{z}_{n_k}(t(i)-T_{n_k}), W_{i+1}) - \int_0^t h_\infty(\tilde{z}^{\lim}(s))ds \right\| \right\|
$$

$$
\leq \left\| \sum_{i=m(T_{n_k})}^{m(T_{n_k}+t)-1} \alpha(i) H_{r_{n_j}}(\tilde{z}_{n_k}(t(i)-T_{n_k}), W_{i+1}) - \int_0^t h_{r_{n_j}}(\tilde{z}^{\lim}(s))ds \right.
$$

$$
\left. - \sum_{i=m(T_{n_k})}^{m(T_{n_k}+t)-1} \alpha(i) H_\infty(\tilde{z}_{n_k}(t(i)), W_{i+1}) + \int_0^t h_\infty(\tilde{z}^{\lim}(s))ds \right\| \qquad \text{(by } |\|a\| - \|b\|| \leq \|a-b\|)
$$

$$
\leq \left\| \sum_{i=m(T_{n_k})}^{m(T_{n_k}+t)-1} \alpha(i)(H_{r_{n_j}}(\tilde{z}_{n_k}(t(i)-T_{n_k}), W_{i+1}) - H_\infty(\tilde{z}_{n_k}(t(i)-T_{n_k}), W_{i+1})) \right\|
$$

$$
+ \left\| \int_0^t h_{r_{n_j}}(\tilde{z}^{\lim}(s)) - h_\infty(\tilde{z}^{\lim}(s))ds \right\|
$$

$$
\leq \left\| \sum_{i=m(T_{n_k})}^{m(T_{n_k}+t)-1} \alpha(i)(H_{r_{n_j}}(\tilde{z}_{n_k}(t(i)-T_{n_k}), W_{i+1}) - H_\infty(\tilde{z}_{n_k}(t(i)-T_{n_k}), W_{i+1})) \right\|
$$

$$
+ \int_0^t \left\| h_{r_{n_j}}(\tilde{z}^{\lim}(s)) - h_\infty(\tilde{z}^{\lim}(s)) \right\| ds \tag{70}
$$

By Lemma H.6, $\tilde{z}_{n_k}(t(i)-T_{n_k})$ is in a compact set $\mathcal{B}_{\hat{z}}$. By Lemma G.2, for the compact set $\mathcal{B}_{\hat{z}}$, $\forall \epsilon > 0, \exists j_1$ such that $\forall j \geq j_1, \forall k, \forall z \in \mathcal{B}, \forall t \in [0, T+1)$,

$$
\left\| \sum_{i=m(T_{n_k})}^{m(T_{n_k}+t)-1} \alpha(i) \left[ H_{r_{n_j}}(z, W_{i+1}) - H_\infty(z, W_{i+1}) \right] \right\| \leq \epsilon. \tag{71}
$$

Similar to the proof of Lemma H.10, we have

$$
\lim_{j \to \infty} h_{r_{n_j}}(\tilde{z}_k(t)) = h_\infty(\tilde{z}_k(t)) \tag{72}
$$

uniformly in $k$ and $t \in [0, T+1)$. By (72), $\forall \epsilon > 0, \exists j_2$ such that $\forall j > j_2, \forall k, \forall t \in [0, T+1)$,

$$
\left\| h_{r_{n_j}}(\tilde{z}_k(t)) - h_\infty(\tilde{z}_k(t)) \right\| \leq \epsilon. \tag{73}
$$

Define $j_0 \doteq \max\{j_1, j_2\}$. $\forall j \geq j_0, \forall k, \forall t \in [0, T+1)$,

$$\left\| \left\| \sum_{i=m(T_{n_k})}^{m(T_{n_k}+t)-1} \alpha(i) H_{r_{n_j}}(\tilde{z}_{n_k}(t(i) - T_{n_k}), W_{i+1}) - \int_0^t h_{r_{n_j}}(\tilde{z}^{\lim}(s))ds \right\| \right.$$

$$\left. - \left\| \sum_{i=m(T_{n_k})}^{m(T_{n_k}+t)-1} \alpha(i) H_\infty(\tilde{z}_{n_k}(t(i) - T_{n_k}), W_{i+1}) - \int_0^t h_\infty(\tilde{z}^{\lim}(s))ds \right\| \right\|$$

$$\leq \left\| \sum_{i=m(T_{n_k})}^{m(T_{n_k}+t)-1} \alpha(i)(H_{r_{n_j}}(\tilde{z}_{n_k}(t(i) - T_{n_k}), W_{i+1}) - H_\infty(\tilde{z}_{n_k}(t(i) - T_{n_k}), W_{i+1})) \right\| + (T+1)\epsilon \quad \text{(by (70), (73))}$$

$$\leq \epsilon + (T+1)\epsilon \quad \text{(by (70), (71))}$$

$$\leq (T+2)\epsilon.$$

This completes the proof of uniform convergence. $\qquad\square$

Now, we prove, for each $j$, the convergence with $k \to \infty$.

**Lemma G.9.** $\forall t \in [0, T+1)$, $\forall j$,

$$\lim_{k\to\infty} \left\| \sum_{i=m(T_{n_k})}^{m(T_{n_k}+t)-1} \alpha(i) H_{r_{n_j}}(\tilde{z}_{n_j}(t(i) - T_{n_j}), W_{i+1}) - \int_0^t h_{r_{n_j}}(\tilde{z}^{\lim}(s))ds \right\| = 0.$$

The proof of Lemma G.9 is very similar to the proof of Lemma 18 in Liu et al. (2025b) and is omitted here due to its length. We are now ready to compute the limit in (68).

**Lemma G.10.** $\forall t \in [0, T+1)$,

$$\lim_{k\to\infty} \left\| \sum_{i=m(T_{n_k})}^{m(T_{n_k}+t)-1} \alpha(i) H_{r_{n_k}}(\tilde{z}_{n_k}(t(i) - T_{n_k}), W_{i+1}) - \int_0^t h_{r_{n_k}}(\tilde{z}^{\lim}(s))ds \right\| = 0.$$

*Proof.* It follows immediately from Lemmas G.8 & G.9, the Moore-Osgood theorem, and Lemma H.12. $\qquad\square$

Lemma G.10 confirms that the first term in the RHS of (24) is 0. Moreover, it also enables us to rewrite $\tilde{z}^{\lim}(t)$ from a summation form to an integral form.

$$\tilde{z}^{\lim}(t)$$
$$= \lim_{k\to\infty} \tilde{z}_{n_k}(0) + \sum_{i=m(T_{n_k})}^{m(T_{n_k}+t)-1} \alpha(i) H_{r_{n_k}}(\tilde{z}_{n_k}(t(i) - T_{n_k}), W_{i+1})$$
$$= \lim_{k\to\infty} \tilde{z}_{n_k}(0) + \int_0^t h_{r_{n_k}}(\tilde{z}^{\lim}(s))ds. \quad \text{(by Lemma G.10)} \quad (74)$$

This, together with a few Gronwall's inequality arguments, confirms that the discretization error indeed diminishes along $\{n_k\}$.

**Lemma G.11.** $\forall t \in [0, T+1)$,

$$\lim_{k\to\infty} \|f_{n_k}(t)\| = 0.$$

*Proof.* $\forall t \in [0, T+1)$,

$$\lim_{k\to\infty} \|f_{n_k}(t)\|$$

$$\leq \lim_{k\to\infty} \left\| \sum_{i=m(T_{n_k})}^{m(T_{n_k}+t)-1} \alpha(i) H_{r_{n_k}}(\tilde{z}_{n_k}(t(i)-T_{n_k}), W_{i+1}) - \int_0^t h_{r_{n_k}}(\tilde{z}^{\lim}(s))ds \right\|$$

$$+ \lim_{k\to\infty} \left\| \int_0^t h_{r_{n_k}}(\tilde{z}^{\lim}(s))ds - \int_0^t h_{r_{n_k}}(z_{n_k}(s))ds \right\| + 0 \qquad \text{(by (24) and Lemma C.5)}$$

$$= \lim_{k\to\infty} \left\| \int_0^t h_{r_{n_k}}(\tilde{z}^{\lim}(s))ds - \int_0^t h_{r_{n_k}}(z_{n_k}(s))ds \right\|$$

$$= \left\| \int_0^t h_\infty(\tilde{z}^{\lim}(s))ds - \int_0^t h_\infty(z^{\lim}(s))ds \right\|. \qquad \text{(by Lemma H.13 and Lemma H.14)} \quad (75)$$

We now show the relationship between $\tilde{z}^{\lim}(t)$ and $z^{\lim}(t)$.

$$\left\| \tilde{z}^{\lim}(t) - z^{\lim}(t) \right\| \qquad (76)$$

$$= \left\| \lim_{k\to\infty} \left[ \tilde{z}_{n_k}(0) + \int_0^t h_{r_{n_k}}(\tilde{z}^{\lim}(s))ds \right] - \left[ \tilde{z}^{\lim}(0) + \int_0^t h_\infty(z^{\lim}(s))ds \right] \right\| \qquad \text{(by (23) and (74))}$$

$$= \left\| \tilde{z}^{\lim}(0) + \int_0^t h_\infty(\tilde{z}^{\lim}(s))ds - \left[ \tilde{z}^{\lim}(0) + \int_0^t h_\infty(z^{\lim}(s))ds \right] \right\| \qquad \text{(by Lemma H.13)}$$

$$= \left\| \int_0^t h_\infty(\tilde{z}^{\lim}(s))ds - \int_0^t h_\infty(z^{\lim}(s))ds \right\| \qquad (77)$$

$$\leq \int_0^t L \left\| \tilde{z}^{\lim}(s) - z^{\lim}(s) \right\| ds \qquad \text{(by Lemma H.2)}$$

$$\leq 0. \qquad \text{(by Gronwall inequality in Theorem A.1)}$$

Thus,

$$\left\| \lim_{k\to\infty} f_{n_k}(t) \right\|$$

$$\leq \left\| \int_0^t h_\infty(\tilde{z}^{\lim}(s))ds - \int_0^t h_\infty(z^{\lim}(s))ds \right\| \qquad \text{(by (75))}$$

$$= \left\| \tilde{z}^{\lim}(t) - z^{\lim}(t) \right\| \qquad \text{(by (77))}$$

$$\leq 0. \qquad \text{(by (76))}$$

$\square$

### G.7. Proof of Lemma 4.5

*The proof is similar to Lemma 1 from Chapter 3.2 of Borkar (2009).*

*Proof.* By Lyapunov stability, there is a $\delta > 0$ such that any trajectory beginning within $B(\lambda_\infty(y), \delta)$ stays within $\frac{\epsilon}{2}$ of the equilibrium $\lambda_\infty(y)$.

For an initial condition $x$, let $T_x$ be a time at which the trajectory is within $\frac{\delta}{2}$ of the equilibrium. Let $x_1$ be some other initial condition. By definition, Lipschitzness, and the Gronwall inequality, we have

$$\|\eta_\infty^y(t, x) - \eta_\infty^y(t, x_1)\| \leq \|x - x_1\| + L \int_0^t \|\eta_\infty^y(s, x) - \eta_\infty^y(s, x_1)\| ds$$

$$\leq \|x - x_1\| e^{LT_x}$$

for all $t \leq T_x$.

So there is a neighborhood $V_x$ such that for all $x_1 \in V_x$, $\eta_\infty^y(T_x, x_1)$ is within $\delta$ of the equilibrium, which by Lyapunov stability implies that it will always be within $\epsilon$ of the equilibrium after $T_x$.

By compactness, we can cover the set $K$ by a finite number of such intervals and obtain a finite number of times $T_{x_1}, \ldots, T_{x_n}$ and then take the maximum time to be the value of $T_\epsilon$. $\qquad\square$

### G.8. Proof of Lemma 4.6

*The proof is similar to Lemma 2 from chapter 3.2 of Borkar (2009).*

*Proof.* We have

$$\eta_c^{y'(t)}(t, x) = x + \int_0^t h_c(\eta_c^{y'(s)}(s, x), y'(s)) ds,$$

$$\eta_\infty^y(t, x) = x + \int_0^t h_\infty(\eta_\infty^y(s, x), y) ds.$$

Let us define the error term:

$$e(t) = \left\| \eta_c^{y'(t)}(t, x) - \eta_\infty^y(t, x) \right\|.$$

We can bound $e(t)$ by two terms:

$$e(t) \leq \int_0^t \left\| h_c(\eta_c^{y'(s)}(s, x), y'(s)) - h_c(\eta_\infty^y(s, x), y'(s)) \right\| ds$$

$$+ \int_0^t \| h_c(\eta_\infty^y(s, x), y'(s)) - h_\infty(\eta_\infty^y(s, x), y) \| ds$$

To bound the first term, we use Lipschitzness:

$$\int_0^t \left\| h_c(\eta_c^{y'(s)}(s, x), y'(s)) - h_c(\eta_\infty^y(s, x), y'(s)) \right\| ds \leq L \int_0^t \left\| \eta_c^{y'(t)}(t, x) - \eta_\infty^y(t, x) \right\| ds$$

$$= L \int_0^t e(s) ds.$$

To bound the second term, we use Lipschitzness and Assumption B.4:

$$\int_0^t \| h_c(\eta_\infty^y(s, x), y'(s)) - h_\infty(\eta_\infty^y(s, x), y) \| ds \leq \int_0^t \| h_c(\eta_\infty^y(s, x), y'(s)) - h_\infty(\eta_\infty^y(s, x), y'(s)) \| ds$$

$$+ \int_0^t \| h_\infty(\eta_\infty^y(s, x), y'(s)) - h_\infty(\eta_\infty^y(s, x), y) \| ds$$

$$\leq \int_0^t \epsilon(c) ds + L \int_0^t \| y'(s) - y \| ds$$

$$\leq T\epsilon(c) + TL\rho.$$

To conclude, we will use the Gronwall Inequality (Appendix A.1):

$$e(t) \leq T\epsilon(c) + TL\rho + L \int_0^t e(s) ds$$

$$\leq (L\rho + \epsilon(c)) Te^{LT}.$$

$\qquad\square$

## G.9. Proof of Lemma D.3

*Proof.* We know that

$$\sup_n \|\tilde{y}_n(0)\| \le 1.$$

Without loss of generality, let $t_1 \le t_2$.

$$\limsup_n \sup_{0 \le t_2 - t_1 \le \delta} \|\tilde{y}_n(t_1) - \tilde{y}_n(t_2)\| = \limsup_n \sup_{0 \le t_2 - t_1 \le \delta} \left\| \sum_{i=m(T_n+t_1)}^{m(T_n+t_2)-1} \beta(i) G_{r_n}(\tilde{z}_n(t(i) - T_n), W_{i+1}) \right\|$$

$\forall \xi > 0$, by (91), $\exists \delta_0$, such that $\forall 0 < \delta \le \delta_0$,

$$\sup_{c \ge 1} \limsup_n \sup_{0 \le t_2 - t_1 \le \delta} \left\| \sum_{i=m(T_n+t_1)}^{m(T_n+t_2)-1} \beta(i) G_c(0, W_{i+1}) \right\| \le \xi. \tag{78}$$

By (97), $\exists \delta_1$, such that $\forall 0 < \delta \le \delta_1$,

$$\limsup_n \sup_{0 \le t_2 - t_1 \le \delta} \sum_{i=m(T_n+t_1)}^{m(T_n+t_2)-1} \beta(i) L(W_{i+1}) \le \xi. \tag{79}$$

Without loss of generality, let $t_1 \leq t_2$. Then $\forall \delta \leq \min\{\delta_0, \delta_1\}$, we have

$$\limsup_n \sup_{0 \leq t_2 - t_1 \leq \delta} \left\| \sum_{i=m(T_n+t_1)}^{m(T_n+t_2)-1} \beta(i) G_{r_n}(\tilde{z}_n(t(i) - T_n), W_{i+1}) \right\|$$

$$\leq \limsup_n \sup_{0 \leq t_2 - t_1 \leq \delta} \left\| \sum_{i=m(T_n+t_1)}^{m(T_n+t_2)-1} \beta(i) G_{r_n}(\tilde{z}_n(t(i) - T_n), W_{i+1}) \right\| - \left\| \sum_{i=m(T_n+t_1)}^{m(T_n+t_2)-1} \beta(i) G_{r_n}(0, W_{i+1}) \right\|$$

$$+ \limsup_n \sup_{0 \leq t_2 - t_1 \leq \delta} \left\| \sum_{i=m(T_n+t_1)}^{m(T_n+t_2)-1} \beta(i) G_{r_n}(0, W_{i+1}) \right\|$$

$$\leq \limsup_n \sup_{0 \leq t_2 - t_1 \leq \delta} \left\| \sum_{i=m(T_n+t_1)}^{m(T_n+t_2)-1} \beta(i) G_{r_n}(\tilde{z}_n(t(i) - T_n), W_{i+1}) \right\| - \left\| \sum_{i=m(T_n+t_1)}^{m(T_n+t_2)-1} \beta(i) G_{r_n}(0, W_{i+1}) \right\|$$

$$+ \sup_{c \geq 1} \limsup_n \sup_{0 \leq t_2 - t_1 \leq \delta} \left\| \sum_{i=m(T_n+t_1)}^{m(T_n+t_2)-1} \beta(i) G_c(0, W_{i+1}) \right\|$$

$$\leq \limsup_n \sup_{0 \leq t_2 - t_1 \leq \delta} \left\| \sum_{i=m(T_n+t_1)}^{m(T_n+t_2)-1} \beta(i) G_{r_n}(\tilde{z}_n(t(i) - T_n), W_{i+1}) \right\| - \left\| \sum_{i=m(T_n+t_1)}^{m(T_n+t_2)-1} \beta(i) G_{r_n}(0, W_{i+1}) \right\|$$

$$+ \xi \tag{by (78)}$$

$$\leq \limsup_n \sup_{0 \leq t_2 - t_1 \leq \delta} \left\| \sum_{i=m(T_n+t_1)}^{m(T_n+t_2)-1} \beta(i) G_{r_n}(\tilde{z}_n(t(i) - T_n), W_{i+1}) - \sum_{i=m(T_n+t_1)}^{m(T_n+t_2)-1} \beta(i) G_{r_n}(0, W_{i+1}) \right\|$$

$$+ \xi$$

$$\leq \limsup_n \sup_{0 \leq t_2 - t_1 \leq \delta} \sum_{i=m(T_n+t_1)}^{m(T_n+t_2)-1} \beta(i) \| G_{r_n}(\tilde{z}_n(t(i) - T_n), W_{i+1}) - G_{r_n}(0, W_{i+1}) \| + \xi$$

$$\leq \limsup_n \sup_{0 \leq t_2 - t_1 \leq \delta} \sum_{i=m(T_n+t_1)}^{m(T_n+t_2)-1} \beta(i) L(W_{i+1}) \| \tilde{z}_n(t(i) - T_n) \| + \xi$$

$$\leq E \limsup_n \sup_{0 \leq t_2 - t_1 \leq \delta} \sum_{i=m(T_n+t_1)}^{m(T_n+t_2)-1} \beta(i) L(W_{i+1}) + \xi \tag{by Lemma H.19}$$

$$\leq E\xi + \xi. \tag{by (79)}$$

$\square$

### G.10. Proof of Lemma D.4

*Proof.* Since $\sup_n r_n = \infty$ and $r_n$ is monotonic, $\lim_{n \to \infty} r_n = \infty$, and every subsequence also converges to infinity.

Since $\{f_{n_{k,0}}\}$ is equicontinuous, by the Arzelà-Ascoli theorem (see Appendix A.4), there exists a subsequence $n_{k,1} \subseteq n_{k,0}$ such that $\{f_{n_{k,1}}\}$ converges uniformly to a continuous limit $f^{\text{lim}}$. Similarly, since $\{\tilde{y}_{n_{k,1}}(t)\}$ is equicontinuous, there is a subsequence $n_k \subseteq n_{k,1}$ such that $\{\tilde{z}_{n_k}(t)\}$ converges uniformly in $t$ to a continuous limit $\tilde{y}^{\text{lim}}(t)$.

The proof that $\lim_{n \to \infty} y_{n_k}(t) = y^{\text{lim}}(t)$ uniformly is by lemma H.18. $\square$

## H. Auxiliary Lemmas

The next lemma establishes some results about the relationship between the step sizes and $T$. It applies to the fast timescale, but a similar result to (80) holds for $\beta(i)$ in the slow timescale.

**Lemma H.1.**

$$\forall n, \ T_{n+1} - T_n \geq T,$$
$$\lim_{n \to \infty} T_{n+1} - T_n = T.$$

*Moreover, $\forall \tau > 0, t_1, t_2$ such that $-\tau \leq t_1 \leq t_2 \leq \tau$, we have*

$$\lim_{n \to \infty} \sum_{i=m(t(n)+t_1)}^{m(t(n)+t_2)-1} \alpha(i) = t_2 - t_1, \tag{80}$$

$$\lim_{n \to \infty} \sum_{i=m(t(n)+t_1)}^{m(t(n)+t_2)-1} \beta(i) = 0. \tag{81}$$

*Proof.* $\forall n$,

$$T_{n+1} - T_n$$
$$= t(m(T_n + T) + 1) - T_n$$
$$\geq T_n + T - T_n$$
$$\geq T.$$

Thus,

$$\lim_{n \to \infty} T_{n+1} - T_n \geq T.$$

With

$$\lim_{n \to \infty} T_{n+1} - T_n$$
$$= \lim_{n \to \infty} t(m(T_n + T) + 1) - T_n$$
$$= \lim_{n \to \infty} t(m(T_n + T)) + \alpha(m(T_n + T)) - T_n$$
$$\leq \lim_{n \to \infty} T_n + T + \alpha(m(T_n + T)) - T_n$$
$$= T,$$

by the squeeze theorem, we have $\lim_{n \to \infty} T_{n+1} - T_n = T$.

To prove (80), $\forall \tau, \forall -\tau \leq t_1 \leq t_2 \leq \tau$, it suffices to only consider large $n$ such that $t(n) - \tau \geq 0$. We have

$$\lim_{n \to \infty} \sum_{i=m(t(n)+t_1)}^{m(t(n)+t_2)-1} \alpha(i)$$
$$= \lim_{n \to \infty} t(m(t(n) + t_2)) - t(m(t(n) + t_1))$$
$$\leq \lim_{n \to \infty} t(n) + t_2 - t(m(t(n) + t_1))$$
$$\leq \lim_{n \to \infty} t(n) + t_2 - (t(n) + t_1 - \alpha(m(t(n) + t_1)))$$
$$= t_2 - t_1 + \lim_{n \to \infty} \alpha(m(t(n) + t_1))$$
$$= t_2 - t_1 \tag{by (7)}$$

and

$$\lim_{n\to\infty} \sum_{i=m(t(n)+t_1)}^{m(t(n)+t_2)-1} \alpha(i)$$

$$= \lim_{n\to\infty} t(m(t(n)+t_2)) - t(m(t(n)+t_1))$$

$$\geq \lim_{n\to\infty} t(n) + t_2 - \alpha(m(t(n)+t_2)) - t(m(t(n)+t_1))$$

$$\geq \lim_{n\to\infty} t(n) + t_2 - \alpha(m(t(n)+t_2)) - (t(n)+t_1)$$

$$= \lim_{n\to\infty} t_2 - t_1 - \alpha(m(t(n)+t_2))$$

$$= t_2 - t_1. \tag{by (7)}$$

By the squeeze theorem, we have

$$\lim_{n\to\infty} \sum_{i=m(t(n)+t_1)}^{m(t(n)+t_2)-1} \alpha(i) = t_2 - t_1.$$

To prove (81), fix $\epsilon > 0$. By (8), there exists $N_1 \in \mathbb{N}$ such that for all $n > m(t(N_1)+t_1)$,

$$\frac{\beta(n)}{\alpha(n)} < \epsilon. \tag{82}$$

By (80), there exists $N_2 \in \mathbb{N}$ such that for all $n > N_2$,

$$\sum_{i=m(t(n)+t_1)}^{m(t(n)+t_2)-1} \alpha(i) < t_2 - t_1 + \epsilon. \tag{83}$$

Let $N = \max\{N_1, N_2\}$. Then for all $n > N$,

$$\sum_{i=m(t(n)+t_1)}^{m(t(n)+t_2)-1} \beta(i)$$

$$< \epsilon \sum_{i=m(t(n)+t_1)}^{m(t(n)+t_2)-1} \alpha(i) \tag{by (82)}$$

$$< \epsilon(t_2 - t_1 + \epsilon). \tag{by (83)}$$

Since $t_2 - t_1$ is a constant and $\epsilon$ can be made arbitrarily small, we have

$$\lim_{n\to\infty} \sum_{i=m(t(n)+t_1)}^{m(t(n)+t_2)-1} \beta(i) = 0.$$

$$\square$$

This lemma establishes Lipschitzness for a few classes of functions.

**Lemma H.2.** *For any $x, x', y, y', w, c \geq 1$, including $c = \infty$,*

$$\|H_c(x,y,w) - H_c(x',y',w)\| \leq L(w)\|(x,y) - (x',y')\|, \tag{84}$$

$$\|h_c(x,y) - h_c(x',y')\| \leq L\|(x,y) - (x',y')\|. \tag{85}$$

$$\|G_c(x,y,w) - G_c(x',y',w)\| \leq L(w)\|(x,y) - (x',y')\|, \tag{86}$$

$$\|g_c(x,y) - g_c(x',y')\| \leq L\|(x,y) - (x',y')\|. \tag{87}$$

*Proof.* To prove (84), we first consider $1 \le c < \infty$,

$$
\begin{aligned}
&\|H_c(x, y, w) - H_c(x', y', w)\| \\
=&\left\| \frac{H(cx, cy, w)}{c} - \frac{H(cx', cy, w)}{c} \right\| && \text{(by (9))} \\
\le&\frac{\|H(cx, cy, w) - H(cx', cy', w)\|}{c} \\
\le&L(w)\frac{\|(cx, cy) - (cx', cy')\|}{c} && \text{(by (12))} \\
=&L(w)\|(x, y) - (x', y')\|.
\end{aligned}
$$

By (13),

$$
\|H_\infty(x, y, w) - H_\infty(x', y', w)\| \le L(w)\|(x, y) - (x', y')\|.
$$

To prove (85), $\forall x, x', y, y', \forall c \ge 1$ including $c = \infty$,

$$
\begin{aligned}
&\|h_c(x, y) - h_c(x', y')\| \\
=&\|\mathbb{E}_{w \sim \omega}\left[ H_c(x, y, w) - H_c(x', y', w) \right]\| \\
\le&\mathbb{E}_{w \sim \omega}\left[\| H_c(x, y, w) - H_c(x', y', w)\|\right] \\
\le&\mathbb{E}_{w \sim \omega}\left[L(w)\|(x, y) - (x', y')\|\right] \\
\le&L\|(x, y) - (x', y')\|.
\end{aligned}
$$

By similar arguments, (86) and (87) also follow. $\qquad\square$

The following lemma gives some regularity results in the fast timescale.

**Lemma H.3.** $\forall x, y,$

$$
\sup_{c \ge 1} \|h_c(0)\| < \infty, \sup_{c \ge 1} \|g_c(0)\| < \infty, \tag{88}
$$

$$
\sup_{c \ge 1} \limsup_{n} \sup_{0 \le t_1 \le t_2 \le T_{n+1} - T_n} \left\| \sum_{i=m(T_n+t_1)}^{m(T_n+t_2)-1} \alpha(i) \left[ H_c(x, y, W_{i+1}) - h_c(x, y) \right] \right\| = 0 \quad a.s., \tag{89}
$$

$$
\sup_{c \ge 1} \sup_{n} \sup_{0 \le t_1 \le t_2 \le T_{n+1} - T_n} \left\| \sum_{i=m(T_n+t_1)}^{m(T_n+t_2)-1} \alpha(i) H_c(0, W_{i+1}) \right\| < \infty \quad a.s., \tag{90}
$$

$$
\lim_{\delta \to 0^+} \sup_{c \ge 1} \limsup_{n} \sup_{0 \le t_2 - t_1 \le \delta} \left\| \sum_{i=m(T_n+t_1)}^{m(T_n+t_2)-1} \alpha(i) H_c(0, W_{i+1}) \right\| = 0 \quad a.s. \tag{91}
$$

*By the same arguments, we can show the last three results with the substitutions of $G$ for $H$ and $\beta(i)$ for $\alpha(i)$.*

*Proof.* **Proof of** (88):

$$
\sup_{c \ge 1} \|h_c(0)\| = \sup_{c \ge 1} \left\| \frac{h(0)}{c} \right\| \le \sup_{c \ge 1} \|h(0)\| = \|h(0)\| < \infty, \tag{92}
$$

$$
\sup_{c \ge 1} \|g_c(0)\| = \sup_{c \ge 1} \left\| \frac{g(0)}{c} \right\| \le \sup_{c \ge 1} \|g(0)\| = \|g(0)\| < \infty.
$$

**Proof of** (89)**:** $\forall x$,

$$\sup_{c \geq 1} \limsup_n \sup_{0 \leq t_1 \leq t_2 \leq T_{n+1}-T_n} \left\| \sum_{i=m(T_n+t_1)}^{m(T_n+t_2)-1} \alpha(i) \left[ H_c(x,y,W_{i+1}) - h_c(x,y) \right] \right\|$$

$$= \sup_{c \geq 1} \limsup_n \sup_{0 \leq t_1 \leq t_2 \leq T_{n+1}-T_n} \left\| \sum_{i=m(T_n+t_1)}^{m(T_n+t_2)-1} \alpha(i) \left[ \frac{H(cx,cy,W_{i+1})}{c} - \frac{h(cx,cy)}{c} \right] \right\|$$

$$= \sup_{c \geq 1} \frac{1}{c} \limsup_n \sup_{0 \leq t_1 \leq t_2 \leq T_{n+1}-T_n} \left\| \sum_{i=m(T_n+t_1)}^{m(T_n+t_2)-1} \alpha(i) \left[ H(cx,cy,W_{i+1}) - h(cx,cy) \right] \right\|$$

$$\leq \sup_{c \geq 1} \frac{1}{c} \limsup_n \sup_{0 \leq t_1 \leq t_2 \leq T+\sup_j \alpha(j)} \left\| \sum_{i=m(T_n+t_1)}^{m(T_n+t_2)-1} \alpha(i) \left[ H(cx,cy,W_{i+1}) - h(cx,cy) \right] \right\|$$

$$\hspace{6cm} (\forall n, T_{n+1} - T_n \leq T + \sup_j \alpha(j))$$

$$= \sup_{c \geq 1} \frac{1}{c} \cdot 0$$

$$= 0. \tag{93}$$

**Proof of** (90)**:**

$$\limsup_n \sup_{0 \leq t_1 \leq t_2 \leq T_{n+1}-T_n} \left\| \sum_{i=m(T_n+t_1)}^{m(T_n+t_2)-1} \alpha(i) H(0,W_{i+1}) \right\|$$

$$= \limsup_n \sup_{0 \leq t_1 \leq t_2 \leq T_{n+1}-T_n} \left\| \sum_{i=m(T_n+t_1)}^{m(T_n+t_2)-1} \alpha(i) [H(0,W_{i+1}) - h(0) + h(0)] \right\|$$

$$\leq \limsup_n \sup_{0 \leq t_1 \leq t_2 \leq T_{n+1}-T_n} \left\| \sum_{i=m(T_n+t_1)}^{m(T_n+t_2)-1} \alpha(i) [H(0,W_{i+1}) - h(0)] \right\|$$

$$+ \limsup_n \sup_{0 \leq t_1 \leq t_2 \leq T_{n+1}-T_n} \left\| \sum_{i=m(T_n+t_1)}^{m(T_n+t_2)-1} \alpha(i) h(0) \right\|$$

$$\leq \limsup_n \sup_{0 \leq t_1 \leq t_2 \leq T+\sup_j \alpha(j)} \left\| \sum_{i=m(T_n+t_1)}^{m(T_n+t_2)-1} \alpha(i) [H(0,W_{i+1}) - h(0)] \right\|$$

$$+ \limsup_n \sup_{0 \leq t_1 \leq t_2 \leq T+\sup_j \alpha(j)} \left\| \sum_{i=m(T_n+t_1)}^{m(T_n+t_2)-1} \alpha(i) h(0) \right\| \hspace{1cm} (\forall n, T_{n+1} - T_n \leq T + \sup_j \alpha(j))$$

$$= \limsup_n \sup_{0 \leq t_1 \leq t_2 \leq T+\sup_j \alpha(j)} \left\| \sum_{i=m(T_n+t_1)}^{m(T_n+t_2)-1} \alpha(i) h(0) \right\|$$

$$= \|h(0)\| \limsup_n \sup_{0 \leq t_1 \leq t_2 \leq T+\sup_j \alpha(j)} \sum_{i=m(T_n+t_1)}^{m(T_n+t_2)-1} \alpha(i)$$

$$= \|h(0)\| (T + \sup_j \alpha(j)) \hspace{4cm} \text{(by Lemma H.1)}$$

$$< \infty. \tag{94}$$

We now consider $c$ in the above bounds. We first get

$$\sup_{c \geq 1} \sup_{n} \sup_{0 \leq t_1 \leq t_2 \leq T_{n+1}-T_n} \left\| \sum_{i=m(T_n+t_1)}^{m(T_n+t_2)-1} \alpha(i) H_c(0, W_{i+1}) \right\|$$

$$= \sup_{c \geq 1} \sup_{n} \sup_{0 \leq t_1 \leq t_2 \leq T_{n+1}-T_n} \left\| \sum_{i=m(T_n+t_1)}^{m(T_n+t_2)-1} \alpha(i) \frac{H(0, W_{i+1})}{c} \right\| \qquad \text{(by (9))}$$

$$= \sup_{n} \sup_{0 \leq t_1 \leq t_2 \leq T_{n+1}-T_n} \left\| \sum_{i=m(T_n+t_1)}^{m(T_n+t_2)-1} \alpha(i) H(0, W_{i+1}) \right\| \qquad \text{(by } c \geq 1\text{)}$$

$$< \infty. \qquad \text{(by (94))}$$

**Proof of (91):**

$$\limsup_{\delta \to 0^+} \sup_{c \geq 1} \limsup_{n} \sup_{0 \leq t_2 - t_1 \leq \delta} \left\| \sum_{i=m(T_n+t_1)}^{m(T_n+t_2)-1} \alpha(i) H_c(0, W_{i+1}) \right\|$$

$$\leq \limsup_{\delta \to 0^+} \sup_{c \geq 1} \limsup_{n} \sup_{0 \leq t_2 - t_1 \leq \delta} \left\| \sum_{i=m(T_n+t_1)}^{m(T_n+t_2)-1} \alpha(i) \left[ H_c(0, W_{i+1}) - h_c(0) \right] \right\|$$

$$+ \limsup_{\delta \to 0^+} \sup_{c \geq 1} \limsup_{n} \sup_{0 \leq t_2 - t_1 \leq \delta} \left\| \sum_{i=m(T_n+t_1)}^{m(T_n+t_2)-1} \alpha(i) h_c(0) \right\|$$

$$\leq 0 + \limsup_{\delta \to 0^+} \sup_{c \geq 1} \limsup_{n} \sup_{0 \leq t_2 - t_1 \leq \delta} \left\| \sum_{i=m(T_n+t_1)}^{m(T_n+t_2)-1} \alpha(i) h_c(0) \right\| \qquad \text{(by (93))}$$

$$\leq 0 + \limsup_{\delta \to 0^+} \sup_{c \geq 1} \limsup_{n} \sup_{0 \leq t_2 - t_1 \leq \delta} \left\| \sum_{i=m(T_n+t_1)}^{m(T_n+t_2)-1} \alpha(i) \frac{h(0)}{c} \right\|$$

$$\leq 0 + \|h(0)\| \limsup_{\delta \to 0^+} \sup_{c \geq 1} \frac{1}{c} \limsup_{n} \sup_{0 \leq t_2 - t_1 \leq \delta} \sum_{i=m(T_n+t_1)}^{m(T_n+t_2)-1} \alpha(i)$$

$$\leq \|h(0)\| \limsup_{\delta \to 0^+} \sup_{c \geq 1} \frac{1}{c} \delta \qquad \text{(by (80))}$$

$$= \|h(0)\| \lim_{\delta \to 0^+} \delta$$

$$= 0.$$

$\square$

We now have some more regularity results in the fast timescale.

**Lemma H.4.**

$$\sup_{n} \sup_{0 \leq t_1 \leq t_2 \leq T_{n+1}-T_n} \left\| \sum_{i=m(T_n+t_1)}^{m(T_n+t_2)-1} \alpha(i)L(W_{i+1}) \right\| < \infty \quad a.s., \tag{95}$$

$$\lim_{n \to \infty} \sup_{0 \leq t_1 \leq t_2 \leq T_{n+1}-T_n} \left\| \sum_{i=m(T_n+t_1)}^{m(T_n+t_2)-1} \beta(i)L(W_{i+1}) \right\| = 0 \quad a.s., \tag{96}$$

$$\lim_{\delta \to 0^+} \limsup_{n} \sup_{0 \leq t_2-t_1 \leq \delta} \left\| \sum_{i=m(T_n+t_1)}^{m(T_n+t_2)-1} \alpha(i)L(W_{i+1}) \right\| = 0 \quad a.s., \tag{97}$$

$$\sup_{n} \sup_{0 \leq t_1 \leq t_2 \leq T_{n+1}-T_n} \left\| \sum_{i=m(T_n+t_1)}^{m(T_n+t_2)-1} \alpha(i)L_b(W_{i+1}) \right\| < \infty \quad a.s. \tag{98}$$

These proofs are similar to the proofs of Lemma H.3 and are thus omitted, except for the proof of (96) (since this is a two-timescale novelty and we need it for Lemma C.5). For the other statements, we can also replace $\alpha(i)$ with $\beta(i)$.

*Proof.* Here we show 96:

$$\lim_{n} \sup_{0 \leq t_1 \leq t_2 \leq T_{n+1}-T_n} \left\| \sum_{i=m(T_n+t_1)}^{m(T_n+t_2)-1} \beta(i)L(W_{i+1}) \right\|$$

$$= \lim_{n} \sup_{0 \leq t_1 \leq t_2 \leq T_{n+1}-T_n} \left\| \sum_{i=m(T_n+t_1)}^{m(T_n+t_2)-1} \beta(i)[L(W_{i+1}) - L + L] \right\|$$

$$\leq \lim_{n} \sup_{0 \leq t_1 \leq t_2 \leq T_{n+1}-T_n} \left\| \sum_{i=m(T_n+t_1)}^{m(T_n+t_2)-1} \beta(i)[L(W_{i+1}) - L] \right\|$$

$$+ \lim_{n} \sup_{0 \leq t_1 \leq t_2 \leq T_{n+1}-T_n} \left\| \sum_{i=m(T_n+t_1)}^{m(T_n+t_2)-1} \beta(i)L \right\|$$

$$\leq \lim_{n} \sup_{0 \leq t_1 \leq t_2 \leq T+\sup_j \alpha(j)} \left\| \sum_{i=m(T_n+t_1)}^{m(T_n+t_2)-1} \beta(i)[L(W_{i+1}) - L] \right\|$$

$$+ \lim_{n} \sup_{0 \leq t_1 \leq t_2 \leq T+\sup_j \alpha(j)} \left\| \sum_{i=m(T_n+t_1)}^{m(T_n+t_2)-1} \beta(i)L \right\| \qquad (\forall n, T_{n+1} - T_n \leq T + \sup_j \alpha(j))$$

$$= \lim_{n} \sup_{0 \leq t_1 \leq t_2 \leq T+\sup_j \alpha(j)} \left\| \sum_{i=m(T_n+t_1)}^{m(T_n+t_2)-1} \beta(i)L \right\|$$

$$= \|L\| \lim_{n} \sup_{0 \leq t_1 \leq t_2 \leq T+\sup_j \alpha(j)} \sum_{i=m(T_n+t_1)}^{m(T_n+t_2)-1} \beta(i)$$

$$= 0. \qquad \text{(by Lemma H.1)}$$

$\square$

We now show that several different quantities are bounded in the fast timescale.

**Lemma H.5.** *Fix a sample path $\{x_0, y_0, \{W_i\}_{i=1}^{\infty}\}$, there exist constants $C_H, C_G$ such that*

$$LT \leq C_H, LT \leq C_G \tag{99}$$

$$\sup_{c \geq 1} \|h_c(0)\| \leq \frac{C_H}{T}, \sup_{c \geq 1} \|g_c(0)\| \leq \frac{C_G}{T} \tag{100}$$

$$\sup_{c \geq 1} \sup_n \sup_{0 \leq t_1 \leq t_2 \leq T_{n+1}-T_n} \left\| \sum_{i=m(T_n+t_1)}^{m(T_n+t_2)-1} \alpha(i) H_c(0, W_{i+1}) \right\| \leq C_H, \tag{101}$$

$$\sup_{c \geq 1} \sup_n \sup_{0 \leq t_1 \leq t_2 \leq T_{n+1}-T_n} \left\| \sum_{i=m(T_n+t_1)}^{m(T_n+t_2)-1} \beta(i) G_c(0, W_{i+1}) \right\| \leq C_G,$$

$$\sup_n \sup_{0 \leq t_1 \leq t_2 \leq T_{n+1}-T_n} \sum_{i=m(T_n+t_1)}^{m(T_n+t_2)-1} \alpha(i) L(W_{i+1}) \leq C_H. \tag{102}$$

*We can replace $\alpha(i)$ with $\beta(i)$ in the last statement. Moreover, for convenience of presentation, we denote*

$$C_{\hat{x}} \doteq [1 + C_H] e^{C_H}, C_{\hat{y}} \doteq [1 + C_G] e^{C_G}, C' \doteq C_G + C_H, C_{\hat{z}} \doteq [1 + C'] e^{C'} \tag{103}$$

*Proof.* Fix a sample path $\{x_0, y_0, \{W_i\}_{i=1}^{\infty}\}$,

$$LT < \infty, \qquad \qquad (L \text{ and } T \text{ are constants})$$

$$\sup_{c \geq 1} \|h_c(0)\| T < \infty, \qquad \qquad (\text{by (92)})$$

$$\sup_{c \geq 1} \sup_n \sup_{0 \leq t_1 \leq t_2 \leq T_{n+1}-T_n} \left\| \sum_{i=m(T_n+t_1)}^{m(T_n+t_2)-1} \alpha(i) H_c(0, W_{i+1}) \right\| < \infty, \qquad (\text{by (90)})$$

$$\sup_n \sup_{0 \leq t_1 \leq t_2 \leq T_{n+1}-T_n} \sum_{i=m(T_n+t_1)}^{m(T_n+t_2)-1} \alpha(i) L(W_{i+1}) < \infty. \qquad (\text{by (95)})$$

Thus, there exists a constant $C_H$ such that

$$LT \leq C_H$$

$$\sup_{c \geq 1} \|h_c(0)\| \leq \frac{C_H}{T},$$

$$\sup_{c \geq 1} \sup_n \sup_{0 \leq t_1 \leq t_2 \leq T_{n+1}-T_n} \left\| \sum_{i=m(T_n+t_1)}^{m(T_n+t_2)-1} \alpha(i) H_c(0, W_{i+1}) \right\| \leq C_H,$$

$$\sup_n \sup_{0 \leq t_1 \leq t_2 \leq T_{n+1}-T_n} \sum_{i=m(T_n+t_1)}^{m(T_n+t_2)-1} \alpha(i) L(W_{i+1}) \leq C_H.$$

$\square$

We show that the scaled iterates are bounded in the fast timescale.

**Lemma H.6.** $\sup_{n, t \in [0, T+1)} \|\tilde{z}_n(t)\| \leq C_{\hat{z}}$.

*Proof.* $\forall n \in \mathbb{N}, t \in [0, T+1)$,

$$
\|\tilde{z}_n(t)\|
$$

$$
= \left\| \tilde{z}_n(0) + \sum_{i=m(T_n)}^{m(T_n+t)-1} (\alpha(i)H_{r_n}(\tilde{z}_n(t(i)-T_n), W_{i+1}), \beta(i)G_{r_n}(\tilde{z}_n(t(i)-T_n), W_{i+1})) \right\|
$$

$$
\leq \|\tilde{z}_n(0)\| + \left\| \sum_{i=m(T_n)}^{m(T_n+t)-1} \alpha(i) \left[ H_{r_n}(\tilde{z}_n(t(i)-T_n), W_{i+1}) - H_{r_n}(0, W_{i+1}) \right] + \sum_{i=m(T_n)}^{m(T_n+t)-1} \alpha(i)H_{r_n}(0, W_{i+1}) \right\|
$$

$$
+ \left\| \sum_{i=m(T_n)}^{m(T_n+t)-1} \beta(i) \left[ G_{r_n}(\tilde{z}_n(t(i)-T_n), W_{i+1}) - G_{r_n}(0, W_{i+1}) \right] + \sum_{i=m(T_n)}^{m(T_n+t)-1} \beta(i)G_{r_n}(0, W_{i+1}) \right\|
$$

$$
\leq \|\tilde{z}_n(0)\| + \sum_{i=m(T_n)}^{m(T_n+t)-1} \alpha(i)\|H_{r_n}(\tilde{z}_n(t(i)-T_n), W_{i+1}) - H_{r_n}(0, W_{i+1})\| + C_H
$$

$$
+ \sum_{i=m(T_n)}^{m(T_n+t)-1} \beta(i)\|G_{r_n}(\tilde{z}_n(t(i)-T_n), W_{i+1}) - G_{r_n}(0, W_{i+1})\| + C_G
$$

$$
\leq \|\tilde{z}_n(0)\| + \sum_{i=m(T_n)}^{m(T_n+t)-1} (\alpha(i) + \beta(i))L(W_{i+1})\|\tilde{z}_n(t(i)-T_n)\| + C' \qquad \text{(by (101))}
$$

$$
\leq 1 + \sum_{i=m(T_n)}^{m(T_n+t)-1} (\alpha(i) + \beta(i))L(W_{i+1})\|\tilde{z}_n(t(i)-T_n)\| + C' \qquad \text{(by (19))}
$$

$$
\leq [1 + C']\, e^{\sum_{i=m(T_n)}^{m(T_n+t)-1}(\alpha(i)+\beta(i))L(W_{i+1})}
$$

$$
\text{(by } \tilde{z}_n(t) = \tilde{z}_n(t(m(T_n+t)) - T_n) \text{ and discrete Gronwall inequality in Theorem A.3)}
$$

$$
\leq [1 + C']\, e^{C'} \qquad \text{(by (102))}
$$

$$
= C_{\hat{z}}. \qquad \text{(by (103))}
$$

$\square$

We show that the trajectories of these ODEs are bounded in the fast timescale.

**Lemma H.7.** $\sup_{n,t\in[0,T+1)} \|z_n(t)\| \leq C_{\hat{x}}$.

*Proof.* $\forall n, t \in [0, T+1)$,

$$\|z_n(t)\|$$
$$=\left\|z_n(0) + \int_0^t (h_{r_n}(z_n(s)), 0)ds\right\|$$
$$\leq \|z_n(0)\| + \left\|\int_0^t h_{r_n}(z_n(s))ds\right\|$$
$$\leq \|z_n(0)\| + \int_0^t \|h_{r_n}(z_n(s)) - h_{r_n}(0)\|ds + \int_0^t \|h_{r_n}(0)\|ds$$
$$\leq \|z_n(0)\| + \int_0^t L\|z_n(s)\|ds + \int_0^t \|h_{r_n}(0)\|ds \qquad \text{(by Lemma H.2)}$$
$$\leq \|z_n(0)\| + \int_0^t L\|z_n(s)\|ds + (T+1)\|h_{r_n}(0)\|$$
$$\leq \|z_n(0)\| + \int_0^t L\|z_n(s)\|ds + (T+1)\frac{C_H}{T+1} \qquad \text{(by (100))}$$
$$\leq 1 + \int_0^t L\|z_n(s)\|ds + C_H \qquad \text{(by (19), (20))}$$
$$\leq [1 + C_H]\, e^{L(T+1)} \qquad \text{(by Gronwall inequality in Theorem A.1)}$$
$$\leq [1 + C_H]\, e^{C_H} \qquad \text{(by (99))}$$
$$= C_{\hat{x}} \qquad \text{(by (103))}$$

$\square$

We show that applying $h_{r_n}$ to the ODE trajectories still is bounded.

**Lemma H.8.** $\sup_{n, t \in [0, T+1)} \|h_{r_n}(z_n(t))\| < \infty$.

*Proof.* $\forall n, \forall t \in [0, T+1)$,

$$\|h_{r_n}(z_n(t))\|$$
$$\leq \|h_{r_n}(z_n(t)) - h_{r_n}(0)\| + \|h_{r_n}(0)\|$$
$$\leq L\|z_n(t)\| + \|h_{r_n}(0)\| \qquad \text{(by Lemma H.2)}$$
$$\leq LC_{\hat{x}} + \|h_{r_n}(0)\| \qquad \text{(by Lemma H.7)}$$
$$\leq LC_{\hat{x}} + \frac{C_H}{T}. \qquad \text{(by (18) and (100))}$$

Thus, because $C_{\hat{x}}, C_H$ are independent of $n, t$, $\sup_{n, t \in [0, T+1)} \|h_{r_n}(z_n(t))\| < \infty$.

$\square$

In the fast timescale, we show that the limiting trajectory is bounded.

**Lemma H.9.** $\sup_{t \in [0, T+1)} \left\|z^{\lim}(t)\right\| \leq C_{\hat{x}}$.

*Proof.* $\forall t \in [0, T+1)$,

$$
\left\| z^{\lim}(t) \right\|
$$

$$
= \left\| z^{\lim}(0) + \int_0^t (h_\infty(z^{\lim}(s)), 0) ds \right\|
$$

$$
\leq \left\| z^{\lim}(0) \right\| + \left\| \int_0^t h_\infty(z^{\lim}(s)) ds \right\|
$$

$$
= \left\| z^{\lim}(0) \right\| + \left\| \int_0^t \left[ h_\infty(z^{\lim}(s)) - h_\infty(0) \right] ds + \int_0^t h_\infty(0) ds \right\|
$$

$$
\leq \left\| z^{\lim}(0) \right\| + \int_0^t \left\| h_\infty(z^{\lim}(s)) - h_\infty(0) \right\| ds + \int_0^t \left\| h_\infty(0) \right\| ds
$$

$$
\leq \left\| z^{\lim}(0) \right\| + \int_0^t L \left\| z^{\lim}(s) \right\| ds + \int_0^t \left\| h_\infty(0) \right\| ds \qquad \text{(by Lemma H.2)}
$$

$$
\leq 1 + \int_0^t L \left\| z^{\lim}(s) \right\| ds + \int_0^t \left\| h_\infty(0) \right\| ds \qquad \text{(by (19), (20))}
$$

$$
\leq 1 + \int_0^t L \left\| z^{\lim}(s) \right\| ds + (T+1) \left\| h_\infty(0) \right\|
$$

$$
\leq 1 + \int_0^t L \left\| z^{\lim}(s) \right\| ds + C_H \qquad \text{(by Assumption B.6 and (100))}
$$

$$
\leq [1 + C_H] e^{\int_0^t L ds} \qquad \text{(by Gronwall inequality in Theorem A.1)}
$$

$$
\leq [1 + C_H] e^{L(T+1)}
$$

$$
\leq C_{\hat{x}}. \qquad \text{(by (99), (103))}
$$

$\square$

**Lemma H.10.** $\lim_{k \to \infty} h_{r_{n_k}}(z^{\lim}(t)) = h_\infty(z^{\lim}(t))$ *uniformly in* $t \in [0, T+1)$.

*Proof.* By Assumption B.6, $\lim_{k \to \infty} h_{r_{n_k}}(v) = h_\infty(v)$ uniformly in a compact set $\{v | v \in \mathbb{R}^d, \|v\| \leq C_x\}$. By Lemma H.9, $\{z^{\lim}(t) | t \in [0, T+1)\} \subseteq \{v | v \in \mathbb{R}^d, \|v\| \leq C_x\}$. Therefore, $\lim_{k \to \infty} h_{r_{n_k}}(z^{\lim}(t)) = h_\infty(z^{\lim}(t))$ uniformly in $\{z^{\lim}(t) | t \in [0, T+1)\}$ and in $t \in [0, T+1)$. $\square$

**Lemma H.11.** $\forall t \in [0, T+1)$, *we have*

$$
\lim_{k \to \infty} z_{n_k}(t) = z^{\lim}(t).
$$

*Moreover, the convergence is uniform in t on* $[0, T+1)$.

*Proof.* By (22), $\forall \delta > 0$, there exists a $k_1$ such that $\forall k \geq k_1$, $\forall t \in [0, T+1)$,

$$
\left\| \tilde{z}_{n_k}(t) - \tilde{z}^{\lim}(t) \right\| \leq \delta. \tag{104}
$$

By Lemma H.10, there exists a $k_2$ such that $\forall k \geq k_2$, $\forall t \in [0, T+1)$,

$$
\left\| h_{r_{n_k}}(z^{\lim}(t)) - h_\infty(z^{\lim}(t)) \right\| \leq \delta. \tag{105}
$$

$\forall k \geq \max\{k_1, k_2\}, \forall t \in [0, T+1)$

$$
\left\| z_{n_k}(t) - z^{\lim}(t) \right\|
$$

$$
= \left\| \tilde{z}_{n_k}(0) + \int_0^t (h_{r_{n_k}}(z_{n_k}(s)), 0) ds - \tilde{z}^{\lim}(0) - \int_0^t (h_\infty(z^{\lim}(s)), 0) ds \right\|
$$

$$
\leq \left\| \tilde{z}_{n_k}(0) - \tilde{z}^{\lim}(0) \right\| + \left\| \int_0^t h_{r_{n_k}}(z_{n_k}(s)) ds - \int_0^t h_\infty(z^{\lim}(s)) ds \right\|
$$

$$
\leq \delta + \left\| \int_0^t h_{r_{n_k}}(z_{n_k}(s)) - h_\infty(z^{\lim}(s)) ds \right\| \qquad \text{(by (104))}
$$

$$
\leq \delta + \int_0^t \left\| h_{r_{n_k}}(z_{n_k}(s)) - h_{r_{n_k}}(z^{\lim}(s)) \right\| ds + \int_0^t \left\| h_{r_{n_k}}(z^{\lim}(s)) - h_\infty(z^{\lim}(s)) \right\| ds
$$

$$
\leq \delta + L \int_0^t \left\| z_{n_k}(s) - z^{\lim}(s) \right\| ds + \int_0^t \left\| h_{r_{n_k}}(z^{\lim}(s)) - h_\infty(z^{\lim}(s)) \right\| ds \qquad \text{(by Lemma H.2)}
$$

$$
\leq \delta + t\delta + L \int_0^t \left\| z_{n_k}(s) - z^{\lim}(s) \right\| ds \qquad \text{(by (105))}
$$

$$
\leq (\delta + t\delta) e^{Lt} \qquad \text{(by Gronwall inequality in Theorem A.1)}
$$

$$
\leq (\delta + (T+1)\delta) e^{L(T+1)},
$$

which completes the proof. $\qquad\qquad\square$

**Lemma H.12.** *For any function $f : \mathbb{R} \times \mathbb{R} \to \mathbb{R}$, if $\lim_{\substack{a \to \infty \\ b \to \infty}} f(a, b) = L$ then $\lim_{c \to \infty} f(c, c) = L$ where $L$ is a constant.*

*Proof.* By definition, $\forall \epsilon > 0, \exists a_0, b_0$ such that $\forall a > a_0, b > b_0, \|f(a, b) - L\| < \epsilon$. Thus, $\forall \epsilon > 0, \exists c_0 = \max\{a_0, b_0\}$ such that $\forall c > c_0, \|f(c, c) - L\| < \epsilon$. $\qquad\square$

Here we are pulling the limit into the integral.

**Lemma H.13.** $\forall t \in [0, T+1)$,

$$
\lim_{k \to \infty} \int_0^t h_{r_{n_k}}(\tilde{z}^{\lim}(s)) ds = \int_0^t h_\infty(\tilde{z}^{\lim}(s)) ds.
$$

*Proof.* From Lemma H.6, it is easy to see that

$$
\sup_{t \in [0, T+1)} \left\| \tilde{z}^{\lim}(t) \right\| < \infty,
$$

which, similar to Lemma H.8, implies that

$$
\sup_{k, t \in [0, T+1)} \left\| h_{r_{n_k}} \left( \tilde{z}^{\lim}(t) \right) \right\| < \infty.
$$

By the dominated convergence theorem, $\forall t \in [0, T+1)$,

$$
\lim_{k \to \infty} \int_0^t h_{r_{n_k}}(\tilde{z}^{\lim}(s)) ds = \int_0^t \lim_{k \to \infty} h_{r_{n_k}}(\tilde{z}^{\lim}(s)) ds = \int_0^t h_\infty(\tilde{z}^{\lim}(s)) ds,
$$

which completes the proof. $\qquad\square$

Finally, we put all the previous lemmas together and get convergence.

**Lemma H.14.** $\forall t \in [0, T+1)$,

$$
\lim_{k \to \infty} \int_0^t h_{r_{n_k}}(z_{n_k}(s)) ds = \int_0^t h_\infty(z^{\lim}(s)) ds.
$$

*Proof.* $\forall \epsilon > 0$, by Lemma H.10, $\exists k_0$ such that $\forall k \geq k_0, \forall t \in [0, T)$,

$$\left\| h_{r_{n_k}}(z^{\lim}(s)) - h_\infty(z^{\lim}(s)) \right\| \leq \epsilon. \tag{106}$$

By Lemma H.11, $\exists k_1$ such that $\forall k \geq k_1, \forall t \in [0, T+1)$,

$$\left\| z_{n_k}(t) - z^{\lim}(t) \right\| \leq \epsilon. \tag{107}$$

Thus, $\forall k \geq \max\{k_0, k_1\}, \forall t \in [0, T+1)$,

$$
\begin{aligned}
&\left\| \int_0^t h_{r_{n_k}}(z_{n_k}(s))ds - \int_0^t h_\infty(z^{\lim}(s))ds \right\| \\
&\leq \left\| \int_0^t h_{r_{n_k}}(z_{n_k}(s))ds - \int_0^t h_{r_{n_k}}(z^{\lim}(s))ds \right\| + \left\| \int_0^t h_{r_{n_k}}(z^{\lim}(s))ds - \int_0^t h_\infty(z^{\lim}(s))ds \right\| \\
&\leq \int_0^t \left\| h_{r_{n_k}}(z_{n_k}(s)) - h_{r_{n_k}}(z^{\lim}(s)) \right\| ds + \int_0^t \left\| h_{r_{n_k}}(z^{\lim}(s)) - h_\infty(z^{\lim}(s)) \right\| ds \\
&\leq \int_0^t \left\| h_{r_{n_k}}(z_{n_k}(s)) - h_{r_{n_k}}(z^{\lim}(s)) \right\| ds + (T+1)\epsilon && \text{(by (106))} \\
&\leq \int_0^t L \left\| z_{n_k}(s) - z^{\lim}(s) \right\| ds + T\epsilon && \text{(by Lemma H.2)} \\
&\leq L(T+1)\epsilon + (T+1)\epsilon. && \text{(by (107))}
\end{aligned}
$$

Thus, $\forall t \in [0, T+1)$,

$$\lim_{k \to \infty} \int_0^t h_{r_{n_k}}(z_{n_k}(s))ds = \int_0^t h_\infty(z^{\lim}(s))ds.$$

$\square$

This lemma uses the Gronwall inequality to show that in the fast timescale, the iterates can't grow too much in one period $T$.

**Lemma H.15.** $\forall n, \forall t \in [0, T_{n+1} - T_n]$,

$$\|\bar{z}(T_n + t)\| \leq \left( \|\bar{z}(T_n)\|C' + C' \right) e^{C'} + \|\bar{z}(T_n)\|,$$

*and in particular,*

$$\|\bar{z}(T_{n+1})\| \leq \left( \|\bar{z}(T_n)\|C' + C' \right) e^{C'} + \|\bar{z}(T_n)\|$$

*where $C'$ is a positive constant defined in Lemma H.5.*

*Proof.* We first show the difference between $\bar{z}(T_{n+1})$ and $\bar{z}(T_n)$ by the following derivations. $\forall n, \forall t \in [0, T_{n+1} - T_n]$,

$$
\begin{aligned}
&\|\bar{z}(T_n + t) - \bar{z}(T_n)\| \\
=&\|\bar{z}(t(m(T_n + t))) - \bar{z}(T_n)\| \\
=&\left\|\bar{z}(T_n) + \sum_{i=m(T_n)}^{m(T_n+t)-1} (\alpha(i)H(\bar{z}(t(i)), W_{i+1}), \beta(i)G(\bar{z}(t(i)), W_{i+1})) - \bar{z}(T_n)\right\| \\
=&\left\|\sum_{i=m(T_n)}^{m(T_n+t)-1} (\alpha(i)H(\bar{z}(t(i)), W_{i+1}), \beta(i)G(\bar{z}(t(i)), W_{i+1}))\right\| \\
\leq& \sum_{i=m(T_n)}^{m(T_n+t)-1} \alpha(i)\|H(\bar{z}(t(i)), W_{i+1}) - H(\bar{z}(T_n), W_{i+1})\| + \beta(i)\|G(\bar{z}(t(i)), W_{i+1}) - G(\bar{z}(T_n), W_{i+1})\| \\
&+ \left\|\sum_{i=m(T_n)}^{m(T_n+t)-1} \alpha(i)H(\bar{z}(T_n), W_{i+1})\right\| + \left\|\sum_{i=m(T_n)}^{m(T_n+t)-1} \beta(i)G(\bar{z}(T_n), W_{i+1})\right\| \\
\leq& \sum_{i=m(T_n)}^{m(T_n+t)-1} (\alpha(i) + \beta(i))L(W_{i+1})\|\bar{z}(t(i)) - \bar{z}(T_n)\| \\
&+ \left\|\sum_{i=m(T_n)}^{m(T_n+t)-1} \alpha(i)H(\bar{z}(T_n), W_{i+1})\right\| + \left\|\sum_{i=m(T_n)}^{m(T_n+t)-1} \beta(i)G(\bar{z}(T_n), W_{i+1})\right\| \\
\leq& \sum_{i=m(T_n)}^{m(T_n+t)-1} (\alpha(i) + \beta(i))L(W_{i+1})\|\bar{z}(t(i)) - \bar{z}(T_n)\| + \sum_{i=m(T_n)}^{m(T_n+t)-1} (\alpha(i) + \beta(i))L(W_{i+1})\|\bar{z}(T_n)\| \\
&+ \left\|\sum_{i=m(T_n)}^{m(T_n+t)-1} \alpha(i)H(0, W_{i+1})\right\| + \left\|\sum_{i=m(T_n)}^{m(T_n+t)-1} \beta(i)G(0, W_{i+1})\right\| \qquad \text{(by Assumption B.5)} \\
=& \sum_{i=m(T_n)}^{m(T_n+t)-1} (\alpha(i) + \beta(i))L(W_{i+1})\|\bar{z}(t(i)) - \bar{z}(T_n)\| + \|\bar{z}(T_n)\| \sum_{i=m(T_n)}^{m(T_n+t)-1} (\alpha(i) + \beta(i))L(W_{i+1}) \\
&+ \left\|\sum_{i=m(T_n)}^{m(T_n+t)-1} \alpha(i)H(0, W_{i+1})\right\| + \left\|\sum_{i=m(T_n)}^{m(T_n+t)-1} \beta(i)G(0, W_{i+1})\right\| \\
\leq& \sum_{i=m(T_n)}^{m(T_n+t)-1} (\alpha(i) + \beta(i))L(W_{i+1})\|\bar{z}(t(i)) - \bar{z}(T_n)\| + \|\bar{z}(T_n)\|C' + \left\|\sum_{i=m(T_n)}^{m(T_n+t)-1} \alpha(i)H(0, W_{i+1})\right\| \\
&+ \left\|\sum_{i=m(T_n)}^{m(T_n+t)-1} \beta(i)G(0, W_{i+1})\right\| \qquad \text{(by (102))} \\
\leq& \sum_{i=m(T_n)}^{m(T_n+t)-1} (\alpha(i) + \beta(i))L(W_{i+1})\|\bar{z}(t(i)) - \bar{z}(T_n)\| + [\|\bar{z}(T_n)\|C' + C'] \qquad \text{(by (101))} \\
\leq& [\|\bar{z}(T_n)\|C' + C']\, e^{\sum_{i=m(T_n)}^{m(T_n+t)-1} (\alpha(i)+\beta(i))L(W_{i+1})} \qquad \text{(by discrete Gronwall inequality in Theorem A.3)} \\
\leq& [\|\bar{z}(T_n)\|C' + C']\, e^{C'} \qquad \text{(by (102))}
\end{aligned}
$$

$\square$

The following lemmas are in the slow timescale.

The next lemma shows the boundedness of $\|y^{\lim}(t)\|$ where $C, C_0$ are arbitrary constants.

**Lemma H.16.** $\sup_{t \in [0, T+1)} \|y^{\lim}(t)\| \leq C$.

*Proof.* $\forall t \in [0, T+1)$,

$$\|y^{\lim}(t)\|$$
$$= \left\|y^{\lim}(0) + \int_0^t g_\infty(\lambda_\infty(y^{\lim}(s)), y^{\lim}(s))ds\right\|$$
$$\leq \|y^{\lim}(0)\| + \left\|\int_0^t g_\infty(\lambda_\infty(y^{\lim}(s)), y^{\lim}(s))ds\right\|$$
$$= \|y^{\lim}(0)\| + \left\|\int_0^t \left[g_\infty(\lambda_\infty(y^{\lim}(s)), y^{\lim}(s)) - g_\infty(0)\right]ds + \int_0^t g_\infty(0)ds\right\|$$
$$\leq \|y^{\lim}(0)\| + \int_0^t \|g_\infty(\lambda_\infty(y^{\lim}(s)), y^{\lim}(s)) - g_\infty(0)\|ds + \int_0^t \|g_\infty(0)\|ds$$
$$\leq \|y^{\lim}(0)\| + \int_0^t L\|\lambda_\infty(y^{\lim}(s))\| + L\|y^{\lim}(s)\|ds + \int_0^t \|g_\infty(0)\|ds \qquad \text{(by Lemma H.2)}$$
$$\leq 1 + \int_0^t L \cdot L_\lambda \|y^{\lim}(s)\| + L\|y^{\lim}(s)\|ds + \int_0^t \|g_\infty(0)\|ds$$
$$\leq 1 + \int_0^t L(L_\lambda + 1)\|y^{\lim}(s)\|ds + (T+1)\|g_\infty(0)\|$$
$$\leq 1 + \int_0^t L(L_\lambda + 1)\|y^{\lim}(s)\|ds + C_0$$
$$\leq [1 + C_0] e^{\int_0^t L(L_\lambda + 1)ds} \qquad \text{(by Gronwall inequality in Theorem A.1)}$$
$$\leq [1 + C_0] e^{L(L_\lambda + 1)(T+1)}$$
$$\leq C.$$

$\square$

The following shows a uniform convergence on the sequence $g_{r_{n_k}}$.

**Lemma H.17.** $\lim_{k \to \infty} g_{r_{n_k}}(\lambda_\infty(y^{\lim}(t)), y^{\lim}(t)) = g_\infty(\lambda_\infty(y^{\lim}(t)), y^{\lim}(t))$ *uniformly in* $t \in [0, T+1)$.

*Proof.* By Assumption B.6, $\lim_{k \to \infty} g_{r_{n_k}}(v) = g_\infty(v)$ uniformly in a compact set $\{v | v \in \mathbb{R}^{d_1 + d_2}, \|v\| \leq C\}$. By Lemma H.16, $\{y^{\lim}(t) | t \in [0, T+1)\} \subseteq \{v | v \in \mathbb{R}^d, \|v\| \leq C\}$. Additionally, $\lambda_\infty$ is Lipschitz, so $\{(\lambda_\infty(y^{\lim}(t)), y^{\lim}(t)) | t \in [0, T+1)\} \subseteq \{v | v \in \mathbb{R}^{d_1 + d_2}, \|v\| \leq C\}$.

Therefore, $\lim_{k \to \infty} g_{r_{n_k}}(\lambda_\infty(y^{\lim}(t)), y^{\lim}(t)) = g_\infty(\lambda_\infty(y^{\lim}(t)), y^{\lim}(t))$ uniformly in $\{y^{\lim}(t) | t \in [0, T+1)\}$ and in $t \in [0, T+1)$. $\square$

The next lemma gives us the uniform convergence of $y_{n_k}$ we desire.

**Lemma H.18.** $\forall t \in [0, T+1)$, *we have*

$$\lim_{k \to \infty} y_{n_k}(t) = y^{\lim}(t).$$

*Moreover, the convergence is uniform in $t$ on $[0, T+1)$.*

*Proof.* By (40), $\forall \delta > 0$, there exists a $k_1$ such that $\forall k \geq k_1$, $\forall t \in [0, T+1)$,

$$\|\tilde{y}_{n_k}(t) - \tilde{y}^{\lim}(t)\| \leq \delta. \tag{108}$$

By Lemma H.17, there exists a $k_2$ such that $\forall k \geq k_2, \forall t \in [0, T+1)$,

$$\left\| g_{r_{n_k}}(\lambda_\infty(y^{\lim}(t)), y^{\lim}(t)) - g_\infty(\lambda_\infty(y^{\lim}(t)), y^{\lim}(t)) \right\| \leq \delta. \tag{109}$$

$\forall k \geq \max\{k_1, k_2\}, \forall t \in [0, T+1)$

$$
\begin{aligned}
&\left\| y_{n_k}(t) - y^{\lim}(t) \right\| \\
&= \left\| \tilde{y}_{n_k}(0) + \int_0^t g_{r_{n_k}}(\lambda_\infty(y_{n_k}(s)), y_{n_k}(s))ds - \tilde{y}^{\lim}(0) - \int_0^t g_\infty(\lambda_\infty(y^{\lim}(s)), y^{\lim}(s))ds \right\| \\
&\leq \left\| \tilde{y}_{n_k}(0) - \tilde{y}^{\lim}(0) \right\| + \left\| \int_0^t g_{r_{n_k}}(\lambda_\infty(y_{n_k}(s)), y_{n_k}(s))ds - \int_0^t g_\infty(\lambda_\infty(y^{\lim}(s)), y^{\lim}(s))ds \right\| \\
&\leq \delta + \left\| \int_0^t g_{r_{n_k}}(\lambda_\infty(y_{n_k}(s)), y_{n_k}(s)) - g_\infty(\lambda_\infty(y^{\lim}(s)), y^{\lim}(s))ds \right\| && \text{(by (108))} \\
&\leq \delta + \int_0^t \left\| g_{r_{n_k}}(\lambda_\infty(y_{n_k}(s)), y_{n_k}(s)) - g_{r_{n_k}}(\lambda_\infty(y^{\lim}(s)), y^{\lim}(s)) \right\| ds \\
&\quad + \int_0^t \left\| g_{r_{n_k}}(\lambda_\infty(y^{\lim}(s)), y^{\lim}(s)) - g_\infty(\lambda_\infty(y^{\lim}(s)), y^{\lim}(s)) \right\| ds \\
&\leq \delta + L \int_0^t \left\| (\lambda_\infty(y_{n_k}(s)), y_{n_k}(s)) - (\lambda_\infty(y^{\lim}(s)), y^{\lim}(s)) \right\| ds \\
&\quad + \int_0^t \left\| g_{r_{n_k}}(\lambda_\infty(y^{\lim}(s)), y^{\lim}(s)) - g_\infty(\lambda_\infty(y^{\lim}(s)), y^{\lim}(s)) \right\| && \text{(by Lemma H.2)} \\
&\leq \delta + t\delta + L(L_\lambda + 1) \int_0^t \left\| y_{n_k}(s) - y^{\lim}(s) \right\| ds && \text{(by (109))} \\
&\leq (\delta + t\delta)e^{L(L_\lambda + 1)t} && \text{(by Gronwall inequality in Theorem A.1)} \\
&\leq (\delta + T\delta)e^{L(L_\lambda + 1)T},
\end{aligned}
$$

which completes the proof. $\qquad\square$

The next lemma applies the Gronwall inequality to control the growth of the iterates in the slow timescale.

**Lemma H.19.** *There are some constants $A, B$ such that*

$$\|y_l^{max}\| \leq A \left\| y_{m(T_n)}^{max} \right\| + B$$

*where $m(T_n) \leq l \leq m(T_{n+1})$. As a consequence, there are constants $C$ and $D$ such that*

$$\|z_l^{max}\| \leq C \left\| z_{m(T_n)}^{max} \right\| + D. \tag{110}$$

*This implies that $\|\tilde{z}_n(t)\|$ is bounded since for all $n$, $\|\tilde{z}_n(0)\| \leq 1$.*

*Proof.* We know that

$$
\begin{aligned}
\|y_l\| &= \left\| y_{m(T_n)} + \sum_{i=m(T_n)}^l \beta(i) G(x_i, y_i, W_{i+1}) \right\| \\
&\leq \|y_n^{max}\| + \sum_{i=m(T_n)}^l \beta(i) \|G(x_i, y_i, W_{i+1})\|
\end{aligned} \tag{111}
$$

Since the second term of (111) is monotonically increasing in $l$, we know that the following holds:

$$\|y_l^{\max}\| \leq \left\|y_{m(T_n)}^{\max}\right\| + \sum_{i=m(T_n)}^{l} \beta(i)\|G(x_i, y_i, W_{i+1})\|$$

$$\leq \left\|y_{m(T_n)}^{\max}\right\| + \sum_{i=m(T_n)}^{l} \beta(i)L(W_{i+1})(\|x_i\| + \|y_i\|) + \sum_{i=m(T_n)}^{l} \beta(i)G(0, W_{i+1})$$

$$\leq \left\|y_{m(T_n)}^{\max}\right\| + \sum_{i=m(T_n)}^{l} \beta(i)L(W_{i+1})(\|x_i\| + \|y_i\|) + E \tag{90}$$

$$\leq \left\|y_{m(T_n)}^{\max}\right\| + \sum_{i=m(T_n)}^{l} \beta(i)L(W_{i+1})(K(1 + \|y_i^{\max}\|) + \|y_i^{\max}\|) + E \tag{Lemma 3.1}$$

$$\leq \left\|y_{m(T_n)}^{\max}\right\| + \sum_{i=m(T_n)}^{l} \beta(i)L(W_{i+1})(K + (K+1)\|y_i^{\max}\|) + E$$

$$\leq \left\|y_{m(T_n)}^{\max}\right\| + K\sum_{i=m(T_n)}^{l} \beta(i)L(W_{i+1}) + (K+1)\sum_{i=m(T_n)}^{l} \beta(i)L(W_{i+1})\|y_i^{\max}\| + E$$

$$\leq \left\|y_{m(T_n)}^{\max}\right\| + KC' + (K+1)\sum_{i=m(T_n)}^{l} \beta(i)L(W_{i+1})\|y_i^{\max}\| + E \tag{102}$$

$$\leq \left(\left\|y_{m(T_n)}^{\max}\right\| + KC'\right)e^{(K+1)\sum_{i=m(T_n)}^{l}\beta(i)L(W_{i+1})} + E \quad \text{(by discrete Grownwall inequality in Appendix A.3)}$$

$$\leq \left(\left\|y_{m(T_n)}^{\max}\right\| + KC'\right)e^{(K+1)C'} + E \tag{102}$$

Now we show (110):

$$\|z_l^{max}\| \leq \|y_l^{\max}\| + K(1 + \|y_l^{\max}\|) \tag{Lemma 3.1}$$

$$\leq K + (K+1)(A\left\|y_{m(T_n)}^{\max}\right\| + B)$$

$$\leq K + (K+1)(A\left\|z_{m(T_n)}^{max}\right\| + B)$$

$$\square$$

