# OpenReview forum: "Convergence of Two-Timescale Markovian Stochastic Approximations with Applications in Reinforcement Learning"
_ICML.cc/2026/Conference — ICML 2026 regular_

### Official Review · Reviewer_4iU3 · 2026-03-09

**Soundness:** 3
**Presentation:** 3
**Significance:** 2
**Originality:** 2
**Overall Recommendation:** 3
**Confidence:** 3

**Summary:**

The paper claims to give the first stability and convergence result for two-timescale schemes with Markovian noise (under general, verifiable conditions, without using projections or requiring "compact" noise), which can be applied to TD learning with gradient correction (TDC).

**Compliance With Llm Reviewing Policy:**

Affirmed.

**Final Justification:**

Since my main concern is about the (a bit unfair) positioning and the authors said they would better frame their contributions with the related works, I raise my score from 2 to 3. It is not higher because: (i) the issue has caused some negative impact on the evaluation, and (ii) I am not sure whether the contribution (after clarification, it has several enhancements added to existing works) reach the acceptance bar.

**Key Questions For Authors:**

They are listed in "Strengths And Weaknesses".

**Limitations:**

They are also listed in "Strengths And Weaknesses".

**Strengths And Weaknesses:**

(Significance and originality.) After reading only the first two sections (Introduction and Related work), I don't feel good about the positioning of the paper, as the authors seem to make their contributions sound more significant than they actually are.

"Together, our results extend SA theory, establishing the first theoretical foundation for analysis of two-timescale algorithms with the realistic noise models inherent to RL." (line 041 left)

It seems that the paper lays a foundation for general two-timescale SA with Markovian noise. However, as the authors surveyed, there are already quite some works in this setting (Karmakar & Bhatnagar, 2018; 2021; Panda & Bhatnagar, 2025). Their issues that the current paper fix are specific features like "the faster timescale does not impact the slower timescale" (line 113 left), "requires the noise to be in a compact space" (line 125 left), and "require a projection on the fast timescale" (line 131 left), just to make the result "general enough to be suitable for our purpose" (line 124 left), which is the TDC application. I think this is a bit far way from the foundation claimed in the abstract (quoted at the beginning).

Moreover, the aforementioned three most related papers don't appear in Table 1, which "situate our work in the SA literature, highlighting the crucial missing pieces: stability and convergence guarantees for two-timescale algorithms with Markovian noise" (line 078 left). The three papers are about the crucial missing pieces. Why are they not in Table 1?

(Soundness and presentation.) For the assumptions, the authors claim "these assumptions are mild, minimal, and well-supported in literature that studies stability of SA" (line 195 left).  They may want to either avoid these strong claims or provide strong evidence/explanations for them. After all, those "minimal" assumptions take more than two full pages in Appendix B...

For the main result Theorem 4.3 and its proof described in Section 7, they are about $\lambda$, the equilibrium of (3), while the assumptions in Appendix B are about $\lambda_\infty$, the equilibrium of the limited ODEs. What is going on? In Section 7, the authors may want to verbally explain how to pass the limit.

Overall, I think the main issue is the positioning. There is contribution in this paper but the authors should fairly describe the contribution. In particular, the issue with Table 1 goes beyond ordinary overstatement of significance: it concerns whether the paper presents the closest prior literature in a fair way, so that readers can accurately judge what is genuinely new.

---

> ### Author Rebuttal · Authors · 2026-03-31
>
> We thank the reviewer for putting the time and sharing their perspective. We respond below:
>
> >(Significance and originality.) After reading only the first two sections (Introduction and Related work), I don't feel good about the positioning of the paper, as the authors seem to make their contributions sound more significant than they actually are…Moreover, the aforementioned three most related papers don't appear in Table 1, which "situate our work in the SA literature, highlighting the crucial missing pieces: stability and convergence guarantees for two-timescale algorithms with Markovian noise" (line 078 left). The three papers are about the crucial missing pieces. Why are they not in Table 1?
>
> We appreciate the reviewer bringing forward this concern. The purpose of the table was to make it clear which setting our paper was meant to address, in comparison to the other foundational stability works. However, we see how that comes across as omitting works in a similar setting to ours. In our current revision, we have added those works (specifically Karmakar 2021 and Panda 2025; Karmakar 2018 assumes stability so is not relevant to the table) to the table to more fairly represent them.
>
> We would like to note that eliminating that the faster timescale does not impact the slower timescale, that the noise must be in a compact space, and that a projection is required on the fast timescale, is significant and, when combined with the already difficult task of addressing the Markovian two-timescale case, represents a significant contribution. The goal of the language of “foundation” was to indicate that our work is general enough to address the Markovian two-timescale case without many of the restrictions previous works have had. Still, we thank the reviewer for indicating that our language was too extreme and we have softened it.
>
> >For the assumptions, the authors claim "these assumptions are mild, minimal, and well-supported in literature that studies stability of SA" (line 195 left). They may want to either avoid these strong claims or provide strong evidence/explanations for them. After all, those "minimal" assumptions take more than two full pages in Appendix B…
>
> We again thank the reviewer for bringing our attention to this concern. We recognize that on its face, the sheer quantity of assumptions may make a statement like that look absurd, but the nature of the SA theory field is such that if we want to be explicit and write out the assumptions with commentary, they will take up a lot of space. We stand by “well-supported in the literature” but understand how “mild, minimal” may seem ridiculous, and in our current revision, have reworded this. The idea we wanted to get across was that our assumptions were in line with prior work and are much easier to verify (as demonstrated in our proof of TDC) than the assumption of stability that many convergence proofs of two-timescale schemes take (which is much shorter to write out but impossible to verify unless one just applies projections, which introduce bias).
>
> >For the main result Theorem 4.3 and its proof described in Section 7, they are about
> , the equilibrium of (3), while the assumptions in Appendix B are about
> , the equilibrium of the limited ODEs. What is going on? In Section 7, the authors may want to verbally explain how to pass the limit.
>
> We thank the reviewer for bringing this potential point of confusion to our attention. The simple fix we are adding is, in assumption B.6, to also define $\lambda$. $\lambda_{\infty}$ is only relevant to establishing stability, and $\lambda$ is only relevant to establishing convergence. In Theorem 4.3, as stated, we use the assumptions to establish stability and then establish convergence, so we need to have both be defined clearly. Since the reviewer brought it up, there is no need to pass any limit, we just need to define both terms. We appreciate the reviewer giving us the chance to address this!
>
> If the reviewer has any other concerns or questions, we are happy to continue discussing!
>
> Works referenced in this response:
>
> [Karmakar’18] Karmakar, P. and Bhatnagar, S. Two time-scale stochastic approximation with controlled markov noise and off-policy temporal-difference learning. Mathematics of Operations Research, 2018.
>
> [Karmakar’21] Karmakar, P. and Bhatnagar, S. Stochastic approximation with iterate-dependent markov noise under verifiable conditions in compact state space with the stability of iterates
> not ensured. IEEE Transactions on Automatic Control, 2021.
>
> [Panda’25] Panda, P. and Bhatnagar, S. Two-timescale critic-actor for
> average reward mdps with function approximation. AAAI, 2025.

---

> > ### Author Rebuttal · Reviewer_4iU3 · 2026-04-02
> >
> > Thank you for the response that clears things up. As the authors will improve the positioning, I will improve my score.

---

> > > ### Author Response · Authors · 2026-04-06
> > >
> > > We sincerely thank the reviewer for their time, the constructive discussion throughout this rebuttal phase, and for increasing the score!
> > > We are very glad that our responses and clarifications have fully addressed your initial concerns. We are committed to incorporating all of these clarifications into the final revision.
> > > As this discussion period comes to a close, we hope that the resolution of these issues and our planned improvements give you full confidence in championing the paper for acceptance. Thank you again for your valuable feedback, which has genuinely helped strengthen our work!

---

### Official Review · Reviewer_2WLr · 2026-03-11

**Soundness:** 3
**Presentation:** 2
**Significance:** 2
**Originality:** 2
**Overall Recommendation:** 4
**Confidence:** 3

**Summary:**

This paper studies convergence of two-timescale stochastic approximation (SA) under Markovian noise, motivated by reinforcement learning settings where samples are generated by a Markov process.

The authors establish stability and almost sure convergence for a general class of coupled two-timescale recursions without requiring projections or compact noise assumptions.

The key technical step (they claimed) is Lemma 4.1, which connects the two timescales by showing that the fast iterates can be bounded by the maximal slow iterate seen previously. Using this idea, the authors derive general stability and convergence results and apply the framework to prove convergence of TDC with eligibility traces.

**Compliance With Llm Reviewing Policy:**

Affirmed.

**Final Justification:**

We thank the reviewer for the thoughtful follow-up and for acknowledging that all concerns will be addressed if the author indeed does what they plan. So I raise the point.

**Key Questions For Authors:**

See each point in Weaknesses.

**Limitations:**

No. The paper briefly mentions societal impact but does not clearly discuss the limitations of the theoretical framework. It would be helpful for the authors to comment on the scope of their assumptions, the asymptotic nature of the results, and the lack of empirical validation.

**Strengths And Weaknesses:**

## Strengths

1. The paper studies an important theoretical problem: convergence of two-timescale stochastic approximation with Markovian noise, which is relevant to reinforcement learning.

2. The framework is general and aims to remove several restrictive assumptions used in previous analyses.

3. The paper provides a concrete application to TDC with eligibility traces, showing that the framework can be instantiated for a practical RL algorithm.

------

## Weaknesses

1. **The novelty and usefulness of Lemma 4.1 are not clearly explained.**
The paper claims that the key technical innovation is Lemma 4.1, which bounds the fast iterates by the maximal slow iterate seen previously. However, the paper does not clearly explain why this observation is particularly useful or what technical difficulty in previous analyses it resolves. Section 1.1 provides some intuition, but it is still unclear why this lemma is essential and why prior work did not use a similar idea. It would be helpful if the authors provided a clearer high-level explanation addressing questions such as:

    - What specific technical obstacle in prior two-timescale analyses does Lemma 4.1 overcome?

    - Why does bounding fast iterates via the maximal slow iterate enable stability and convergence proofs that were previously difficult?

From my reading, the overall analysis seems to combine ideas from Liu (2025) with techniques used in Lakshminarayanan & Bhatnagar (2017) for two-timescale stochastic approximation. However, the paper does not clearly delineate what part of the argument is fundamentally new versus what is adapted from prior work, especially how important Lemma 4.1 is in this process. A clearer conceptual explanation would significantly improve the paper.

2. **The proof structure is difficult to follow.**
Section 5 is somewhat hard to follow. Many details are deferred to the appendix, making it difficult to understand the overall proof strategy. A high-level proof roadmap or diagram summarizing the main steps (e.g., stability → boundedness → convergence) would help readers understand the overall argument.

3. **Dependence of constants is unclear.**

Several results introduce constants (e.g., $C_1$ in Lemma 5.7 and $C_2$ in Lemma 5.8), but the paper does not explain how these constants depend on the problem parameters. It is unclear whether these are universal constants or whether they depend on quantities such as Lipschitz constants, step sizes, or properties of the Markov chain. Even if the explicit forms are complicated, the appendix should at least clarify which quantities these constants depend on, as this affects the interpretation of the bounds and the scope of the theoretical guarantees.

4. **No empirical validation.**

The paper is purely theoretical and contains no experiments. While theory-only papers are acceptable, the paper repeatedly motivates the results using reinforcement learning applications. Some empirical illustration would help demonstrate that the assumptions and theory are relevant for practical algorithms.

For example, it would be useful to provide a small experiment on a standard RL benchmark showing the behavior predicted by Lemma 4.1, namely that the fast iterates remain controlled by the magnitude of the slow iterates during training. Such an experiment would not constitute a proof of the lemma, but it could help illustrate the phenomenon in practice and provide evidence that the theoretical framework captures realistic algorithm behavior.

---

> ### Author Rebuttal · Authors · 2026-03-31
>
> We appreciate the reviewer taking time to review our work. We are glad they note that we sought to develop a more general framework for two-timescale SA and apply it to TDC with eligibility traces. We now respond to the questions:
>
> >What specific technical obstacle in prior two-timescale analyses does Lemma 4.1 overcome? Why does bounding fast iterates via the maximal slow iterate enable stability and convergence proofs that were previously difficult?
>
> We appreciate the reviewer spotlighting this important issue. In some way, we need to relate the growth of the fast iterate to the growth of the slow iterate. In most works addressing the stability of two-timescale SA, they bound the fast iterate by the current slow iterate. But even from what was provided in Lakshminarayanan’s proof of this statement in the i.i.d. Case, we were not able to replicate. And in our general markovian scheme without projections and with noncompact noise, such a result was out of reach. Specifically, while there are analytical mechanisms to prevent the fast iterate from growing too large, there isn’t one to prevent the slow iterate from becoming too small. However, the weaker result of bounding fast iterates via the maximal slow iterate is relatively straightforward. This forces adaptations and therefore novelties in the slow timescale analysis that Lakshminarayanan does not have to do. However, our Lemma 4.1 enabled this alternate path that worked out in the end to show overall stability.
>
> >The proof structure is difficult to follow…A high-level proof roadmap or diagram…would help readers understand the overall argument.
>
> We appreciate the opportunity to improve our presentation. The suggestion to include a diagram to help visualize the methodology is a great one that we will try to incorporate to clarify our proof structure.
>
> >Several results introduce constants (e.g., in Lemma 5.7 and in Lemma 5.8), but the paper does not explain how these constants depend on the problem parameters. It is unclear whether these are universal constants or whether they depend on quantities such as Lipschitz constants, step sizes, or properties of the Markov chain. Even if the explicit forms are complicated, the appendix should at least clarify which quantities these constants depend on, as this affects the interpretation of the bounds and the scope of the theoretical guarantees.
>
> We thank the reviewer for bringing this to our attention. We want to make the presentation clear. In most cases, (including 5.7 and 5.8) constants are sample path dependent. That is, for a specific sequence generated by the stochastic approximation scheme, there will be a value for each of these constants, and a sample path comes with specific values for the Lipschitz constants, and step sizes.
> The difficulty is that since this is an asymptotic analysis, there are many cases where we will instantiate a variable, e.g. in appendix C.9 we effectively say  “ let $n_0$ such that for all $n > n_0$, the discretization error $f_n(t)$ is less than $\epsilon$ for all $t$,” and later we have $C_1 > \max_{i \leq n_0} \norm{\bar{x}(T_i)}$. If the reviewer is suggesting that in these instances, it would help to explicitly state which variables $C_1$ depends on (as opposed to an expression for C_1 which we believe would be more confusing than illuminating) we certainly will endeavor to do so.
>
> >it would be useful to provide a small experiment on a standard RL benchmark showing the behavior predicted by Lemma 4.1, namely that the fast iterates remain controlled by the magnitude of the slow iterates during training…
>
> We thank the reviewer for the suggestion. We did not do any experiments as the scope of this work is in theoretical contributions to stochastic approximation (many works in this area, e.g. Borkar 2009, Liu et al 2025, do not perform experiments). We would like to point also to this Sutton paper: https://icml.cc/Conferences/2009/papers/546.pdf, where TDC (albeit with no traces) was first presented and has experiments showing that it seems to converge.
>
> However, a specific experiment regarding Lemma 4.1, showing the relationship between the slow and fast iterates is an interesting idea, which we will look into. Thanks for the great suggestion!
>
> If there are any other questions or clarifications the reviewer would like, we are happy to continue discussing!
>
> Works referenced in this response:
>
> [Sutton’09] Sutton, R. S., et al. Fast gradient-descent methods for temporal-difference learning with linear function approximation. ICML, 2009.
>
> [Borkar’09] Borkar, V. Stochastic approximation: a dynamical systems viewpoint. Springer, 2009.
>
> [Lakshminarayanan’17] A stability criterion for two timescale stochastic approximation schemes.
> Automatica, 2017.
>
> [Liu’25] Liu, S., et al. The ode method for stochastic approximation and reinforcement learning with markovian noise. JMLR, 2025.

---

> > ### Author Rebuttal · Reviewer_2WLr · 2026-04-04
> >
> > We thank the reviewer for the thoughtful follow-up and for acknowledging that all concerns will be addressed if the author indeed does what they plan. I respond to some points below.
> >
> > (1) It would still be helpful to explicitly state which variables $C_1$ depends on, rather than leaving this implicit.
> >
> > (2) I understand this is primarily theoretical work, but adding a small empirical illustration would strengthen the paper. Since Lemma 4.1 is central, a simple experiment validating the relationship between fast and slow iterates would make the result more convincing. In particular, asymptotic analysis may hide large constants, and an empirical result would help illustrate their magnitude in practice.

---

> > > ### Author Response · Authors · 2026-04-06
> > >
> > > The authors are grateful for the reviewer’s response and the score improvement! All right, we’re glad we’re on the same page regarding communication of the variables that $C_1$ and friends depend on–we will make this more explicit in the final version. We will try running a simple experiment concerning lemma 4.1 and include it if the result is illuminating. Thanks for all your feedback in this process!

---

### Official Review · Reviewer_hERS · 2026-03-12

**Soundness:** 3
**Presentation:** 2
**Significance:** 2
**Originality:** 2
**Overall Recommendation:** 5
**Confidence:** 3

**Summary:**

The paper studies the asymptotic convergence of two-timescale stochastic approximation (SA) with Markovian sampling scheme using the ODE framework of Borkar and Meyn together with a recent extension to the Markovian setting by Liu. It first establishes boundedness of the iterates by deriving an inequality relating the slow and fast timescale variables--which authors claim to be the novelty of the theoretical result. The authors find its application to TDC with eligibility traces which is a typical two-time scale RL algorithm with unbounded space.

**Compliance With Llm Reviewing Policy:**

Affirmed.

**Final Justification:**

My concerns have been mostly addressed. In particular, the comparison with the result of Lakshminarayanan’s has been well addressed. Overall, I believe the results of the paper would be useful to the community. Therefore, I increase my score to acceptance.

**Key Questions For Authors:**

1. It may be challenging, but could the authors provide an estimate of the magnitude of $C_1C_2C_3$ in Lemam 5.1 is?

2. Why is the choosing the scaling term as in Definition 6.1 is innovation? How does it differ from the choice of Borkar and Meyn?

**Limitations:**

yes

**Strengths And Weaknesses:**

**Strength**

1. The authors establish stability and asymptotic convergence results for two-timescale stochastic approximation under a Markovian sampling scheme--which fills the gap in the literature of insufficient result of asymptotic convergence of two-time sclae Markovian SA. They address the challenging problem of boundedness by building on the framework of Borkar and Meyn. Notably, the analysis does not require a projection step or compactness assumptions on the noise.


2. The lemma (Lemma 4.1), which bounds the fast iterate in terms of the slow iterate, appears to be a new and simple result that is easy to apply. Moreover, even though I did not fully go through the Appendices, the proof looks solid.


**Weakness**

1. There has been extensive research on the non-asymptotic analysis of two-timescale methods, for instance by Kaledin et al. and Zeng et al. Given these results, which provide explicit finite-sample convergence guarantees, it is unclear whether establishing asymptotic convergence for Markovian stochastic approximation alone constitutes a sufficiently strong contribution.

Kaledin, Maxim, et al. "Finite time analysis of linear two-timescale stochastic approximation with Markovian noise." Conference on Learning Theory. PMLR, 2020.

Zeng, Sihan, and Thinh Doan. "Fast two-time-scale stochastic gradient method with applications in reinforcement learning." The Thirty Seventh Annual Conference on Learning Theory. PMLR, 2024.

2. A primary concern is that the novelty of the technical contribution is not sufficiently clear. In particular, it is not evident how the present analysis differs in a substantive way from the proof of Lakshminarayanan and Bhatnagar. Given that the Markovian setting has already been studied, for instance by Liu, one might expect that the argument of Lakshminarayanan and Bhatnagar could be extended to this setting by incorporating Liu’s techniques. From this perspective, it remains unclear what fundamentally new idea is introduced in Lemma 4.1, which is presented as a key innovation.

---

> ### Author Rebuttal · Authors · 2026-03-31
>
> We thank the reviewer for taking the time to examine our work. We appreciate the reviewer noting how our work addresses stability in the two-timescale Markovian without using projections or requiring compactness of noise. We now respond to the reviewer’s questions:
>
> >There has been extensive research on the non-asymptotic analysis of two-timescale methods ...it is unclear whether establishing asymptotic convergence for Markovian stochastic approximation alone constitutes a sufficiently strong contribution.
>
> We thank the reviewer for calling our attention to these works. We agree that finite-sample convergence rates are desirable results. However, it seems to us that these works don’t quite cover the full generality of the problem we are addressing (in particular, it seems that the noise conditions are not general enough to cover TDC with eligibility traces).
>
> We hope that our work could potentially step toward future work on finite sample analysis of our general problem of interest. Additionally, we think that the continued technical development of the ODE method for addressing stability could be of interest in its own right.
>
> >A primary concern is that the novelty of the technical contribution is not sufficiently clear…it remains unclear what fundamentally new idea is introduced in Lemma 4.1, which is presented as a key innovation.
>
> We appreciate the opportunity to clarify our technical contribution. A straightforward incorporation of Liu’s techniques was not sufficient. Liu only showed diminishing discretization error on a subsequence, which is sufficient in the single-timescale case. However, to obtain a diminishing discretization error in the slow timescale, we require Lemma 5.10 to show that the fast iterates track the equilibria of the slow iterates. This, in turn, requires diminishing discretization error across the whole sequence in the fast timescale, precluding us from using the same scaling term $r_n$ in the fast timescale as Liu, which then changed the analysis.
>
> The fundamentally new idea in Lemma 4.1 is in bounding the fast iterate by the max of all previous slow iterates, as opposed to bounding by the current slow iterate (which is a stronger statement). We were not able to replicate Lakshminarayanan’s proof of the stronger statement in the i.i.d. case–but certainly that stronger statement is at least difficult (if not impossible) to show directly in the Markovian case. The weaker statement, Lemma 4.1, is easier to prove, is a unique type of statement that prior works in the two-timescale ODE method literature have not shown, and engenders methodological novelty in the slow timescale analysis.
>
> >Why is the choosing the scaling term as in Definition 6.1 is innovation? How does it differ from the choice of Borkar and Meyn?
>
> We thank the reviewer for the opportunity to clarify some of our methodological novelty of our analysis in the slow timescale. Borkar’s choice is similar to Liu–let $r_n$ be the max of $1$ and $\bar{z}_{T_n}$. This means that the sequence of scaling factors only consider the iterate at $T_n$ and not necessarily be monotonically increasing. In contrast, our slow timescale scaling factor is the max over all iterates $\bar{z}_n$ seen thus far. So, it considers the norm of every single previous iterate, not just the ones appearing at each $T_n$. This results from our choice in Lemma 4.1, to bound the fast iterate by the maximal slow iterate seen so far.
>
> >It may be challenging, but could the authors provide an estimate of the magnitude of
>  in Lemam 5.1 is?
>
> We thank the reviewer for this question. Unfortunately, since our work is a general, asymptotic analysis, we do not currently have an expression of the magnitude of that term in terms of other variables, which would likely include $T$, the period between rescalings. It is worth noting, however, that this Lemma 4.1 is part of a larger proof by contradiction. Eventually, from Theorem 4.2, we have boundedness of the iterates, which would then render Lemma 4.1 trivially true. Getting an estimate for this value relating the magnitude of the slow and fast iterates would likely require a different analytical approach, perhaps more in line with finite-sample analysis.
>
> If there are any new questions or clarifications the reviewer would like, we are happy to discuss more!
>
> Works referenced in this response:
>
> [Sutton’09] Sutton, R. S., et al. Fast gradient-descent methods for temporal-difference learning with linear function approximation. ICML, 2009.
>
> [Borkar’09] Borkar, V. Stochastic approximation: a dynamical systems viewpoint. Springer, 2009.
>
> [Lakshminarayanan’17] A stability criterion for two timescale stochastic approximation schemes.
> Automatica, 2017.
>
> [Liu’25] Liu, S., et al. The ode method for stochastic approximation and reinforcement learning with markovian noise. JMLR, 2025.

---

> > ### Author Rebuttal · Reviewer_hERS · 2026-04-02
> >
> > Thanks the authors for the detailed response. Could the authors clarify why bounding the iterate using the current slow iterate, as in Lakshminarayanan’s proof, constitutes a stronger statement than bounding the fast iterate by the maximum over all previous slow iterates?

---

> > > ### Author Response · Authors · 2026-04-06
> > >
> > > We are happy to provide clarification on this! The maximum over all the slow iterates so far (including the current slow iterate) is greater than or equal to the current slow iterate. So it is a stronger statement to say that the current fast iterate is bounded in some way by the current slow iterate because it implies that the current fast iterate is bounded by the max over all slow iterates up to and including the current one.
> > >
> > > We see how our earlier response may have been misleading, since we stated “The fundamentally new idea in Lemma 4.1 is in bounding the fast iterate by the max of all previous slow iterates” when really we meant “...bounding the fast iterate by the max over all the slow iterates previous to and including the current slow iterate.”
> > >
> > > However, just in case, here is a concrete example: suppose our first few iterates–formatted as (fast iterate, slow iterate)--are (10, 15), (20, 30), (21, 10). Here, we have that the fast iterates are bounded by the maximum of all the slow iterates up to and including the current slow iterate, but we do not have that the fast iterates are bounded by the current slow iterate, in the last pair (21, 10).
> > >
> > > We hope this clarification was helpful!

---

### Official Review · Reviewer_fmwn · 2026-03-14

**Soundness:** 3
**Presentation:** 3
**Significance:** 4
**Originality:** 3
**Overall Recommendation:** 5
**Confidence:** 4

**Summary:**

This paper develops a general asymptotic theory for two-timescale stochastic approximation with Markovian noise, focusing on the hard step of proving stability without projections; it then uses that stability result to establish convergence in general settings, including unbounded and uncountable spaces. As its main RL application, it applies the theory to prove convergence of TDC with eligibility traces, framing this as a representative off-policy prediction algorithm with linear function approximation.

**Compliance With Llm Reviewing Policy:**

Affirmed.

**Final Justification:**

overall I support this paper.

**Key Questions For Authors:**

- Could you elaborate on Assumption B.6? I feel assuming \lambda_{\infty} being homogeneous is too strong. What kind of problems satisfy this assumption?
- What settings satisfy Assumption B.4? It seems that only linear setting satisfies this assumption. Can you elaborate?

**Limitations:**

The assumptions B.4 and B.6 are restrictive, and besides TDC, it is not clear if they are valid in any other setting.

**Strengths And Weaknesses:**

Soundness: The statements and the results seem to be sound, although I did not check all the proofs.

Presentation: The assumptions in the paper are written simply in the middle of the paragraph, without any proper labeling. The authors should have assumptions written in a more formal way. I acknowledge that the assumptions are written in appendix B, but they have to be present in the main paper.

Significance: The result is interesting since this is the first work that establishes the asymptotic convergence of TDC with eligibility traces without the need for any projection. However, it is worth mentioning that the community nowadays is mostly interested in finite time convergence rather than asymptotic convergence. That said, I think it is a good paper.

Originality: The paper provide an insight into finite time analysis of TDC with eligibility traces.

---

> ### Author Rebuttal · Authors · 2026-03-31
>
> We thank the reviewer for taking the time to examine our work and provide thorough feedback. We especially appreciate the reviewer recognizing the contribution our work makes in proving stability without projection and addressing convergence in general settings. We now respond to the reviewer’s questions:
>
> >The assumptions in the paper are written simply in the middle of the paragraph, without any proper labeling. The authors should have assumptions written in a more formal way. I acknowledge that the assumptions are written in appendix B, but they have to be present in the main paper.
>
> We thank the reviewer for pointing out this issue with the presentation. We chose to present the assumptions this way to save space in the main text and to avoid throwing too many technical details at the reader in the main text. However, in our newest revision, we are trying to incorporate this suggestion by more explicitly labeling the assumptions in the main text, while the full details will remain in the appendix.
>
> >Could you elaborate on Assumption B.6? I feel assuming \lambda_{\infty} being homogeneous is too strong. What kind of problems satisfy this assumption?
>
> We appreciate the reviewer bringing up this point regarding the homogeneity. There are two places where it is used: one place is in the proof of Lemma 5.10 (in appendix C.14), to help show that the fast iterates will be close to the $\lambda_{\infty}$ of the slow iterates, and the other is in the proof of Lemma 6.3 (in appendix D.8), to help in the proof of diminishing discretization error in the slow timescale. In our current revision, we are now noting the application of homogeneity in each instance.
>
> Since we only use homogeneity in those two instances, our hope is that perhaps it may not be too difficult to accomplish those without using it, letting us relax homogeneity to just Lipschitzness. One approach may be to define a scaled $\lambda_c$ similar to $H_c$.
>
> Since the assumption holds for TDC, we took it since it made the analysis simpler. The assumption does hold for most temporal difference (TD) algorithms with linear function approximation since the solution is linear (see also Sutton et al. 2009, Yu 2017, Liu et al. 2025)–$\lambda_{\infty}$ will be linear (and thus homogeneous) if $\lambda$ is affine.
>
> >What settings satisfy Assumption B.4? It seems that only linear setting satisfies this assumption. Can you elaborate?
>
> We thank the reviewer for the opportunity to elaborate more on this assumption. It is correct that, in x and y (the iterates), $H$ must have controlled growth so that $H_c$ converges as $c$ increases without bound. However, $H$ and $H_{\infty}$ do not have to be linear. Additionally, similar assumptions are commonly taken in stability works (see chapter 3 of Borkar 2009, Liu et al. 2025). Perhaps we do not fully understand what the reviewer means by linear setting here, and we are happy to provide further clarification.
>
> Works referenced in this response:
>
> [Sutton’09] Sutton, R. S., et al. Fast gradient-descent methods for temporal-difference learning with linear function approximation. ICML, 2009.
>
> [Yu’17] Yu, H. On convergence of some gradient-based temporal-differences algorithms for off-policy learning. arXiv preprint, 2017.
>
> [Borkar’09] Borkar, V. Stochastic approximation: a dynamical systems viewpoint. Springer, 2009.
>
> [Liu’25] Liu, S., et al. The ode method for stochastic approximation and reinforcement learning with markovian noise. JMLR, 2025.

---

> > ### Author Rebuttal · Reviewer_fmwn · 2026-04-04
> >
> > My concerns are fully addressed. I remain positive about this submission.

---

> > > ### Author Response · Authors · 2026-04-06
> > >
> > > We are glad that all of the reviewer’s comments were addressed and that they remain positive about our work!

---

### Decision · Program_Chairs · 2026-04-30

**Decision:**

Accept (regular)

**Comment:**

While reviewers raised concerns regarding presentation, positioning, and clarity of the technical contributions, they agreed that the paper makes a substantive theoretical contribution to stochastic approximation. In particular, the work establishes stability and asymptotic convergence results for two-timescale stochastic approximation under Markovian noise without projections or compactness assumptions, and provides a nontrivial application to TDC with eligibility traces, addressing a long-standing gap in the literature.

Overall, 2 out of 3 reviewers recommend acceptance (two accepts, one weak reject), and I support acceptance.

The main concerns, regarding positioning relative to prior work, clarity of the key technical ideas (in particular Lemma 4.1), and presentation of assumptions, are valid but largely addressed in the rebuttal, with reviewers acknowledging improvements and, in some cases, increasing their scores.

While some exposition issues remain, they do not detract from the core contribution. The paper fills an important theoretical gap and provides a useful foundation for analyzing two-timescale algorithms under realistic Markovian sampling in reinforcement learning.